# The Cannabinoid CB_1_ Receptor Inverse Agonist/Antagonist SR141716A Activates the Adenylate Cyclase/PKA Signaling Pathway Among Other Intracellular Emetic Signals to Evoke Vomiting in Least Shrews (*Cryptotis parva*)

**DOI:** 10.3390/ijms26209884

**Published:** 2025-10-11

**Authors:** Yina Sun, Louiza Belkacemi, Weixia Zhong, Zollie Daily, Nissar A. Darmani

**Affiliations:** Department of Basic Medical Sciences, College of Osteopathic Medicine of the Pacific, Western University of Health Sciences, 309 East Second Street, Pomona, CA 91766, USA; yinasun@westernu.edu (Y.S.); lbfgb@missouri.edu (L.B.); wzhong@scripps.edu (W.Z.); zollie.daily@westernu.edu (Z.D.)

**Keywords:** cannabinoid CB_1_ receptors, SR141716A, emesis, intracellular signaling, least shrew, dorsal vagal complex

## Abstract

Intracellular emetic signals involved in the cannabinoid CB_1_ receptor inverse agonist/antagonist SR141716A were investigated. SR141716A (20 mg/kg, i.p.)-evoked vomiting occurred via both the central and peripheral mechanisms. This was accompanied by robust emesis-associated increases in the following: (i) c-*fos*- and phospho-glycogen synthase kinase-3α/β (p-GSK-3αβ)-expression in the shrew’s dorsal vagal complex (DVC), (ii) phospho-extracellular signal-regulated kinase1/2 (p-ERK_1/2_) expression in both the DVC and jejunal enteric nervous system, and (iii) time-dependent upregulation of cAMP levels and phosphorylation of protein kinase A (PKA), protein kinase B (Akt), GSK-3α/β, ERK_1/2_, and protein kinase C αβII (PKCαβII) in the brainstem. SR141716A-evoked emetic parameters were attenuated by diverse inhibitors of the following: PKA, ERK_1/2_, GSK-3, phosphatidylinositol 3-kinase (PI3K)-Akt pathway, phospholipase C (PLC), PKC, Ca^2+^/calmodulin-dependent protein kinase II (CaMKII), L-type Ca^2+^ channel (LTCC), store-operated Ca^2+^ entry (SOCE), inositol trisphosphate receptor (IP3R), ryanodine receptor (RyRs), both 5-HT_3_-, and D_2/3_-receptor antagonists, and the transient receptor potential vanilloid 1 receptor (TRPV1R) agonist. SR141716A appears to evoke vomiting via inverse agonist activity involving emesis-associated kinases, including cAMP/PKA, ERK_1/2_, PI3K/Akt/GSK-3, PLC/PKCαβII, and CaMKII, which depend upon Ca^2+^ mobilization linking extracellular Ca^2+^ entry via plasma membrane Ca^2+^ channels (LTCC, SOCE, TRIPV1R) and intracellular Ca^2+^ release via IP3Rs and RyRs. The 5-HT_3_, NK_1_, and D_2/3_ receptors also contribute to SR141716A-mediated vomiting.

## 1. Introduction

Emesis is a protective reflex that rapidly expels ingested toxins from the gastrointestinal tract. The pathophysiology of nausea and vomiting indicates that emetic signaling processes are controlled by a balance amongst the central nervous system (CNS) [dorsal vagal complex emetic nuclei (DVC) in the brainstem, i.e., the nucleus tractus solitarius (NTS), the dorsal motor nucleus of the vagus (DMNX), and area postrema (AP)], the vagus, and the gastrointestinal enteric nervous system (ENS) [1]. The NTS neurons receive visceral sensory information, whereas the DMNX is the nuclei of origin of vagal motor fibers [2]. Both the AP and the NTS have fenestrated capillaries, which allow the diffusion of some circulating chemicals into the brainstem [3].

The endocannabinoid system consists of the following: (i) two well-investigated endocannabinoids, namely anandamide (AEA) and 2-arachidonoylglycerol (2-AG), (ii) two well-studied cannabinoid receptors, CB_1_ and CB_2_, and (iii) the enzymes that regulate the synthesis and catabolism of endocannabinoids [4]. The CB_1_ receptor is distributed throughout the body, and is predominantly found in neurons of the central and peripheral nervous systems [5,6,7], including the DVC emetic nuclei in the brainstem, as well as the gastrointestinal ENS [8,9,10]. Animal studies have shown that cannabinoid CB_1_/CB_2_ receptor agonists prevent chemotherapy-induced emesis via the activation of cannabinoid CB_1_ receptors since their antiemetic activity is countered by low to intermediate doses of the CB_1_ receptor inverse agonist/antagonist SR141716A [N-piperidino-5-(4-chlorophenyl)-1-(2,4-dichloro-phenyl)-4-methylpyrazole-3-carboxamide] [8,11,12,13]. The inverse agonist action and/or possible blockade of CB_1_ receptors may induce vomiting [14]. We have already shown that large doses (>10 mg/kg) of SR141716A by itself evokes emesis in least shrews in a dose-dependent manner [14,15]. Its 20 mg/kg dose was found to cause vomiting in all tested shrews [14], which corresponds to an equivalent human dose of 54 mg, assuming a human weight of 60 kg [16]. In clinical trials, a diverse range of doses of SR141716A (1–90 mg per day) have been utilized, with a dose of 20 mg/day being associated with a higher incidence of vomiting relative to placebo in randomized trials [17,18]. SR141716A-induced vomiting can be inhibited to various degrees by the following drugs: (1) structurally different classes of cannabinoid CB_1/2_ receptor agonists, including delta (9)-tetrahydrocannabinol (Δ^9^-THC) [14]; (2) the inhibitor of the mammalian target of rapamycin (mTOR), temsirolimus [19]; (3) the neurokinin NK_1_ receptor antagonist, netupitant [20]; and (4) the adrenergic α_2_-receptor agonists clonidine and dexmedetomidine [21].

CB_1_ receptors mainly act via G_i/0_ classes of G proteins to suppress the adenylate cyclase (AC)/protein kinase A (PKA) signaling system, which reduces cyclic adenosine monophosphate (cAMP) production; however, CB_1_ receptors can also exert their action via other mechanisms, including the activation of potassium and direct suppression of voltage-gated calcium channels (VGCCs) such as L-, N- and P/Q-type calcium channels [22,23,24]. In addition, CB_1_ receptor stimulation also suppresses L-type Ca^2+^ currents via cAMP/PKA, Ca^2+^/calmodulin-dependent protein kinase II (CaMKII) and mitogen-activated protein kinases/extracellular signal-regulated kinase (MAPK/ERK) signaling pathways, and T-type Ca^2+^ currents through the CaMKII pathway [25]. Downstream signaling of the cannabinoid CB_1_ receptor is also associated with the activation of other protein kinases such as phosphatidylinositol-3-kinase (PI3K)/protein kinase B (Akt), phospholipase C (PLC), and MAPK p42/p44 and p38 [22,26]. On the other hand, selective CB_1_ receptor antagonists/inverse agonists such as AM-251, SR141716A, or MJ08 may induce: (i) increases in cAMP levels in some rodent and human brain regions as well as in cell lines, followed by significant increases in PKA activity [27,28,29]; (ii) activation of ERK_1/2_, and cAMP response element-binding (CREB) protein phosphorylation [30]; (iii) increases in c-*fos* expression in several regions of rodent brain [31]; (iv) increases in calcium influx through the cell membrane [32,33,34]; (v) suppression of protein synthesis when intracellular calcium is chelated [35]; and (vi) the release of emesis-evoking neurotransmitters such as serotonin (5-HT), dopamine (DA), and substance P (SP) [15,36]. Many of these effects are associated with vomiting [1,37,38,39]. However, to date, intracellular emetic signals involved in SR141716A-evoked vomiting have not yet been fully investigated.

Thus, in the current study, we focused on the following: (1) the central and peripheral involvement of emetic loci underlying a fully effective emetic dose of SR141716A (20 mg/kg, i.p.) [14] by means of c-*fos* and phospho-ERK_1/2_ immunohistochemistry to demonstrate whether SR141716A can activate the emetic DVC area in the brainstem, the peripheral ENS in the jejunum, or both; (2) SR141716A-induced release of 5-HT and SP, as well as phospho-glycogen synthase kinase-3α/β (GSK-3α/β) immunohistochemistry in the brainstem DVC; (3) the cAMP levels and time-dependent profile of PKA (Thr197) activation, as well as phosphorylation of Akt (Ser473), GSK-3α/β (Ser21/9), ERK_1/2_ (Thr202/204), and PKCα/βII (Thr638/641) in the least shrew brainstem at 15 and 30 min post administration of SR141716A (20 mg/kg, i.p.); (4) the receptor/signaling mechanisms by which SR141716A (20 mg/kg, i.p.) may evoke emesis via the use of diverse antiemetics, including the antagonists/inhibitors of (i) ERK_1/2_ (U0126 and PD98059) [40], (ii) GSK-3 (AR-A014418 and SB216763) [41], (iii) PI3K/Akt pathway (LY294002) [42], (iv) PLC (U73122) [43], (v) protein kinase C (PKC; GF109203X) [43], (vi) PKA (H-89) [44], (vii) CaMKII (KN62) [45], (viii) L-type Ca^2+^ channel (LTCC; nifedipine and amlodipine) [46], (ix) store-operated Ca^2+^ entry (SOCE; YM-58483 and MRS-1845) [42], (x) the inositol trisphosphate receptor (IP3R; 2-APB) [45], (xi) ryanodine receptors (RyRs) Ca^2+^-release channels located on the endoplasmic reticulum (ER) (dantrolene) [45], (xii) the classical antiemetics acting on the cell membrane receptors such as the 5-HT_3_ receptor (palonosetron) [45] and dopamine D_2/3_ receptor (sulpiride) [47], as well as (xiii) the transient receptor potential vanilloid 1 receptor (TRPV1R) agonist, resiniferatoxin (RTX) [48].

## 2. Results

### 2.1. Acute Exposure to SR141716A Did Not Affect the Expression of CB_1_ Receptors in the Least Shrew Brainstem DVC

The brainstem DVC constitutes the primary emetic nuclei including the AP, NTS, and DMNX. We determined the expression of the CB_1_ receptors by immunostaining 20 μm shrew brainstem sections at 30 min post vehicle (*n* = 5)- and SR141716A (20 mg/kg, i.p., *n* = 5)-treatments. In the least shrew brainstem DVC area, the most highly expressed CB_1_ receptors were in the dorsomedial region of the NTS and the DMNX. A lower density of CB_1_ receptors was present in the adjacent subnuclei of NTS and AP (Figure 1). Figure 1A,B,H show that the CB_1_ receptors were present in the shrew brainstem DVC sections from vehicle-treated controls, with mean gray values of 21.25 ± 1.67, 31.7 ± 2.71, and 25.84 ± 1.79 in the AP, NTS, and DMNX, respectively. Compared to the vehicle-treated control group, SR141716A (20 mg/kg, i.p.) did not cause any significant change in the mean gray values of the CB_1_ receptors. The mean results were 25.57 ± 1.31 in the AP (*p* = 0.0952 vs. vehicle), 32.44 ± 1.07 in the NTS (*p* = 0.6905 vs. vehicle), and 28.24 ± 0.67 in the DMNX (*p* = 0.3095 vs. vehicle), respectively (Figure 1H; unpaired *t*-test).

### 2.2. Administration of SR141716A (20 mg/kg, i.p.) Activates the Brainstem Emetic DVC

#### 2.2.1. SR141716A Administration Significantly Increased c-*fos* Expression in the Brainstem DVC Emetic Nuclei

C-*fos* induction is a recognized tool for the evaluation of neuronal activation following peripheral stimulation with an agonist [49]. Thus, we performed immunohistochemistry to examine whether c-*fos* expression can be evoked following administration of SR141716A (20 mg/kg, i.p.). Figure 2A,B show that few c-*fos*-positive cells were observed in the DVC emetic nuclei in shrew brainstem sections from vehicle-treated controls, with mean values of 25.2 ± 7.34, 69.2 ± 21.18, and 54.6 ± 10.46 in the AP, NTS, and DMNX, respectively (Figure 2M). Relative to the vehicle-treated control group, a 20 mg/kg (i.p.) dose of SR141716A caused significant increases in c-*fos* expression; the average numbers of c-*fos*-positive cells increased to 70 ± 9.38 in the AP (*p* = 0.0055 vs. vehicle), 198.6 ± 33.33 in the NTS (*p* = 0.0112 vs. vehicle), and 99.4 ± 12.33 in the DMNX (*p* = 0.0243 vs. vehicle), respectively (Figure 2C,D,M; unpaired *t*-test). The c-*fos* expression of neurons in the ENS (located in the jejunum) was also examined, and the number of c-*fos* cells in the ENS was counted, double-labeling with NeuN. The ENS demonstrated a low c-*fos* staining signal (Figure 2E–L), with the mean number of c-*fos*/NeuN-positive cells being 2.88 ± 0.54 in the vehicle-treated group and 2.96 ± 0.59 for the SR141716A-treated group (Figure 2N). Statistical analysis showed no significant difference for jejunal c-*fos* expression between the vehicle- and SR141716A-treated groups (*p* = 0.9185; unpaired *t*-test) (Figure 2N).

#### 2.2.2. SR141716A Administration Evokes ERK_1/2_ Phosphorylation in Brainstem DVC and the Jejunum

ERK_1/2_ phosphorylation has also been used to indicate neuronal activation [50]. To determine the role of ERK_1/2_ in SR141716A-induced emesis, immunohistochemistry was performed to examine ERK_1/2_ phosphorylation evoked by SR141716A (20 mg/kg, i.p.). Figure 3 shows that at 15 min post SR141716A administration, significant increases in the phosphorylation of ERK_1/2_ occurred in the brainstem DVC and the ENS located in the jejunum. As shown in Figure 3A,C,D,G,H, in vehicle-treated shrews, the mean gray values for phospho-ERK_1/2_ expression were 9.28 ± 0.64/µm^2^, 6.53 ± 0.51/µm^2^, and 8.26 ± 0.83/µm^2^ in the AP, DMNX, and NTS in the brainstem DVC, as well as 3.44 ± 0.57/µm^2^ in the ENS in the jejunum, respectively. Following vomiting induced by SR141716A, the mean gray values for phospho-ERK_1/2_ expression were significantly increased to 13.07 ± 0.59 in the AP (*p* = 0.0048 vs. vehicle), 12.33 ± 1.04 in DMNX (*p* = 0.0024 vs. vehicle), and 13.11 ± 0.82 in the NTS (*p* = 0.0061 vs. vehicle), as well as 14.91 ± 3.16/µm^2^ (*p* = 0.0117 vs. vehicle) in the jejunum, respectively (Figure 3B,E–H; unpaired *t*-test).

#### 2.2.3. SR141716A Administration Induces GSK-3α/β Phosphorylation in Brainstem DVC

We performed immunohistochemistry to examine GSK-3α/β phosphorylation 15 min post administration of vehicle or SR141716A (20 mg/kg, i.p.). Figure 4A–D,I,J show that in the brainstem DVC from vehicle-treated shrews, the mean gray values for phospho-GSK-3α expression were 23.09 ± 2.71/µm^2^, 22.27 ± 2.65/µm^2^, and 25.77 ± 3.31/µm^2^ in the AP, DMNX, and NTS; the mean gray values for phospho-GSK-3β expression were 20.56 ± 1.92/µm^2^, 19.92 ± 2.33/µm^2^, and 20.38 ± 2.29/µm^2^ in the AP, DMNX and NTS, respectively. At 15 min following the administration of SR141716A (20 mg/kg, i.p.), the mean gray values for phospho-GSK-3α expression were significantly increased to 34 ± 3.06 in the AP (*p* = 0.0234 vs. vehicle), 33.13 ± 3.91 in DMNX (*p* = 0.0442 vs. vehicle), and 40.38 ± 5.26 in NTS (*p* = 0.0405 vs. vehicle). The mean gray values for phospho-GSK-3β expression were also significantly increased to 32.17 ± 2.54 in the AP (*p* = 0.0045 vs. vehicle), 29.14 ± 1.41 in DMNX (*p* = 0.0069 vs. vehicle), and 30.45 ± 2.3 in NTS (*p* = 0.0113 vs. vehicle), respectively (Figure 4E–J; unpaired *t*-test).

#### 2.2.4. SR141716A Lacks Significant Effects on the Release of 5-HT and SP in Brainstem DVC

Previous studies have shown that emetogen (e.g., cisplatin, FPL64176 or thapsigargin)-evoked emesis was accompanied by increases in 5-HT or SP-immunoreactivity in brainstem emetic nuclei [51,52,53]. In this study, we tested whether 15 min exposure to SR141716A (20 mg/kg, i.p.) could also increase 5-HT and SP-immunoreactivity in the brainstem DVC containing emetic nuclei AP, NTS, and DMNX. Figure 5A shows that the current results are similar to those we described in a previous study [53]: the highest density of 5-HT-positive fibers is in the dorsomedial subdivision of the NTS. A lower of 5-HT-immunoreactive profile was noted in the adjacent subnuclei of the NTS and DMNX, and the AP had fewer 5-HT-containing neurons in the vehicle-treated control group. Figure 5B shows that SP-immunoreactive fibers were found in their highest concentrations within the DMNX (and to a lesser extent in the NTS), but rarely in the AP of the vehicle-treated control group. In vehicle-treated shrews, the mean gray values for 5-HT-immunoreactivity were 23.26 ± 2.02/µm^2^, 27.56 ± 1.8/µm^2^, and 35.06 ± 3.16/µm^2^ in the AP, DMNX, and NTS, and the mean gray values for SP-immunoreactivity were 10.59 ± 0.16/µm^2^, 27.68 ± 2.31/µm^2^, and 17.29 ± 0.99/µm^2^ in the AP, DMNX, and NTS, respectively (Figure 5A–D,I,J). Statistical analysis showed no significant difference for 5-HT and SP-immunoreactivities between the vehicle- and SR141716A-treated groups [51,52,53] 15 min post injection. In the SR141716A-treated group, the mean gray values for 5-HT-immunoreactivity were 21.21 ± 1.82 in the AP (*p* = 0.4792 vs. vehicle), 27.97 ± 1.34 in DMNX (*p* = 0.861 vs. vehicle), and 33.89 ± 2.61 in NTS (*p* = 0.7841 vs. vehicle); the mean gray values for SP-immunoreactivity were 9.74 ± 0.36 in the AP (*p* = 0.0757 vs. vehicle), 27.34 ± 2.67 in DMNX (*p* = 0.9254 vs. vehicle), and 15.82 ± 0.72 in NTS (*p* = 0.2746 vs. vehicle), respectively (Figure 5E–J; unpaired *t*-test).

### 2.3. SR141716A Administration Increased Brainstem cAMP Levels

Increased brain cAMP levels following administration of phosphodiesterase 4 inhibitors such as rolipram can lead to vomiting in different vomit-competent species, including ferrets and least shrews [54,55]. Therefore, we investigated possible changes in cAMP tissue levels in the brainstem following SR141716A treatment at several time points, as shown in Figure 6A,B. Relative to the 0 min control time point, cAMP tissue levels were significantly increased in shrews’ brainstems at 15 min (*p* < 0.01 vs. 0 min) and 30 min (*p* < 0.05 vs. 0 min) post administration of SR141716A (20 mg/kg, i.p.) (Figure 6A), which also evoked vomiting in all tested shrews [14]. However, administration of a non-emetic dose of SR141716A (1 mg/kg, i.p.) failed to induce a change in the cAMP tissue levels in the brainstem of least shrews (Figure 6B).

### 2.4. SR141716A Administration Increased Phosphorylation Levels of Emesis-Associated Proteins in Least Shrew Brainstems

To assess the time-dependent profile of emesis-associated PKAα/β (Thr197), Akt (Ser473)-, GSK-3α/β (Ser21/9)-, ERK_1/2_ (Thr202/204)-, and PKCαβII (Thr638/641) protein phosphorylations, different groups of shrews were sacrificed at 15 and 30 min following SR141716A (20 mg/kg, i.p.) administration. The total and phosphorylated forms of each protein were quantified in the brainstems of the shrews by Western blot.

Elevated cAMP levels can promote the activation of PKA [56,57]. As shown in Figure 6C,D, PKAα/β (Thr197) phosphorylation significantly peaked at 30 min post SR141716A (20 mg/kg, i.p.) treatment (*p* < 0.01 vs. 0 min). In addition, SR141716A significantly increased Akt (Ser473) and PKCαβII (Thr638/641) phosphorylations, which peaked at 15 min (*p* < 0.05 and *p* < 0.01 vs. 0 min) post administration, and then quickly returned to basal levels at 30 min. Likewise, SR141716A rapidly evoked the phosphorylation of GSK-3α/β (Ser21/9) and ERK_1/2_ (Thr202/204), which significantly peaked at 15 min post injection; these elevated levels persisted up to 30 min post treatment (*p* < 0.05 and *p* < 0.01 vs. 0 min, respectively). Under the same conditions, the total PKAα/β, Akt, GSK-3α/β, ERK_1/2_, and PKCαβII levels remained similar (Figure 6C).

### 2.5. Behavioral Studies

#### 2.5.1. SR141716A (20 mg/kg, i.p.)-Induced Vomiting Involves Emesis-Associated PKA, ERK_1/2_, GSK-3α/β, PI3K, PLC, and PKCαβII Signals

We initially examined the antiemetic potential of the selective PKA inhibitor H-89 (0, 0.1, 1, 2, and 5 mg/kg, i.p.) against vomiting caused by SR141716A. H-89 (0, 0.1, 1, 2, and 5 mg/kg, i.p.) suppressed the frequency of SR141716A-induced vomiting in a dose-dependent manner (KW _(4, 58)_ = 22.93; *p* = 0.0001). Indeed, the frequency of the induced vomiting was significantly reduced at doses of 1 (*p* = 0.0084), 2 (*p* = 0.0018), and 5 mg/kg (*p* < 0.0001), with an ID_50_ value of 0.1424 (0.03928–0.5243) mg/kg (Figure 7A). Likewise, significant decreases (χ^2^ _(4, 58)_ = 22.52; *p* = 0.0002) in the percentage of animals vomiting were noted following doses of 0.1 (38.46%; *p* = 0.0128), 1 (58.33%; *p* = 0.0012), 2 (64.29%; *p* = 0.0004), and 5 mg/kg (90.91%; *p* < 0.0001), having an ID_50_ value of 0.6187 (0.2149–1.311) mg/kg (Figure 7B).

Varying doses of two ERK_1/2_ inhibitors, namely U0126 (0, 1, 5, and 10 mg/kg, i.p.) and PD98059 (0, 1, 2.5, and 5 mg/kg, i.p.), were tested next against SR141716A-induced emesis. Relative to the vehicle-pretreated control group, U0126 significantly reduced the mean frequency of SR141716A-induced vomiting during the 30 min observation period [(KW _(3, 32)_ = 20.1; *p* = 0.0002)], and significant reductions in the frequency of vomiting were observed following doses of 5 and 10 mg/kg (*p* = 0.0009 and 0.0003, respectively), with an ID_50_ value of 0.9498 (0.2985–2.312) mg/kg (Figure 7C). Moreover, U0126 significantly protected shrews from the evoked vomiting (χ^2^ _(3, 32)_ = 15.96; *p* < 0.0001), with significant reductions in the percentage of shrews vomiting occurring at doses of 5 mg/kg (75%; *p* = 0.0005) and 10 mg/kg (77.78%; *p* = 0.0003), with an ID_50_ value of 2.36 (1.077–4.829) mg/kg (Figure 7D). PD98059 also dose-dependently reduced the mean frequency of SR141716A-induced vomiting [(KW _(3, 35)_ = 12.52; *p* = 0.0058)], with significant reductions occurring following doses of 2.5 and 5 mg/kg (*p* = 0.0478 and 0.0019, respectively), with an ID_50_ value of 1.017 (0.3104–2.579) mg/kg (Figure 7E). Significant decreases in the percentage of shrews vomiting (χ^2^ _(3, 35)_ = 13.76; *p* = 0.0033) occurred following doses of 1, 2.5, and 5 mg/kg (50%; *p* = 0.0098, 58.33%; *p* = 0.0024, and 85.71%; *p* = 0.0003, respectively), with an ID_50_ value of 1.839 (0.671–4.406) mg/kg (Figure 7F).

We also investigated the antiemetic potential of the selective GSK-3αβ inhibitors AR-A014418 and SB216763 against vomiting evoked by SR141716A. As shown in Figure 7G, AR-A014418 (0, 1, 2.5, 5 and 10 mg/kg) caused a dose-dependent decrease in the frequency of SR141716A-evoked vomiting (KW _(4, 37)_ = 23.28; *p* = 0.0001), with significant reductions occurring at doses of 5 (*p* = 0.0015) and 10 mg/kg (*p* = 0.0002), and with an ID_50_ value of 1.609 (0.8286–2.921) mg/kg. The chi-square test indicated that the number of shrews vomiting in response to SR141716A was also significantly attenuated (χ^2^ _(4, 37)_ = 16.46; *p* = 0.0025) by AR-A014418, with 50% protection following a dose of 5 mg/kg (*p* = 0.0135) and 87.71% protection with 10 mg/kg (*p* = 0.0004), with an ID_50_ value of 4.565 (2.526–8.421) mg/kg (Figure 7H). Figure 7I,J show that SB216763 (0, 0.25, 1 and 2.5 mg/kg) also suppressed vomiting caused by SR141716A in a dose-dependent manner. The mean frequency of SR141716A-induced emesis was significantly reduced (KW _(3, 24)_ = 11.04; *p* = 0.0115) at 2.5 mg/kg (*p* = 0.0058), with an ID_50_ value of 1.376 (0.3428–6.61) mg/kg (Figure 7I). The percentage of shrews vomiting (χ^2^ _(3, 24)_ = 12.18; *p* = 0.0068) was significantly reduced at doses of 1 mg/kg (50%, *p* = 0.0455) and 2.5 mg/kg (87.5%, *p* = 0.0012), with an ID_50_ value of 0.7153 (0.3069–1.592) mg/kg (Figure 7J).

Then, we investigated the antiemetic effect of the selective PI3K inhibitor LY294002 (0, 1, 2.5 and 10 mg/kg, i.p.) against SR141716A-induced vomiting. The administration of LY294002 (0, 1, 2.5 and 10 mg/kg, i.p.) dose-dependently suppressed vomiting (KW _(3, 20)_ = 10.3; *p* = 0.0162) caused by SR141716A. The frequency of SR141716A-induced emesis was significantly reduced at 10 mg/kg (*p* = 0.0057), with an ID_50_ value of 0.9776 (0.3054–2.441) mg/kg (Figure 7K). Likewise, significant decreases (χ^2^ (3, 20) = 9.067; *p* = 0.0284) in the percentage of animals vomiting were also noted at 10 mg/kg (83.33%; *p* = 0.0034), having an ID_50_ value of 3.072 (1.181–8.127) mg/kg (Figure 7L).

Pre-treatment with selective PLC inhibitor U73122 (0, 1, 2.5, 5 and 10 mg/kg, i.p.) significantly reduced both the frequency (KW _(4, 34)_ = 23.54; *p* < 0.0001) and percentage (χ^2^ _(4, 34)_ = 19.96; *p* = 0.0005) of shrews vomiting in a dose-dependent manner in response to the administration of the SR141716A, with respective ID_50_ values of 0.6202 (0.1661–1.419) mg/kg and 1.078 (0.4857–2.081) mg/kg (Figure 7M,N). The mean frequency of SR141716A-induced emesis was significantly reduced at 2.5 (*p* = 0.0047) and 5 mg/kg (*p* = 0.0005), with complete suppression at 10 mg/kg (*p* = 0.0002; Figure 7M). Significant decreases in the percentage of shrews vomiting were also noted at doses of 1 (50%; *p* = 0.0112), 2.5 (62.5%; *p* = 0.0033), and 5 mg/kg (85.71%; *p* = 0.0003), with complete protection at 10 mg/kg (100%; *p* < 0.0001; Figure 7N).

Next, the antiemetic effect of varying doses of the selective PKCαβII inhibitor GF109203X (0, 0.1, 1, 5, and 10 mg/kg, i.p.) was tested against vomiting caused by SR141716A. As shown in Figure 7O,P, GF109203X (0, 0.1, 1, 5 and 10 mg/kg, i.p.) caused dose-dependent decreases in the mean frequency (KW _(4, 39)_ = 23.23; *p* = 0.0001) of vomiting, with significant reductions occurring at doses of 1 mg/kg (*p* = 0.0095) and 5 mg/kg (*p* = 0.001), and with complete suppression at 10 mg/kg (*p* < 0.0001), with an ID_50_ value of 0.08406 (0.02201–0.268) mg/kg. The percentage of shrews vomiting was also reduced in a dose-dependent fashion (χ^2^ _(4, 39)_ = 19.06; *p* = 0.0008), with significant reductions at 0.1 (33.33%; *p* = 0.0466), 1 (55.56%; *p* = 0.006), and 5 mg/kg (66.67%; *p* = 0.0018), and complete protection at 10 mg/kg (100%; *p* < 0.0001), with an ID_50_ value of 0.7633 (0.1972–2.153) mg/kg (Figure 7O,P).

#### 2.5.2. Ca^2+^ Channel Modulators Reduce SR141716A-Induced Emesis

##### Ca^2+^/Calmodulin-Dependent Kinase II (CaMKII)

CaMKII autophosphorylates in response to elevated intracellular Ca^2+^ levels and functions as an intracellular signaling protein [58]. The antiemetic potential of varying doses of the selective CaMKII inhibitor KN62 (0, 1, 2.5, 5 and 10 mg/kg) was examined against vomiting caused by SR141716A (20 mg/kg, i.p.). The mean frequency of SR141716A-induced emesis was significantly reduced (KW _(4, 51)_ = 14.46; *p* = 0.006) by KN62 at doses of 2.5 (*p* = 0.0018), 5 (*p* = 0.0291) and 10 mg/kg (*p* = 0.0107), with an ID_50_ value of 1.197 (0.47–2.549) mg/kg (Figure 8A). The percentage of shrews vomiting was also significantly reduced (χ^2^ _(4, 51)_ = 10.37; *p* = 0.0346) at doses of 1 (50%; *p* = 0.0098), 2.5 (63.64%; *p* = 0.002), 5 (53.85%; *p* = 0.0054) and 10 mg/kg (50%; *p* = 0.0087), with an ID_50_ value of 2.748 (1.149–6.176) mg/kg (Figure 8B).

##### Ca^2+^ Channel Blockers at the Cell Membrane Reduce SR141716A-Induced Emesis

The dose-dependent broad-spectrum antiemetic efficacy of LTCC inhibitors, namely nifedipine (short-acting) and amlodipine (long-acting), has previously been demonstrated in the least shrew model of emesis in our laboratory [45,53]. In this study, pre-treatment with nifedipine (0, 0.1, 0.25, 0.5, 1, and 5 mg/kg, s.c.) 30 min before SR141716A (20 mg/kg, i.p.) administration, caused a dose-dependent suppression of the mean vomit frequency (KW _(5, 30)_ = 31.17, *p* < 0.0001) and the percentage of shrews vomiting (χ^2^ _(5, 30)_ = 30.6; *p* < 0.0001) in response to SR141716A administration (Figure 8C,D). Indeed, relative to the control group pre-treated with nifedipine, s.c. injection of nifedipine at 0.5, 1, and 5 mg/kg completely blocked both the mean vomit frequency (*p* = 0.0017, *p* = 0.0017, and *p* = 0.0017, respectively) (Figure 8C), and the percentage of shrews vomiting (100%, *p* = 0.0005; 100%, *p* = 0.0005; and 100%, *p* = 0.0005; respectively) in response to SR141716A (Figure 8D). The ID_50_ values for vomit frequency and percentage of vomiting were 0.1479 (0.08858–0.2363) mg/kg and 0.2419 (0.1477–0.3879) mg/kg, respectively. Likewise, pre-treatment with amlodipine (0, 0.25, 0.5, 1, 5 and 10 mg/kg, i.p.) significantly attenuated the mean frequency of SR141716A (20 mg/kg, i.p.)-induced vomiting in a dose-dependent manner (KW _(5, 37)_ = 23.34; *p* = 0.0003), with significant reductions occurring at doses of 1 (*p* = 0.0119) and 5 mg/kg (*p* = 0.0028), and complete suppression at 10 mg/kg (*p* = 0.0008; ID_50_ value: 0.5131 (0.2374–1.063) mg/kg; Figure 8E). In addition, the percentage of shrews vomiting in response to SR141716A was also suppressed by amlodipine (χ^2^ _(5, 37)_ = 20.55; *p* < 0.0001) at doses of 1 (62.5%, *p* = 0.0048) and 5 mg/kg (75%, *p* = 0.0012), with complete protection at 10 mg/kg (100%, *p* = 0.0001), and with an ID_50_ value of 0.8631 (0.433–1.756) mg/kg (Figure 8F).

Calcium is stored in the endoplasmic/cytoplasmic reticulum; refilling of the stocks occurs via SOCE, which is an important route by which Ca^2+^ mobilizes [59]. In this study, we examined the antiemetic potential of two SOCE inhibitors, namely YM58483 and MRS-1845, against vomiting induced by SR141716A (20 mg/kg, i.p.). Administration of YM58483 (0, 0.1, 0.5, 1, and 2.5 mg/kg, i.p.) dose-dependently attenuated the frequency of SR141716A-induced vomiting (KW _(4, 48)_ = 21.32, *p* = 0.0003; ID_50_ value: 0.0993 (0.04362–0.1983) mg/kg), with significant attenuations observed at doses of 0.5 (*p* = 0.0179) and 1 mg/kg (*p* = 0.0067), and complete suppression occurring at 2.5 mg/kg (*p* < 0.0001; Figure 8G). Significant decreases in the percentage of animals vomiting (χ^2^ _(4, 48)_ = 20.32; *p* = 0.0004) also occurred at doses of 0.5 (50%; *p* = 0.0087) and 1 mg/kg (58.33%; *p* = 0.0034), and complete protection was observed at 2.5 mg/kg (100%; *p* < 0.0001), with an ID_50_ value of 0.4631 (0.234–0.8631) mg/kg (Figure 8H). As shown in Figure 8I,J, MRS-1845 (0, 0.01, 0.1, 0.5, 1 and 2.5 mg/kg) also caused dose-dependent decreases in the mean frequency of vomiting (KW _(5, 56)_ = 37.34; *p* < 0.0001) induced by SR141716A (20 mg/kg, i.p.;); significant reductions occurred doses of at 0.1 (*p* = 0.0014), 0.5 (*p* = 0.0003), and 1 mg/kg (*p* < 0.0001), with complete suppression at 2.5 mg/kg (*p* < 0.0001; ID_50_ value: 0.01104 (0.003802–0.03104) mg/kg; Figure 8I). Significant reductions (χ^2^ _(5, 56)_ = 30.72; *p* < 0.0001) in the percentage of animals vomiting was also observed following doses of 0.1 (45.45%; *p* = 0.0048), 0.5 (70%; *p* = 0.0002), and 1 mg/kg (77.78%; *p* < 0.0001), with complete protection at 2.5 mg/kg (100%; *p* < 0.0001; ID_50_ value: 0.1473 (0.0642–0.308) mg/kg; Figure 8J).

At low nanomolar doses, the transient receptor potential vanilloid 1 receptor channel agonist resiniferatoxin (RTX) has the capacity to completely block vomiting induced by diverse emetogens in least shrews [48]. Thus, the antiemetic efficacy of RTX was examined against SR141716A (20 mg/kg, i.p.;)-induced vomiting. As shown in Figure 8K,L, RTX (0, 0.25, 1, and 2.5 µg/kg, s.c.) caused potent and dose-dependent decreases in the frequency of SR141716A-iduced vomiting (KW _(3, 20)_ = 16.81; *p* = 0.0008). Significant reductions occurred at 1 (*p* = 0.0027), and complete suppression occurred at 2.5 µg/kg (*p* = 0.0008), with an ID_50_ value of 0.1595 (0.06772–0.3043) µg/kg (Figure 8K). The percentage of animals vomiting in response to SR141716A (χ^2^ _(3, 20)_ = 15.27; *p* = 0.0016) was also potently and significantly reduced following a dose of 1 µg/kg (83.33%, *p* = 0.0034) and complete protection occurred at 2.5 µg/kg (100%, *p* = 0.0005), with an ID_50_ value of 0.3216 (0.1372–0.6667) µg/kg (Figure 8L).

##### Intracellular Ca^2+^ Channel Blockers Reduce SR141716A-Induced Vomiting

Subsequently, we tested whether intracellular Ca^2+^ release channels such as IP3Rs and/or RyRs are involved in SR141716A (20 mg/kg, i.p.)-induced emesis. The IP3R inhibitor 2-APB (0, 0.5, 1 and 5 mg/kg, i.p.) significantly attenuated the mean frequency of SR141716A-induced vomiting in a dose-dependent manner (KW _(3, 20)_ = 18.02; *p* = 0.0004), with significant reductions occurring at doses of 1 (p = 0.0205) and 5 mg/kg (*p* < 0.0001; ID_50_ value: 0.1893 (0.1109–0.2886) mg/kg; Figure 9A). A significant decrease in the percentage of animals vomiting (χ^2^ _(3, 20)_ = 14.76; *p* = 0.002) occurred following a dose of 5 mg/kg (100%; *p* = 0.0005), with an ID_50_ value of 1.414 (0.698–2.931) mg/kg (Figure 9B).

Administration of the RyR inhibitor dantrolene (0, 1, 2.5 and 5 mg/kg) also suppressed the mean frequency of vomiting induced by SR141716A (KW _(3, 20)_ = 14.58; *p* = 0.0022), with significant reductions following doses of 2.5 (*p* = 0.0152) and 5 mg/kg (*p* = 0.001), and with an ID_50_ value of 0.7885 (0.3283–1.544) mg/kg (Figure 9C). Significant decreases (χ^2^ _(3, 20)_ = 13.33; *p* = 0.004) in the percentage of animals vomiting were also noted following doses of 2.5 (66.67%; *p* = 0.0143) and 5 mg/kg (100%; *p* = 0.0005), with an ID_50_ value of 1.214 (0.5072–2.534) mg/kg (Figure 9D).

##### 2.5.3. Classical Serotonin 5-HT_3_- and Dopamine D_2/3_-Receptor Antagonists Reduce SR141716A-Induced Emesis

To investigate the role of serotonergic 5-HT_3_- and dopaminergic D_2/3_ receptors in SR141716A-induced vomiting, different groups of shrews were pre-treated with either the 5-HT_3_ receptor antagonist palonosetron (0, 0.25, 1, 2.5 and 5 mg/kg, s.c.), or the D_2/3_ receptor antagonist sulpiride (0, 1, 2 and 4 mg/kg, s.c.) 30 min prior to SR141716A (20 mg/kg, i.p.) injection. Palonosetron pre-treatment reduced both the mean emesis frequency (KW _(4, 25)_ = 10.17, *p* = 0.0377) and the percentage of shrews vomiting (χ^2^ _(4, 25)_ = 8.571; *p* = 0.0728) in response to SR141716A, as shown in a U-shaped dose–response curve. Indeed, a significant reduction in mean vomit frequency occurred only at 1 mg/kg (*p* = 0.0087; ID_50_ value: 0.2009 (0.004264–0.9339) mg/kg; Figure 10A). Significant decreases in the percentage of animals vomiting occurred at doses of 0.25 (50%; *p* = 0.0455), 1 (83.33%; *p* = 0.0034), 2.5 (50%; *p* = 0.0455), and 5 mg/kg (50%; *p* = 0.0455), with an ID_50_ value of 0.5475 (0.09105–2.673) mg/kg (Figure 10B). Pre-treatment with sulpiride (0, 1, 2 and 4 mg/kg, s.c.) significantly decreased the mean frequency of vomiting (KW _(3, 38)_ = 11.04; *p* = 0.0115) induced by SR141716A (20 mg/kg, i.p.). Significant suppression in the mean vomit frequency occurred only at 4 mg/kg (*p* = 0.0031; ID_50_ value: 1.867 (0.8786–3.857) mg/kg; Figure 10C). However, at the tested doses, sulpiride (0, 1, 2 and 4 mg/kg, s.c.) failed to significantly suppress the percentage of SR141716A-induced vomiting in the animals (χ^2^ _(3, 38)_ = 5.935; *p* = 0.1148; ID_50_ value: 5.642 (2.923–13.17) mg/kg) (Figure 10D).

## 3. Discussion

### 3.1. Acute Brief Exposure to SR141716A Did Not Alter the Cannabinoid CB_1_ Receptor Density in the Shrew Brainstem DVC Nuclei

Our current immunohistochemistry results show that the CB_1_ receptor is highly expressed in the dorsomedial region of the NTS and the DMNX of the brainstem in least shrews; this is expressed to a lesser extent in the adjacent subnuclei of the NTS and the AP in both vehicle- and SR141716A (20 mg/kg., i.p.)-treated shrews. Our previous findings support the current results since CB_1_ receptor immunoreactivity was found to be more prominent in the NTS and DMNX than in the AP region [60]. These findings are consistent with the report that CB_1_ receptor immunoreactivity is highly expressed in the DMNX and in the medial nucleus of the solitary tract (mNTS) in rats [61]. A short acute exposure to SR141716A does not alter CB_1_ receptor density in rodent brains [62]; likewise, SR141716A (20 mg/kg., i.p.) also failed to cause a significant change in the mean gray value of CB_1_ receptor immunoreactivity in the emetic nuclei of the brainstem DVC in least shrews.

### 3.2. SR141716A-Induced Vomiting Involves Multiple Intracellular Signaling Cascades

#### 3.2.1. SR141716A (20 mg/kg) Induces Vomiting Through Both Central and Peripheral Mechanisms

Both c-*fos* and p-ERK_1/2_ are classical tools used to evaluate neuronal activation following peripheral agonist stimulation, and have been validated as neuronal activation indicators in vivo [49,50]. Previous studies showed that the least shrew brainstem DVC displays significant c-*fos* expression following vomiting induced by diverse emetogens [19,43,52,53,63,64,65]. These findings are consistent with the present study in that vomiting evoked by SR141716A (20 mg/kg, i.p.) is followed by significant increases in c-*fos* expression in the shrew brainstem DVC (AP, NTS and DMNX), supporting the central activation of emetic nuclei.

We have also shown that p-ERK_1/2_ in the shrew brainstem DVC can be elicited by i.p. administration of different emetogens [42,43,46,52,53,64,65,66]. Consistent with these results, our current immunohistochemical findings demonstrate that following 15 min exposure to SR141716A, significant p-ERK_1/2_ occurred in all the shrew brainstem DVC emetic nuclei, which further supports our present findings that p-ERK_1/2_ rapidly and significantly peaked at 15 min post SR141716A exposure and persisted for up to 30 min. These findings also support the involvement of the central activation of DVC nuclei in the brainstem in the evoked emetic process. In addition, pronounced immunoreactivity in p-ERK_1/2_ was also observed in the jejunal ENS in shrews, suggesting the involvement of peripheral emetic loci in SR141716A-induced vomiting. However, c-*fos* expression in the ENS in the jejunum of shrews showed a non-significant weak response following SR141716A-induced vomiting. We speculate that the discrepancy in the gut may involve other signaling pathways, since c-*fos* is one downstream target of ERK_1/2_ [67,68]. Moreover, both ERK_1/2_ inhibitors, namely U0126 and PD 98059, used in the current study reduced the frequency of SR141716A-induced vomiting in a dose-dependent manner. Together with the behavioral evidence that Δ^9^-THC evokes its antiemetic effects via both central and peripheral mechanisms [69], our current findings suggest that SR141716A-induced vomiting involves both the central DVC and intestinal emetic loci.

#### 3.2.2. SR141716A-Induced Emesis Involves the cAMP/PKA Signaling Pathway

The role of the cAMP/PKA signaling pathway in emesis is well established since microinjection of cAMP analogs (e.g., 8-bromocAMP) or forskolin (an activator of AC, to enhance endogenous levels of cAMP) in the AP not only increases electrical activity of local neurons, but also induces vomiting in dogs [37]. Moreover, the phosphodiesterase type 4 (PDE4) inhibitor rolipram prevents the degradation of cAMP and elevates its intracellular levels [70,71], subsequently leading to the induction of vomiting in different species, such as ferrets and least shrews [21,38,55]. We have further demonstrated that rolipram-evoked vomiting involves increased levels of cAMP in the shrew brainstem; and inhibition of cAMP production significantly attenuates rolipram-induced emesis in least shrews [55]. It has been shown that SR141716A can also increase cAMP tissue levels in several regions of rat brains, as well as in the frontal cortex in human brains [29]. In the current study, SR141716A (20 mg/kg, i.p.) significantly increased cAMP tissue levels in the brainstem of shrews at 15 and 30 min post injection, whereas PKAα/β phosphorylation significantly peaked at 30 min post injection. Similar time-dependent changes in cAMP and PKA levels have previously been reported in the endothelial cells within human umbilical veins; cellular cAMP levels were significantly increased following 10 min exposure to SR141716A, and significant phosphorylation of PKA occurred at 30 min [72]. Furthermore, our current emesis findings show that the PKA inhibitor H-89 dose-dependently suppressed vomiting caused by SR141716A (20 mg/kg, i.p.), indicating that the cAMP/PKA signaling pathway is involved in SR141716A-induced vomiting. Moreover, a non-emetic dose of SR141716A (1 mg/kg; i.p.) failed to induce any change in cAMP tissue levels in the brainstems of least shrews, further supporting the involvement of cAMP/PKA activation in the production of induced emesis.

#### 3.2.3. SR141716A-Induced Vomiting Involves the PI3K/Akt Signaling Pathways

Following phospho-PI3K activation, phosphatidylinositol (3,4,5)-trisphosphate (PIP3) accumulates at the cell membrane, which then leads to the recruitment of Akt to the plasma membrane. Akt is phosphorylated at Thr308 together with Ser473, which ensures full Akt activation [73,74]. The activation of CB_1_ receptors is highly implicated in the regulation of mitogen-activated protein kinases (MAPKs) and the phosphoinositide 3-kinase (PI3K/Akt) pathway, which have important roles in determining neuronal death or survival, particularly in oxidative stress conditions [75]. We have previously demonstrated that the Akt signaling pathway is involved in emesis induced by several emetogens in least shrews [42,47,52]. In the current study, our Western blot results demonstrate that Akt (Ser473) phosphorylation peaked significantly after 15 min SR141716A (20 mg/kg, i.p.) exposure, which then returned to basal levels at 30 min in the shrew brainstem DVC. Our present results support the latter finding in that the selective PI3K inhibitor LY294002 significantly and dose-dependently suppressed both the frequency and percentage of shrews vomiting caused by SR141716A (20 mg/kg, i.p.). These findings validate the contribution of PI3K/Akt signaling to SR141716A-induced vomiting. Indeed, the inhibition of PI3K through LY294002 results in the dephosphorylation of Akt at Ser473 [76].

The stimulation of CB_1_ receptors leads to the activation of ERK_1/2_ kinases in a variety of cell types and appears to be cell-dependent [77]. Indeed, CB_1_-evoked activation of ERK_1/2_ proteins involves several mechanisms, including the activation of Gi/o proteins [77], PI3K [78], and the inhibition of AC and PKA [79]. The association of ERK_1/2_ signaling with vomiting triggered by the inhibition of Akt was uncovered using the selective ERK_1/2_ inhibitor U0126, which prevented emesis induced by the Akt inhibitor MK-2206 in least shews [65]. Injections of MK-2206 also caused significant ERK_1/2_ phosphorylation in both the brainstem DVC and the jejunal ENS in shrews [65]. These results are in line with previous work demonstrating that PI3K and Akt inhibitors significantly potentiated ERK_1/2_ phosphorylation [80].

#### 3.2.4. SR141716A-Induced Emesis Involves the PI3K/Akt/GSK-3 Signaling Pathway

GSK-3 is a downstream protein of the PI3K pathway activated by the Akt protein. Akt signaling can be activated by the phosphorylation of its downstream target protein GSK-3α/β at Ser21/9 and its subsequent inactivation [81], whereas Akt inhibition activates GSK-3 [82]. GSK-3 is a constitutively active protein, and its activity can be inhibited through the phosphorylation of Ser21 in GSK-3α and Ser9 in GSK-3β [83]. It has been proposed that GSK-3 phosphorylation in vivo may protect against excessive vomiting evoked by multiple emetogens [41]. In fact, phospho-GSK-3α/β Ser21/9 tissue levels in the shrew DVC and the jejunal ENS are upregulated following systemic administration of a variety of emetogens [39,41]. Furthermore, the administration of GSK-3 inhibitors AR-A014418 and SB21676 dose-dependently attenuates shrew vomiting in response to i.p. administration of these emetogens [41]. CB_1_ receptor agonists such as Δ^9^-THC respond via the PI3K/Akt/GSK-3 signaling system in a SR141716A-sensitive manner [84,85]. In the present study, Western blot results show that Akt (Ser473) phosphorylation significantly peaked at 15 min post SR 141716A (20 mg/kg, i.p.) injection, which then quickly returned to basal levels at 30 min, whereas p-GSK-3α/β (Ser21/9) rapidly and significantly peaked at 15 min post injection and persisted for up to 30 min. Our immunohistochemistry data also show that at 15 min post systemic administration of SR 141716A (20 mg/kg, i.p.), phosphorylation of both GSK-3α and GSK-3β subtypes significantly increased in the brainstem DVC (the AP/NTS/DMNV) relative to vehicle-treated control shrews. Furthermore, our behavioral data showed that the GSK-3 inhibitors, namely AR-A014418 and SB216763, dose-dependently attenuated both the frequency and percentage of shrews vomiting in response to i.p. administration of SR141716A.

#### 3.2.5. SR141716A Promotes Vomiting via the PLC/PKCαβII Pathway

G_q_ and G_βγ_ protein-dependent PLC activation is a crucial component in cellular signaling [86]. G_q/11_ protein activates PLC to generate inositol 1,4,5-trisphosphate (IP_3_) and diacylglycerol (DAG). Cytosolic IP_3_ subsequently increases cytosolic Ca^2+^ concentrations via the IP_3_R-mediated release of Ca^2+^ from ER calcium stores. This then triggers PKC phosphorylation/activation, as well as further activation of multiple protein kinases including ERK_1/2_ [87]. G_βγ_ is also able to activate ERK_1/2_ through PLC-dependent or PI3K-dependent pathways [88]. The CB_1_ receptor-induced activation of PLC and the mobilization of Ca^2+^ from internal stores is rather complex, and depends on which cell line is being investigated. Indeed, in NG108-15 cells, this may involve G_i/o_ proteins which regulate the β_2_ isoform of PLC via G_βγ_ [89]; in HEK cells, this occurs via G_q_ proteins [90]; and in insulinoma cells, this occurs through G_q_/PLC [91]. We have already demonstrated that PLC contributes to vomiting since the PLC activator m-3M3FBS induces emesis in least shrews. In addition, the PLC inhibitor U73122 not only suppressed m-3M3FBS-induced vomiting, but also decreased vomiting induced by selective agonists of diverse emetic receptors as well as Ca^2+^ channel modulators [43]. In the present study, U73122 also suppressed SR141716A-induced vomiting, suggesting the involvement of PLC in CB_1_ receptor-mediated vomiting.

PKC is a family of protein kinases involved in regulating the function of other proteins through the phosphorylation of serine and threonine amino acid residues on these proteins. PKC proteins, in turn, are activated by signals such as an increase in the concentration of diacyl glycerol (DAG) or Ca^2+^ [92], which ultimately starts PKC phosphorylation/activation. In shrew brainstems, PKC phosphorylation is associated with emesis, and the PKC inhibitor GF109203X can suppress vomiting caused by several emetogens [42,43,47,52,66]. In the current study, phosphorylation of PKCαβII (Thr638/641) rapidly and significantly peaked at 15 min post injection of SR141716A, and the PKCα/βII inhibitor GF109203X completely suppressed SR141716A-induced vomiting, suggesting the involvement of PKCα/βII signaling in SR141716A-induced vomiting in shrews. Overall, both PLC and PKCαβII proteins are involved in SR141716A-induced emesis through PKCαβII and possibly PLC phosphorylation/activation.

#### 3.2.6. SR141716A Evokes Emesis Partly via Ca^2+^/Calmodulin (CaM)-Dependent Protein Kinase IIα (Ca^2+^/CaMKIIα)

Ca^2+^/CaMKIIα is a serine/threonine-specific protein kinase that is regulated by the Ca^2+^/calmodulin complex, which integrates transient, localized changes in intracellular Ca^2+^ levels [58,93]. CaMKII autophosphorylates in response to elevated intracellular Ca^2+^ and functions as an intracellular signaling protein [58]. We have already reported on the contribution of Ca^2+^/CaMKIIα signaling following 2-Me-5-HT-induced emesis in least shrews [45]. Pre-treatment with the CaMKII inhibitor KN62 suppressed both the evoked vomiting and phosphorylation of CaMKIIα and ERK in the brainstem caused by 2-Me-5-HT [46]. In the present study, KN62 also significantly suppressed the frequency of SR141716A-induced emesis up to 90% at doses greater than 2.5 mg/kg, and the percentage of shrews vomiting was reduced by 50–60% at doses greater than 1 mg/kg. 

### 3.3. SR141716A-Induced Vomiting Involves Ca^2+^-Ion Channel Modulators

Ca^2+^-induced Ca^2+^ release is a specific process that demonstrates how extracellular Ca^2+^ influx via the activation of VGCCs can mobilize intracellular Ca^2+^ release from sarcoplasmic/endoplasmic reticulum (SER) stores, resulting in an increase in the magnitude of the intracellular Ca^2+^ signal [94,95]. Intracellular Ca^2+^ acts as a second messenger for events such as Ca^2+^ influx [96], neurotransmitter release [97], and protein phosphorylation [98]. We have already demonstrated how Ca^2+^ mobilization is linked to vomiting [52,99]. Furthermore, the activation of CB_1_ receptors can result in the inhibition of Ca^2+^-dependent neurotransmitter release from presynaptic nerve terminals. This then leads to inhibition of neurotransmission [100], and further results in a reduction in postsynaptic neuronal activation, ultimately suppressing vomiting [99]. Dose-dependent inhibition of the CB_1_ receptor agonist WIN55212-2 on extracellular Ca^2+^ influx via the VGCCs located on the cell membrane has been shown in multiple prior experiments [25,101,102,103,104].

LTCCs can be activated by membrane depolarization, and subsequently act as the principal route of Ca^2+^ entry in electrically excitable cells such as neurons and muscle [105,106]. A functional link between LTCCs and RyRs also plays a major role in RyRs-mediated Ca^2+^ release from the SER into the cytoplasm following extracellular Ca^2+^ entry through LTCCs involving SOCE [96,107,108]. CB_1_ receptors and their endogenous (e.g., anandamide) and exogenous (e.g., WIN55212-2) ligands inhibit peak LTCC currents in a pertussis toxin-sensitive manner [109]. Furthermore, Hoddah and co-workers [34] showed that the cannabinoid CB_1/2_ receptor agonist WIN 55212-2 partially suppresses L-type Ca^2+^ currents in a voltage-independent manner in GT1-7 neurons via the cAMP/PKA pathway, which was reversed by both the L-type calcium blocker nifedipine and the CB_1_ receptor inverse agonist/antagonist AM-251. Moreover, AM-251 by itself caused a net upregulation of Ca^2+^ currents in these neurons. Likewise, WIN 55212-2 suppresses L-type Ca^2+^ channels via CB_1_ receptors through Gαi-proteins involving the cAMP/PKA pathway in neonatal rat NTS [110]. We have shown that the selective LTCC agonist FPL64176 causes vomiting in least shrews in a dose-dependent manner via both central and peripheral emetic sites [52,111]. Furthermore, LTCCs blockers, namely nifedipine (short-acting) and amlodipine (long-acting), suppress vomiting in a potent and dose-dependent fashion when caused by diverse emetogens [111]. Likewise, in the present study, nifedipine and amlodipine suppressed SR141716A-induced vomiting in a potent and dose-dependent manner, with near complete protection at doses of 1–5 mg/kg, suggesting that the process of Ca^2+^-induced Ca^2+^ release through LTCCs plays a critical role in CB_1_ receptor-mediated vomiting in least shrews. Furthermore, CB_1_ receptor ligands such as WIN55212-2 and AEA evoke concentration-dependent inhibition of peak L-type Ca^2+^ currents, which was abolished by SR141716A [109]. SR141716A appears to evoke vomiting either through (i) the disinhibition of CB_1_ receptors with a subsequent increase in L-type Ca^2+^ influx, which could be blocked by LTCC blockers such as nifedipine or amlodipine, or (ii) via an inverse agonist action by increasing calcium influx through the cell membrane [32,33].

Another route for Ca^2+^ entry through the cell membrane is via SOCE. Pharmacological emptying of SER Ca^2+^ pools can trigger extracellular Ca^2+^ influx via the activation of SOCE. This is mediated by the interaction between the ER Ca^2+^-sensors, stromal interacting molecule 1 and 2 (Stim1 and Stim2), and the Ca^2+^-permeable channels (Orai 1 and Orai 2) [112,113,114]. The activation of CB_1_ receptors can alter cytosolic Ca^2+^ signaling by triggering intracellular Ca^2+^ release from the ER and the intracellular lysosomal calcium store complex; however, the process depends upon the cannabinoid agonist employed and the cell type being tested [115,116]. The role of SOCE in vomiting has also been suggested; following pre-treatment with a potent and selective SOCE inhibitor, MRS1845, which significantly reduced the frequency of vomiting induced by either the HCN blocker ZD7288 [64] or the neurokinin NK_1_ receptor agonist GR73632 [42]. In the present study, the SOCE inhibitors MRS1845 and YM58483 potently and dose-dependently suppressed SR141716A-induced vomiting, with a complete blockade of SR141716A-induced vomiting at 2.5 mg/kg. These results provide evidence for SOCE playing a major role in CB_1_ receptor-mediated SR141716A-induced vomiting in least shrews. As was discussed earlier, LTCCs may be also involved in SOCE; investigation into their combined inhibitors could reveal significant antiemetic potential against SR141716A-evoked emesis.

TRPV1R plays a crucial role in maintaining intracellular Ca^2+^ homeostasis. TRPV1R is a nonselective cation channel and is expressed on the neuronal membrane and in the membranes of intracellular organelles (e.g., mitochondria, ER) [117,118,119,120]. We have previously shown that the selective and ultra-potent TRPV1R agonist RTX has pro- and antiemetic effects in shrews [48,121]. A subcutaneous injection of RTX by itself induces vomiting in the least shrew at doses higher than 10 μg/kg. However, its lower (0.01–5.0 μg/kg, s.c.) doses can suppress vomiting evoked by diverse emetogens [48,64]. Here, we demonstrate that RTX potently and completely (2.5 μg/kg) abolished vomiting caused by SR141716A (20 mg/kg, i.p.). Such potent and broad-spectrum antiemetic potential of ultra-low doses of RTX has also been reported in other vomit-competent animals [122,123,124,125]. This potent functional interaction could be due to the strong colocalization of CB_1_ and TRVP1 receptors [126]. The LTCC agonist nifedipine can prevent the ability of another TRPV1 agonist, capsaicin, to induce neuronal death with apoptotic features, which exemplifies interactions among different calcium channels [127]. Cannabis hyperemesis syndrome (CHS) occurs following chronic administration of large doses of CB_1_ agonists, further illustrating the complexity of endocannabinoid signaling. Interestingly, TRPV1R agonists such as capsaicin provide relief in CHS patients, supporting the idea that TRPV1R-CB_1_ crosstalk and compensatory adaptations of intracellular pathways are critical determinants of the emetic phenotype [128,129]. In fact, we have demonstrated that a combination of CB_1_- and TRPV1 receptor agonists have the capacity to completely abolish cisplatin-induced emesis at doses that are ineffective when used individually [121].

Intracellular Ca^2+^ release from SER stores into the cytoplasm is accomplished by IP3Rs and RyRs which are present on the wall of the SER. We have already demonstrated the important role of intracellular Ca^2+^ release via IP3Rs and RyRs in vomiting which can be differentially modulated by diverse emetogens, e.g., shrew vomiting evoked by either 2-methyl-5HT or FPL64176 was sensitive to the IP3R blocker dantrolene, but not the RyR blocker 2-APB [52]. On the other hand, pre-treatment with 2-APB caused a significant reduction in GR73632-induced vomiting, whereas dantrolene had no effect [42]. The SERCA pump is an important mechanism that transports free cytosolic Ca^2+^ into the lumen of SER to fill its internal Ca^2+^ stores [130,131]. The emetogen thapsigargin is a selective inhibitor of the SERCA pump which also releases intracellular Ca^2+^, and consequently depletes luminal SER Ca^2+^-stores, leading to a rise in the free concentration of cytoplasmic Ca^2+^ [132,133]. This increase in cytosolic calcium concentrations subsequently evokes vomiting in least shrews [53]. Pre-treatment with either dantrolene or 2-APB significantly reduced the frequency of thapsigargin-induced vomiting [53]. Likewise, in the current study, both 2-APB and dantrolene suppressed SR141716A-induced vomiting in a potent and dose-dependent manner, with complete suppression at 5 mg/kg. Although not yet tested, it would be interesting to investigate whether 2-APB or dantrolene could prevent the ability of a large concentration of SR141716A (5 µM) to induce the release of intracellular calcium [134]. The role of intracellular calcium can be further highlighted in the effects of SR141716A, since chelation of intracellular calcium prevents its stimulatory action on protein synthesis [35].

The current study provides in vivo evidence that SR141716A-induced vomiting depends upon both extracellular Ca^2+^ entry via plasma membrane Ca^2+^ channels (such as LTCC, SOCE, TRPV1R), as well as intracellular Ca^2+^ store release through IP3Rs and RyRs. Interestingly, SR 141716A has been shown to increase Ca^2+^ currents in male rat ganglion neurons [135]; moreover, Ca^2+^ green dextran transported from the nodose ganglion showed vagal afferent fibers ramifying throughout the NTS in brainstem slices [136]. These findings further support a potential association between SR141716A and Ca^2+^ dynamics in neurons, as well as the subsequent effects of Ca^2+^ on neuronal activity. Thus, future in-depth analysis of Ca^2+^ fluxes and related neurotransmitters during SR141716A-induced vomiting in shrews will greatly expand our knowledge in this area.

### 3.4. Effects of Diverse Classical Emetic Receptor Antagonists on SR141716A-Induced Vomiting

It is well established that cytotoxic chemotherapeutic drugs used in the treatment of cancer, including cisplatin, produce vomiting via indirect stimulation of 5-HT_3_, NK_1_, and D_2/3_ receptors (etc.), after the release of corresponding neurotransmitters in both the brainstem DVC and gastrointestinal tract emetic loci [1,137]. We have further shown that the neurokinin NK_1_ receptor antagonist netupitant dose-dependently suppresses SR141716A (20 mg/kg, i.p.)-induced vomiting in least shrews [20]. In the present study, we investigated the possibility of involvement of 5-HT_3_ and D_2/3_ receptors in vomiting evoked by SR141716A. Pre-treatment with the selective 5-HT_3_ receptor antagonist palonosetron reduced both the mean frequency and number of shrews vomiting in response to SR141716A in a U-shaped dose–response curve. Palonosetron also decreases the frequency of cisplatin-induced vomiting in least shrews in a U-shaped dose–response manner [111]. Currently, we further found that the D_2/3_ receptors antagonist sulpiride also dose-dependently inhibits SR141716A (20 mg/kg, i.p.)-induced vomiting, indicating that D_2/3_ receptors are also involved in SR141716A (20 mg/kg, i.p.)-induced vomiting.

The classical antagonist antiemetic studies discussed above suggest that SR141716A may cause the release of 5-HT, DA, and/or SP in the brainstem emetic loci of shrews to induce vomiting. In fact, 30 min exposure to a fully effective emetic dose of SR141716A (20 mg/kg) has previously been shown to significantly increase the release and turnover of serotonin and DA in the least shrew forebrain, but not to a significant degree in their brainstem [15]. Likewise, the regional effects of SR141716A on monoamine release have been noted in some, but not all, regions of the brain by other researchers [138,139,140,141]. SR141716A was also found to enhance the capsaicin-induced release of SP in sections of the spinal cord in mice [142]. As expected, according to our immunohistochemistry data in the present study, SR141716A also failed to increase 5-HT or SP immunoreactivities in either the AP, NTS, or DMNX nuclei of the brainstem DVC in least shrews at 15 min post injection. These findings seem to be incompatible with other emetogens such as cisplatin, yohimbine, FPL64176, or thapsigargin, whose vomiting was accompanied by significant increases in immunoreactivities of 5-HT or SP, or both, in one or more regions of the DVC emetic nuclei in least shrews [21,52,53,137]. These findings may reflect the following: (i) regional differences in the ability of SR141716A to induce changes in neurotransmitter levels in different regions of the brain, (ii) not all the cited emetogens concurrently increase both 5-HT and SP levels in the DVC, (iii) the evoked release of emetic neurotransmitter is rapidly metabolized [15], (iv) the release dynamics, or receptor-level sensitivity, may also play a more critical role than static tissue levels, or (v) the initiating site of the emetic action of SR141716A could be somewhere else in the brain or gut, which needs further investigation.

### 3.5. SR141716A Produces its Emetic Signaling Effects via Inverse Agonism

CB_1_ receptor ligands can act as either agonists, antagonists, or inverse agonists. Similar to an agonist, an inverse agonist also binds to the orthosteric site of the receptor but produces an opposite effect to that of agonists. In addition, evidence is accumulating for the presence of constitutively active G-protein-coupled receptors (GPCRs) with increased basal activity, including CB_1_ receptors in some regions of rodent brains, that are spontaneously active in the absence of a cannabinoid agonist. The inverse activity of a ligand may reduce this basal activity [62]. Moreover, GPCRs, including CB_1_ receptors, can adopt multiple conformations states, leading to different signaling events. The different conformations could be stabilized by diverse ligands, evoking ligand-biased specific signal transduction. Thus, the agonist, inverse agonist, or antagonist properties of a ligand can only be interpreted when a particular signal transduction pathway is defined, since the same inverse ligand can behave as an agonist in one intracellular signaling pathway, or as an antagonist, or even inverse agonist, in other signaling pathways for a given receptor [143]. In the case of the CB_1_ receptor, examples of similarly biased agonisms, as well as inverse agonist activity, have been observed when the activation of different Gi/o subtypes was measured [26]. For example, the CB_1/2_ receptor agonist WIN 55212-2 partially suppresses L-type Ca^2+^ currents in a voltage-independent manner in GT1-7 neurons via the cAMP/PKA pathway. This inhibitory effect was not only reversed by low concentrations of the CB_1_ receptor inverse agonist/antagonist AM-251; but at high concentrations, AM-251, in the absence of an agonist, caused a net increase in L-type Ca^2+^ currents in these neurons, which demonstrates AM-251’s inverse activity [34]. Likewise, high concentrations of two other CB_1_-receptor inverse agonists/antagonists, namely SR141716A and MJ08, significantly increase cAMP levels by themselves in CHO-hCB1 cells [28], or in different regions of the brain in rats, as well as in the human frontal cortex [29]. In the current study, SR141716A- induced vomiting and signaling changes appear to be inverse-agonist-related events.

## 4. Materials and Methods

### 4.1. Animals

We used adult least shrews from the Western University of Health Sciences animal breeding facilities. The animals were housed in groups of 5–10 on a 14:10 light/dark cycle, fed and watered at ad libitum. The experimental shrews were 45–60 days old and weighed 4–6 g. The present animal handling protocols were based upon our previous emesis studies [14,45]. All protocols were approved by the Institutional Animal Care and Use Committee of Western University of Health Sciences (protocol number R23IACUC013) and were in accordance with the principles and procedures of the National Institutes of Health Guide for the Care and Use of Laboratory Animals. All efforts were made to minimize animals’ suffering and to reduce the number of animals used in the experiments.

### 4.2. Chemicals

The following drugs were used for the present studies: SR141716A, H-89, U0126, PD98059, AR-A014418, SB216763, LY294002, U73122, GF109203X, KN62, amlodipine, YM-58483, MRS-1845, and RTX were purchased from Tocris (Minneapolis, MN, USA); nifedipine and sulpiride were purchased from Sigma-Aldrich (St. Louis, MO, USA); dantrolene and 2-APB were purchased from Santa Cruz Biotechnology (Dallas, TX, USA); and palonosetron was kindly provided by Helsinn Health Care (Lugano, Switzerland).

Amlodipine, LY294002, dantrolene, 2-APB, GF109203X, AR-A014418, KN62, YM-58483, MRS-1845, and PD98059 were dissolved in 25% DMSO in water. H-89 and SB216763 were dissolved in 2.5% DMSO in water. SR141716A, U73122, nifedipine, U0126, and RTX were reconstituted with twice the stated drug dose in a 1:1:18 solution of ethanol: Emulphor™:0.9% saline so that administration of large amounts of DMSO to the shrews could be avoided. Palonosetron was dissolved in distilled water. Sulpiride was dissolved in distilled water with a 10 μL volume of 1/3 concentrated HCl, which was then back-titrated to pH 5 by the addition of NaOH. All drugs were administered at a volume of 0.1 mL/10 g of body weight.

### 4.3. Behavioral Emesis Studies

On the day of the experiment, shrews were brought into the experimental room from the animal facility, weighed, transferred to 20 × 18 × 21 cm clean clear plastic individual cages, and allowed to acclimate for 2 h, during which time daily food was withdrawn. Drug-naïve male and female shrews were randomly allocated to the control and the experimental groups regardless of their cage of origin. The shrews were given four meal worms (*Tenebrio* sp.) each 30 min before the administration of emetogen, to assist in the identification of wet vomits, as explained previously [14,45]. Administration of SR141716A at 20 mg/kg (i.p.) was previously shown to evoke vomiting with a maximal frequency in 100% of tested shrews [14]. Thus, this dose was used for subsequent biochemical and antagonist studies. All experiments were performed between 8:00 a.m. and 5:00 p.m.

In the drug interaction studies, different groups of shrews were pre-treated at 0 min with an injection of either corresponding vehicle [(intraperitoneal (i.p.) or subcutaneous (s.c.)] or varying doses of the following: (1) selective inhibitors of the signaling pathways, e.g., (i): ERK_1/2_ [U0126 (0, 1, 5, and 10 mg/kg, i.p.) or PD98059 (0, 1, 2.5, and 5 mg/kg, i.p.)] [40]; (ii) GSK-3αβ [AR-A014418 (0, 1, 2.5, 5, and 10 mg/kg, i.p.) or SB216763 (0, 0.25, 1, and 2.5 mg/kg, i.p.)] [41]; (iii) PI3K [LY294002 (0, 1, 2.5, and 10 mg/kg, i.p.)] [42]; (iv) PLC [U73122 (0, 1, 2.5, 5, and 10 mg/kg, i.p.)] [43]; (v) PKCαβII [GF109203X (0, 0.1, 1, 5, and 10 mg/kg, i.p.)] [43]; (vi) PKA [H89 (0, 0.1, 1, 2 and 5 mg/kg, i.p.)] [144]; (vii) CaMKII [KN62 (0, 1, 2.5, 5, and 10 mg/kg, i.p.)] [45]; (2) selective inhibitors of cell membrane Ca^2+^ channels, e.g., (i) LTCC [nifedipine (0, 0.1, 0.25, 0.5, 1, and 5 mg/kg, s.c.) and amlodipine (0, 0.25, 0.5, 1, 5, and 10 mg/kg, i.p.)] [46]; (ii) SOCE [YM-58483 (0, 0.1, 0.5, 1, and 2.5 mg/kg, i.p.) and MRS-1845 (0, 0.01, 0.1, 0.5, 1, and 2.5 mg/kg, i.p.)] [42]; (iii) TRPV1R agonist [RTX (0, 0.25, 1, and 2.5 mg/kg, s.c.)] [48]; (3) antagonists of selective intracellular Ca^2+^ channel modulators, e.g., (i) IP3R [2-APB (0, 0.5, 1, and 5 mg/kg, i.p.)] [45]; (ii) RyRs [dantrolene (0, 1, 2.5, and 5 mg/kg, i.p.)] [45]; (4) selective receptor antagonists, e.g., (i) 5-HT_3_ receptor [palonosetron (0, 0.25, 1, 2.5, and 5 mg/kg, s.c.)] [45]; (ii) D_2/3_ receptor [sulpiride (0, 1, 2, and 4 mg/kg, s.c.)] [47]. After 30 min, vomiting was induced in the pre-treated shrews with a 20 mg/kg dose of SR141716A (i.p.). Each shrew was then placed in the observation cage, and the frequency of vomiting was recorded over the next 30 min. In the emesis studies, the observer was unaware of the treatment conditions. Each shrew was used once and then euthanized with isoflurane (3%) in the anesthesia chamber following the termination of each experiment.

### 4.4. Immunohistochemistry and Image Analysis

#### 4.4.1. Immunohistochemical Distribution of CB_1_ Receptors in the Least Shrew DVC

To examine the distribution and changes in expression of the CB_1_ receptors evoked by SR141716A (20 mg/kg, i.p.) in shrews, (AP/DMNX/NTS), vehicle- and SR141716A (20 mg/kg, i.p.)-treated shrews (*n* = 5 each group) were deeply anesthetized with isoflurane (3%) at 30 min post injection, and then transcardially perfused with 0.01 M phosphate-buffered saline (PBS) followed by ice-cold 4% paraformaldehyde (PFA, pH 7.4) in 0.01 M PBS for 10 min. Brainstems were removed and post-fixed in the same fixative for 2 h, and then placed in 0.1 M PB containing 30% sucrose at 4 °C until they sank. The brainstems were cut into 20 μm sections using a cryostat (Leica, Bannockburn, IL, USA), and pre-incubated in the blocking buffer (0.01 M PBS containing 10% normal donkey serum and 0.3% Triton X-100) for 1 h at room temperature. The brainstem slices were then incubated in a rabbit anti-CB_1_ receptor primary antibody (1:1000, ab3558, Abcam, Cambridge, UK) in blocking buffer at 4 °C overnight. After washing for 3 × 10 min in PBS, sections were incubated in an Alexa Fluor 594 donkey anti-rabbit secondary antibody (1:500, A-21207, Invitrogen, Carlsbad, CA, USA) in 0.01 M PBS containing 0.3% Triton X-100 for 2 h at room temperature. The sections were then washed 3 times (10 min each) and were mounted and cover-slipped with an anti-fade mounting medium containing DAPI (Vector Laboratories, Newark, CA, USA). Host-matched rabbit IgG (1:1000, #3900, Cell Signaling, Danvers, MA, USA) isotype control (substituting for the primary CB_1_ antibody) and a secondary-only control (omitting the primary antibody) were processed in parallel under identical conditions (Appendix A). Tile-scanning images were taken by a confocal microscope (Zeiss LMS 880, Jena, Germany) at 1024 × 1024 pixels with Zen software (Zen 3.5, Jena, Germany) using Plan-Apochromat 63×/1.4 oil DICM27 objective. Cytoarchitectonic differences in the AP, NTS, and DMNV of the brainstems of least shrews have been described in our published studies [41,42,52,53,60,145,146]. This imaging acquisition set-up was applied to the immunohistochemistry methodology performed in this study. For each animal, the mean gray value of the CB_1_ receptors in the AP along with both sides of the NTS and DMNX from 3 sections at 90 μm intervals were calculated via ImageJ (ImageJ 1.54p, Bethesda, MD, USA) by an experimenter blind to experimental conditions. The average value was used in statistical analysis.

#### 4.4.2. c-*fos* Staining and Image Analysis

Since SR141716A at 20 mg/kg (i.p.) evoked vomiting with a maximal frequency in 100% of tested shrews [14], we conducted immunohistochemistry following intraperitoneal administration of this dose of SR141716A [41,42,52,53,60,145,146].

Immunohistochemistry of the least shrew brainstem (20 µm) and jejunal (25 μm) sections was performed as previously reported [43,64]. The jejunal segment of the least shrew small intestine was dissected as described previously [63]. Following vehicle or SR141716A (20 mg/kg, i.p.) injection, vomiting shrews were subjected to c-*fos* staining. At 90 min after the first emesis occurred, shrews were deeply anesthetized and transcardially perfused as described above. Brainstems and jejunum were removed and post-fixed in the same fixative for 2 h, and then placed in 0.1 M PB containing 30% sucrose at 4 °C until they sank. The brainstems and jejunum were cut into 20 μm and 25 μm sections, respectively, as described above, and pre-incubated in the blocking buffer for 1 h at room temperature. The brainstem slices were then incubated in a rabbit anti-c-*fos* primary antibody (1:1000, ab190289, Abcam, Cambridge, UK) in blocking buffer at 4 °C overnight. After washing 3 times (10 min each) in PBS, sections were incubated in an Alexa Fluor 594 donkey anti-rabbit secondary antibody (1:500, A-21207, Invitrogen, Carlsbad, CA, USA) in 0.01 M PBS containing 0.3% Triton X-100 for 2 h at room temperature. To confirm neuronal localization of c-*fos* in the ENS of the jejunum, the jejunal sections were incubated in the same condition, using rabbit anti-c-*fos* primary antibody (1:1000, ab190289, Abcam, Cambridge, UK) and mouse anti-NeuN (neuronal marker) antibody (1:300, MAB 377, Millipore, Burlington, MA, USA) as the primary antibodies. Alexa Fluor 594 donkey anti-rabbit IgG (1:500, A-21207, Invitrogen, Carlsbad, CA, USA) and Alexa Fluor 488 donkey anti-mouse antibody (1:500, A-21202, Invitrogen, Carlsbad, CA, USA) were used as secondary antibodies. Tile-scanning images were taken by a confocal microscope (Zeiss LMS 880, Jena, Germany) at 1024 × 1024 pixels using Zen software with a Plan-Apochromat 20×/0.8 M27 objective. For each animal, the number of c-*fos*-positive cells in the AP, both sides of NTS and DMNX, and the amount of co-staining of c-*fos* and NeuN in 3 jejunal sections from 90 μm intervals were counted manually by an experimenter blind to experimental conditions. The average value was used in statistical analysis.

#### 4.4.3. Phospho-ERK_1/2_ Immunohistochemistry

To analyze phospho-ERK_1/2_ expression evoked at 15 min after SR141716A administration, shrews were deeply anesthetized and transcardially perfused at 15 min after administration of vehicle or SR141716A (20 mg/kg, i.p.). Phospho-ERK_1/2_ immunofluorescence staining was conducted on brainstem and jejunal sections from vehicle- and SR141716A (20 mg/kg, i.p.)-treated shrews (*n* = 4 shrews per group). The sections were incubated under the same conditions as for c-*fos* immunohistochemistry with a rabbit anti-phospho-ERK_1/2_ (Thr202/Thr204) (1:500, 4370, Cell Signaling, Danvers, MA, USA) primary antibody, followed by an Alexa Fluor 594 donkey anti-rabbit (1:500, A-21207, Invitrogen, Carlsbad, CA, USA) secondary antibody incubation. The nuclei of the cells were stained with DAPI. The mean gray value of phospho-ERK_1/2_ immunoreactivity in the brainstem DVC and three jejunal sections at 90 μm intervals from each animal of both treatment groups was determined via ImageJ (ImageJ 1.54p, Bethesda, MD, USA), and the mean value per section from individual shrews was used for statistical analysis [64].

#### 4.4.4. Phospho-GSK-3α/β Immunohistochemistry

To analyze phospho-GSK-3α/β expression evoked at 15 min after SR141716A administration, different groups of shrews (*n* = 6 shrews per group) were perfused at 15 min after the administration of vehicle or SR141716A (20 mg/kg, i.p.). Phospho-GSK-3αβ immunofluorescence staining was conducted on brainstem sections with rabbit anti-phospho-GSK-3α (PSER 21) (1:500, SAB4504419, Sigma-Aldrich, St. Louis, MO, USA) and mouse anti-GSK-3β (1:500, sc-373800, Santa Cruz Biotechnology, Dallas, TX, USA) primary antibodies, followed by Alexa Fluor 594 donkey anti-rabbit (1:500, A-21207, Invitrogen, Carlsbad, CA, USA) and Alexa Fluor 488 donkey anti-mouse (1:500, A-21202, Invitrogen, Carlsbad, CA, USA) secondary antibody incubation. The mean gray values of phospho-GSK-3α/β immunoreactivity in the brainstem DVC were analyzed as described above.

#### 4.4.5. 5-HT and SP Immunohistochemistry

Shrews (*n* = 4 per group) were treated with either the vehicle or SR141716A (20 mg/kg, i.p.); they were then rapidly anesthetized with isoflurane and subjected to perfusion at 15 min post treatment to examine 5-HT and SP immunoreactivity. Brainstem sections were incubated with a mix of goat anti-5-HT antibody (1:1000, ab66047, Abcam, Cambridge, UK) and rat anti-SP primary antibody (1:400, MAB356, EMD Millipore, Billerica, MA, USA), followed by a mix of Alexa Fluor 488 donkey anti-goat antibody (1:500, ab150133, Abcam, Cambridge, UK) and cy3-conjugated donkey anti-rat (1:500, AP189C, EMD Millipore, Billerica, MA, USA) secondary antibody [21]. Fluorescence intensity (mean gray value) of 5-HT and SP values were acquired using ImageJ software as described above.

### 4.5. Cyclic AMP Measurement

Brainstems were collected at indicated time points (0, 15, and 30 min) after drug treatment and immediately frozen at −80 °C before use. Cyclic AMP levels in the brainstem tissue were measured using the DetectXs Direct Cyclic AMP (cAMP) Immunoassay kit (Arbor Assays, Ann Arbor, MI, USA). Samples were homogenized in the sample diluent followed by incubation on ice for 10 min before being centrifuged at 13,000× *g* at 4 °C for 10 min. The supernatants were collected and run in the assay immediately according to the instructions from the manufacturer (Arbor Assays, Ann Arbor, MI, USA) [55].

### 4.6. Western Blot

The time-dependent profile of PKAα/β (Thr197), Akt (Ser473), GSK-3α/β (Ser21/9), ERK_1/2_ (Thr202/204), and PKCα/βII (Thr638/641) phosphorylation was determined using different groups of animals (*n* = 4–6 per group), euthanized at 15 and 30 min following the administration of fully effective emetic dose of SR141716A (20 mg/kg, i.p.). Shrew brainstems collected at each time point were homogenized in a lysis buffer (Thermofisher Scientific, Waltham, MA, USA). Protein extracts from brainstem lysates were subjected to Western blot analysis. Information regarding the primary antibodies is summarized in Table 1. The sections were counterstained with either goat anti-rabbit IRDye 680RD, goat anti-mouse IRDye 800CW (1:10000, LI-COR, Lincoln, NE, USA), or both as per the experimental conditions. Bound antibodies were visualized correspondingly using Odyssey imaging system (Lincoln, NE, USA). The ratios of the phosphorylated forms of PKAα/β (Thr197), Akt (Ser473), GSK-3α/β (Ser21/9), and ERK1/2 (Thr202/204) to their respective total protein forms were calculated. The ratio of phosphorylated forms of PKCα/β II (Thr638/641) was reported to GAPDH (see Table 1 for dilutions). All values were divided by the average value at the time point 0 for normalization and presented as the fold change of the controls [47].

### 4.7. Statistical Analysis

Statistical analyses were performed using Graphpad Prism 8 (Graphpad Software Inc., San Diego, CA, USA). The vomit frequency data were analyzed using the Kruskal–Wallis non-parametric one-way analysis of variance (ANOVA); this was followed by post hoc analysis using Dunn’s multiple comparisons test and expressed as the mean ± SEM. The percentage of animals vomiting across groups at different doses was compared using the Chi-square test. Following Western blot, the three groups (0, 15, and 30 min) were compared using a one-way ANOVA followed by Tukey’s post hoc test to determine statistical significance between the experimental groups and the corresponding control. Statistical significance for differences between two groups was tested via the unpaired *t*-test. *p* < 0.05 was considered statistically significant.

## 5. Conclusions

In summary, our findings demonstrate that CB_1_ receptor inverse agonist activity of SR141716A (20 mg/kg, i.p.) induces vomiting in least shrews via both central and peripheral mechanisms; these are accompanied by robust c-*fos* expression and p-GSK-3αβ in the shrew brainstem DVC, as well as p-ERK_1/2_ in the brainstem DVC and the ENS in the jejunum. The evoked vomiting also involved time-dependent increases in cAMP levels in the shrew brainstem, as well as the activation/phosphorylation of other emesis-associated proteins, including PKA, PI3K/Akt/GSK-3, and PLC/PKCαβII, and CaMKII signaling. Furthermore, SR141716A-induced vomiting also depends on Ca^2+^ mobilization, which involves extracellular Ca^2+^ entry via plasma membrane Ca^2+^ channels (such as LTCC, SOCE, TRIPV1R), as well as intracellular Ca^2+^ release via IP3Rs and RyRs. 5-HT_3_, NK_1_, and D_2/3_ receptors also contribute to SR141716A-mediated emesis, without affecting their corresponding endogenous 5-HT or SP brain tissue levels.

## Figures and Tables

**Figure 1 ijms-26-09884-f001:**
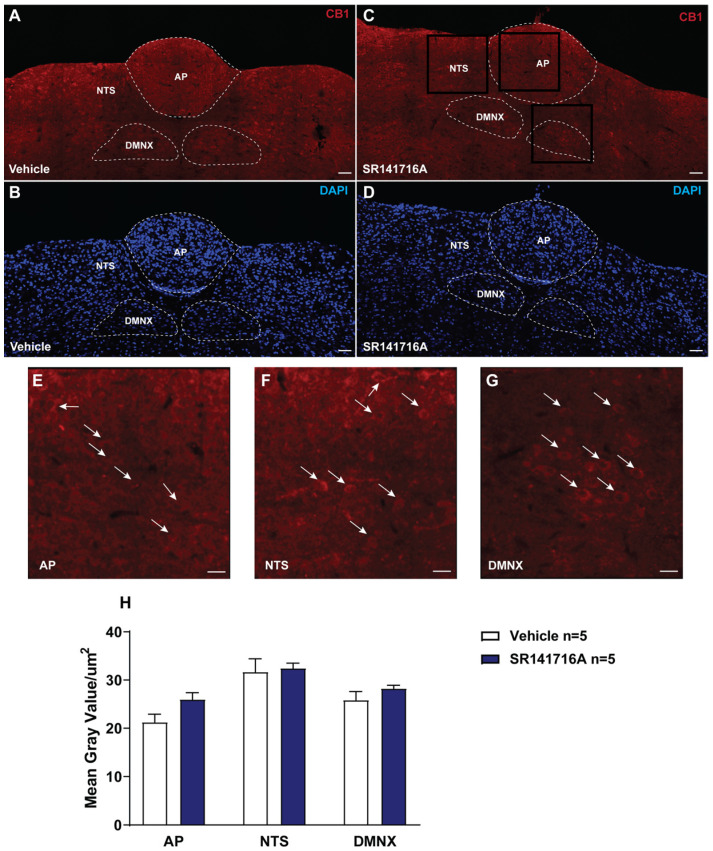
Immunoreactivity of CB_1_ receptors in the dorsal vagal complex (DVC) of the least shrew brainstem. Coronal sections (20 µm) through the DVC were processed for CB_1_ immunofluorescence 30 min after vehicle or SR141716A (20 mg/kg, i.p.). Dashed lines delineate the area postrema (AP), nucleus tractus solitarius (NTS), and dorsal motor nucleus of the vagus (DMNX)—the emetic nuclei analyzed. (**A**,**B**) vehicle: CB_1_ channel (**A**, red) and DAPI counterstain (**B**, blue). (**C**,**D**) SR141716A: CB_1_ channel (**C**, red; boxed regions indicate sites sampled at higher magnification) and DAPI (**D**, blue). CB_1_ immunoreactivity is most prominent in the dorsomedial NTS and DMNX, with lower signal in adjacent NTS subnuclei and AP; SR141716A did not alter this distribution relative to vehicle. (**E**–**G**) high-magnification insets from the boxed regions in panel C showing AP (**E**), NTS (**F**), and DMNX (**G**); arrows indicate representative CB_1_-positive neuronal profiles/neuropil. (**H**) quantification of CB_1_ signal (mean gray value per µm^2^) within predefined ROIs shows no significant difference between groups (vehicle *n* = 5, SR141716A *n* = 5; unpaired *t*-tests). All images were acquired with identical laser power/gain and processed in parallel. Scale bars: (**A**–**D**) 50 µm; (**E**–**G**) 20 µm.

**Figure 2 ijms-26-09884-f002:**
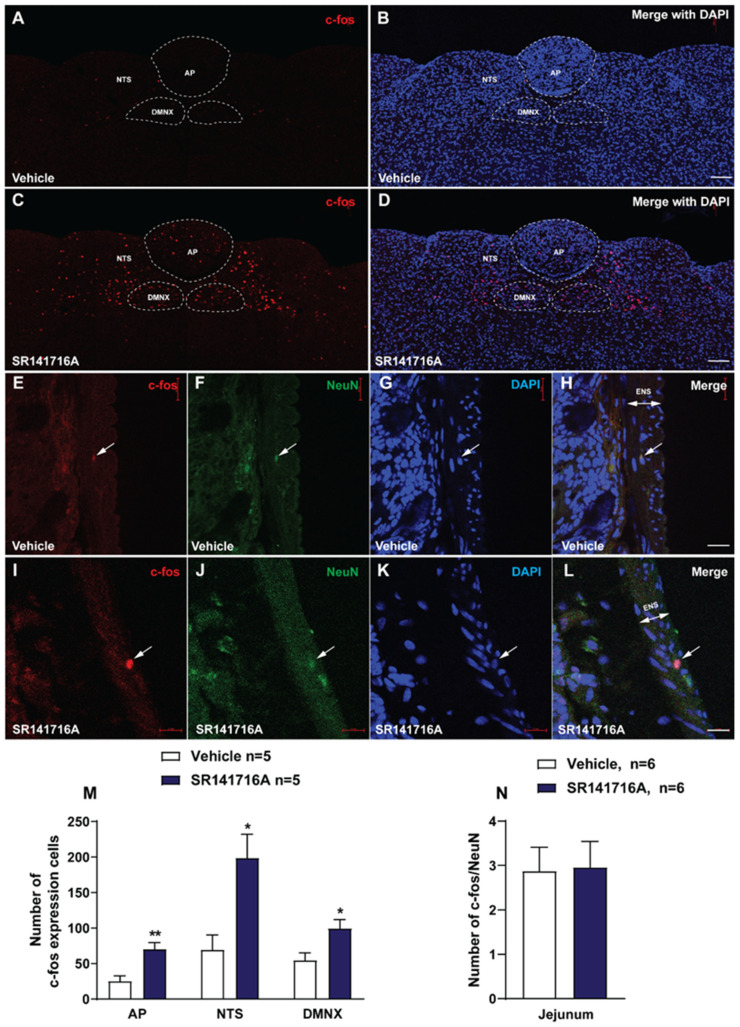
Immunohistochemical analysis of c-*fos* in the emetic nuclei of the brainstem DVC of least shrews, the area postrema (AP), the nucleus tractus solitarius (NTS) and the dorsal motor nucleus of the vagus (DMNX), and the enteric nervous system (ENS) embedded in the wall of the jejunum, following vomiting induced by systemic administration of the CB_1_ receptor inverse agonist/antagonist SR141716A (20 mg/kg, i.p.). Least shrews were sacrificed 90 min post vehicle treatment, or after the first vomiting occurred post administration of SR141716A. Shrew brainstem sections (20 μm) were stained with rabbit anti-c-*fos* primary antibody and Alexa Fluor 594 donkey anti-rabbit secondary antibody (**A**–**D**). The intestinal jejunum sections (25 μm) were stained with a mix of rabbit anti-c-*fos* and mouse anti-NeuN (neuronal marker) primary antibodies, followed by a mix of Alexa Fluor 594 donkey anti-rabbit and Alexa Fluor 488 donkey anti-mouse secondary antibodies (**E**–**L**). Nuclei were stained with DAPI in blue. Representative tile-scanned images (20×) show robust c-*fos* induction in the brainstem DVC (AP, NTS, DMNX) in response to SR141716A (20 mg/kg, i.p.; **C**,**D**) compared to the vehicle-treated group (**A**,**B**; *n* = 5 shrews per group). Scale bar = 100 μm. Quantified data show the SR141716A-induced c-*fos* expression in the AP, NTS, and DMNX in the brainstem of least shrews (**M**). Representative single-filed images (60×) show low c-*fos* expression in the enteric nervous system (ENS) of the intestinal jejunum in vehicle (**E**–**H**)- and SR141716A (20 mg/kg, i.p.; **I**–**L**)-treated shrews (*n* = 6 shrews per group). Scale bar = 20 μm. Dashed lines delineate the boundaries of the AP, NTS, and DMNX (**A**–**D**); arrows indicate representative immunopositive signals (**E**–**L**). Compared to the vehicle group, the administration of SR141716A did not induce a significant increase in c-*fos* expression in NeuN-positive neurons of the ENS within the intestinal jejunum of least shrews (**N**). Values represent the mean number of c-*fos*-positive nuclei of each region per section and are presented as mean ± SEM. * *p* < 0.05, ** *p* < 0.01 vs. vehicle, unpaired *t*-test.

**Figure 3 ijms-26-09884-f003:**
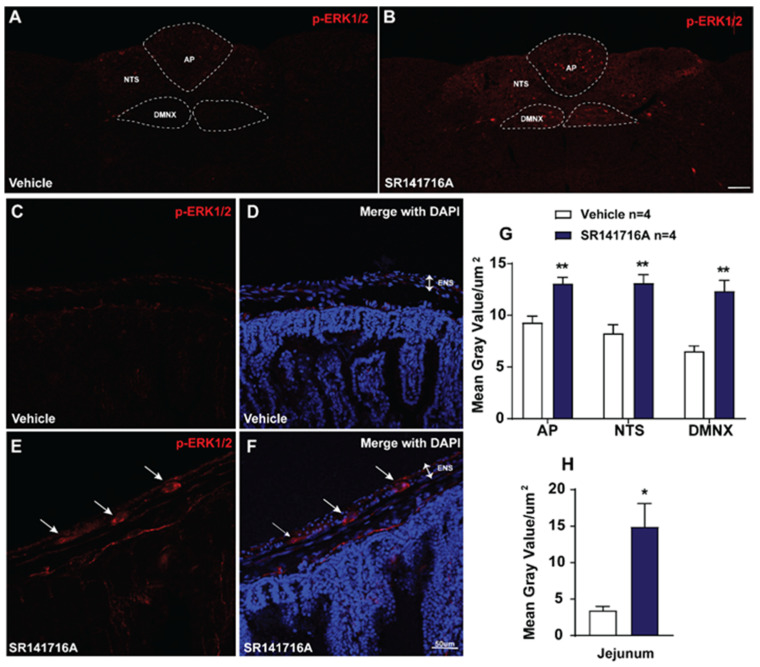
Immunohistochemical analysis of ERK_1/2_ phosphorylation following CB_1_ receptor inverse agonist/antagonist SR141716A (20 mg/kg, i.p.)-induced vomiting. Least shrews were sacrificed 15 min after vehicle or SR141716A injection (20 mg/kg, i.p.) (*n* = 4 shrews per group). Brainstem sections (20 μm) and intestinal jejunum sections (25 μm) were stained with rabbit anti-phospho-ERK_1/2_ antibody and Alexa Fluor 594 donkey anti-rabbit secondary antibody. Nuclei were stained with DAPI in blue. Representative tile scan images (20×) show a strong upregulation of ERK_1/2_ phosphorylation in the brainstem dorsal vagal complex (DVC) in response to SR141716A (**B**) compared with the vehicle-treated group (**A**). Quantified data shows the SR141716A-induced upregulation of ERK_1/2_ phosphorylation in the AP, NTS and DMNX in least shrew brainstem (**G**). Scale bar = 100 μm. Representative single-filed images (60×) also show ERK_1/2_ phosphorylation evoked by SR141716A compared with vehicle in the ENS in the least shrew intestinal jejunum (**C**–**F**). Scale bar = 50 μm. Dashed lines delineate the boundaries of the AP, NTS, and DMNX (**A**,**B**); arrows indicate representative immunopositive signals (**E**,**F**). Statistical analysis of mean gray value of ERK_1/2_ phosphorylation evoked by SR141716A (20 mg/kg, i.p.) in the ENS in least shrew intestinal jejunum (**H**). Values represent the mean gray value of evoked ERK_1/2_ phosphorylation of each region per section and are presented as mean ± SEM. * *p* < 0.05, ** *p* < 0.01 vs. vehicle, unpaired *t*-test.

**Figure 4 ijms-26-09884-f004:**
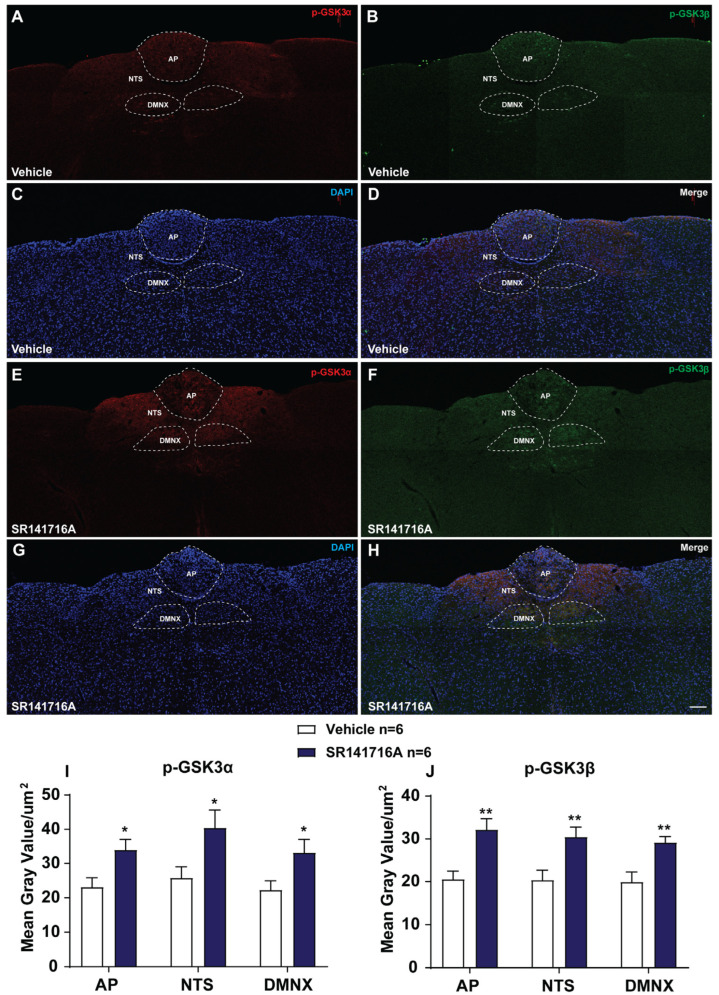
Immunohistochemical analysis of GSK-3α/β phosphorylation following CB_1_ receptor inverse agonist/antagonist SR141716A (20 mg/kg, i.p.)-induced vomiting. Least shrews were sacrificed 15 min after vehicle or SR141716A injection (20 mg/kg, i.p.) (*n* = 6 shrews per group). Brainstem sections (20 μm) were stained with rabbit anti-phospho-GSK-3α (PSER 21) and mouse anti-GSK-3β primary antibodies followed by Alexa Fluor 594 donkey anti-rabbit and Alexa Fluor 488 donkey anti-mouse secondary antibodies incubation. Nuclei were stained with DAPI in blue. Representative tile scan images (20×) show strong upregulation of GSK-3α and GSK-3β phosphorylation in the brainstem dorsal vagal complex (DVC) in response to SR141716A (**E**–**H**) compared with vehicle-treated group (**A**–**D**). Dashed lines delineate the boundaries of the AP, NTS, and DMNX (**A**–**H**). Quantified data shows the SR141716A-induced upregulation of GSK-3α and GSK-3β phosphorylation in the AP, NTS and DMNX in least shrew brainstem (**I**,**J**). Scale bar = 100 μm. Values represent the mean gray value of evoked GSK-3α/β phosphorylation of each region per section and are presented as mean ± SEM. * *p* < 0.05, ** *p* < 0.01 vs. vehicle, unpaired *t*-test.

**Figure 5 ijms-26-09884-f005:**
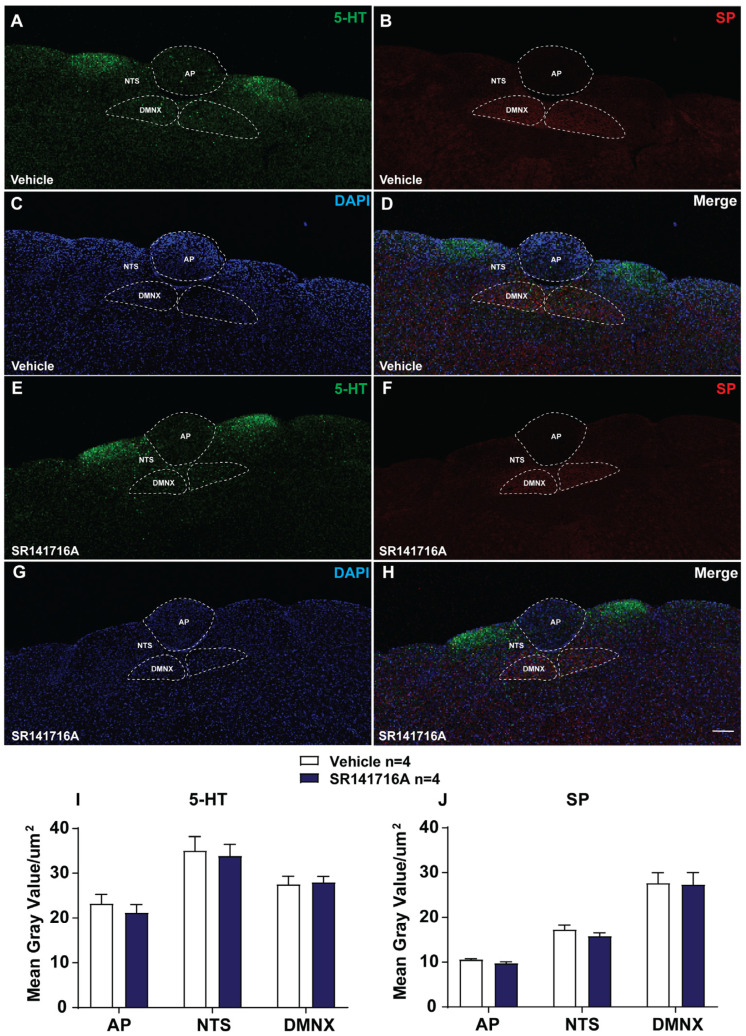
Immunohistochemical analysis of the release of 5-HT and SP following CB_1_ receptor inverse agonist/antagonist SR141716A (20 mg/kg, i.p.)-induced vomiting. Least shrews were sacrificed 15 min after vehicle or SR141716A injection (20 mg/kg, i.p.) (*n* = 4 shrews per group). Brainstem sections (20 μm) were stained with a mix of goat anti-5-HT and rat anti-SP primary antibodies followed by a mix of Alexa Fluor 488 donkey anti-goat and cy3-conjugated donkey anti-rat secondary antibodies. Nuclei were stained with DAPI in blue. Representative tile scan images (20×) show the release of 5-HT and SP in the brainstem dorsal vagal complex (DVC) in response to SR141716A (**E**–**H**) compared with the vehicle-treated group (**A**–**D**). Dashed lines delineate the boundaries of the AP, NTS, and DMNX (**A**–**H**). Quantified data show that SR141716A did not induce significant changes in the release of 5-HT and SP in the AP, NTS, and DMNX in least shrew brainstems compared to the vehicle-treated group (**I**,**J**). Scale bar = 100 μm. Values represent the mean gray value of the released 5-HT and SP of each region per section and are presented as mean ± SEM (unpaired *t*-test).

**Figure 6 ijms-26-09884-f006:**
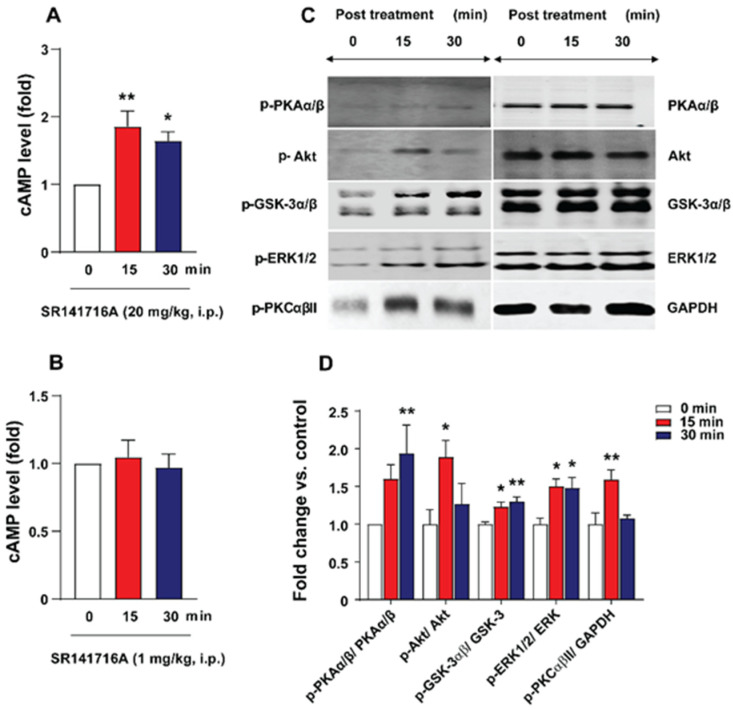
CB_1_ receptor inverse agonist/antagonist SR141716A (20 mg/kg, i.p.) evoked increases in cAMP levels and phosphorylation of PKAα/β (Thr197), Akt (Ser473), GSK-3α/β (Ser21/9), ERK_1/2_ (Thr202/204), and PKCαβII (Thr638/641) proteins in shrew brainstems. ELISA data show that administration of a non-emetic dose of SR141716A (1 mg/kg, i.p.) did not induce any changes in least shrew brainstem cAMP tissue levels (**B**), whereas the administration of a fully effective emetic dose of SR141716A (20 mg/kg, i.p.) resulted in significant and time-dependent increases in cAMP tissue levels in least shrew brainstems (**A**). Representative Western blots of PKAα/β (Thr197), Akt (Ser473), GSK-3α/β (Ser21/9), ERK_1/2_ (Thr202/204), and PKCαβII (Thr638/641) protein phosphorylation in the least shrew brainstems collected from control (0 min) or at the indicated time points after intraperitoneal (i.p.) administration of fully effective emetic dose of SR 141716A (20 mg/kg, i.p.) (**C**). Quantitative analysis of Western blots of phosphorylation of PKAα/β (Thr197), Akt (Ser473), GSK-3α/β (Ser21/9), ERK_1/2_ (Thr202/204), and PKCαβII (Thr638/641) proteins or GAPDH and ratios were obtained (**D**). All ratios were normalized to controls (0 min) and expressed as the fold change of the controls. * *p* < 0.05 and ** *p* < 0.01 vs. 0 min, one-way ANOVA followed by Dunnett’s post hoc test (*n* = 4–8 per group).

**Figure 7 ijms-26-09884-f007:**
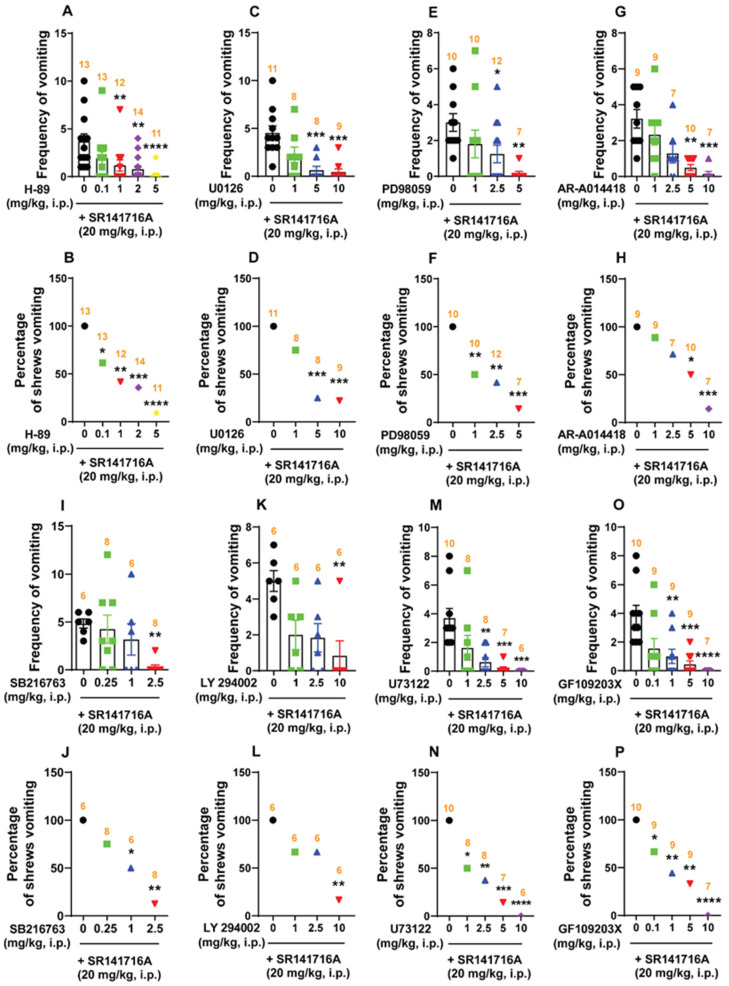
The antiemetic effects of inhibitors of emesis-associated PKA, ERK_1/2_, GSK-3α/β, PI3K, PLC, and PKCαβII proteins on the ability of the CB_1_ receptor inverse agonist/antagonist SR141716A (20 mg/kg, i.p.) to evoke vomiting. Different groups of least shrews were given an injection of either the corresponding vehicle or varying doses of the following: (1) PKA inhibitor H-89 (i.p.); (2) ERK_1/2_ inhibitors U0126 (i.p.) and PD98059 (i.p.); (3) GSK-3α/β inhibitors AR-A014418 (i.p.) and SB216763 (i.p.); (4) PI3K inhibitor LY294002 (i.p.); (5) PLC inhibitor U73122 (i.p.); and (6) PKCαβII inhibitor GF109203X (i.p.), 30 min prior to SR141716A (20 mg/kg, i.p.) injection. Emetic parameters were recorded for the next 30 min. The frequency of emesis (**A**,**C**,**E**,**G**,**I**,**K**,**M**,**O**) was analyzed with Kruskal–Wallis non-parametric one-way ANOVA followed by Dunn’s post hoc test and presented as mean ± SEM. The percentage of shrews vomiting (**B**,**D**,**F**,**H**,**J**,**L**,**N**,**P**) was analyzed using the chi-square test and presented as mean. * *p* < 0.05, ** *p* < 0.01, *** *p* < 0.001, **** *p* < 0.0001 vs. 0 mg/kg. The number of animals in each group is presented at the top of the corresponding column.

**Figure 8 ijms-26-09884-f008:**
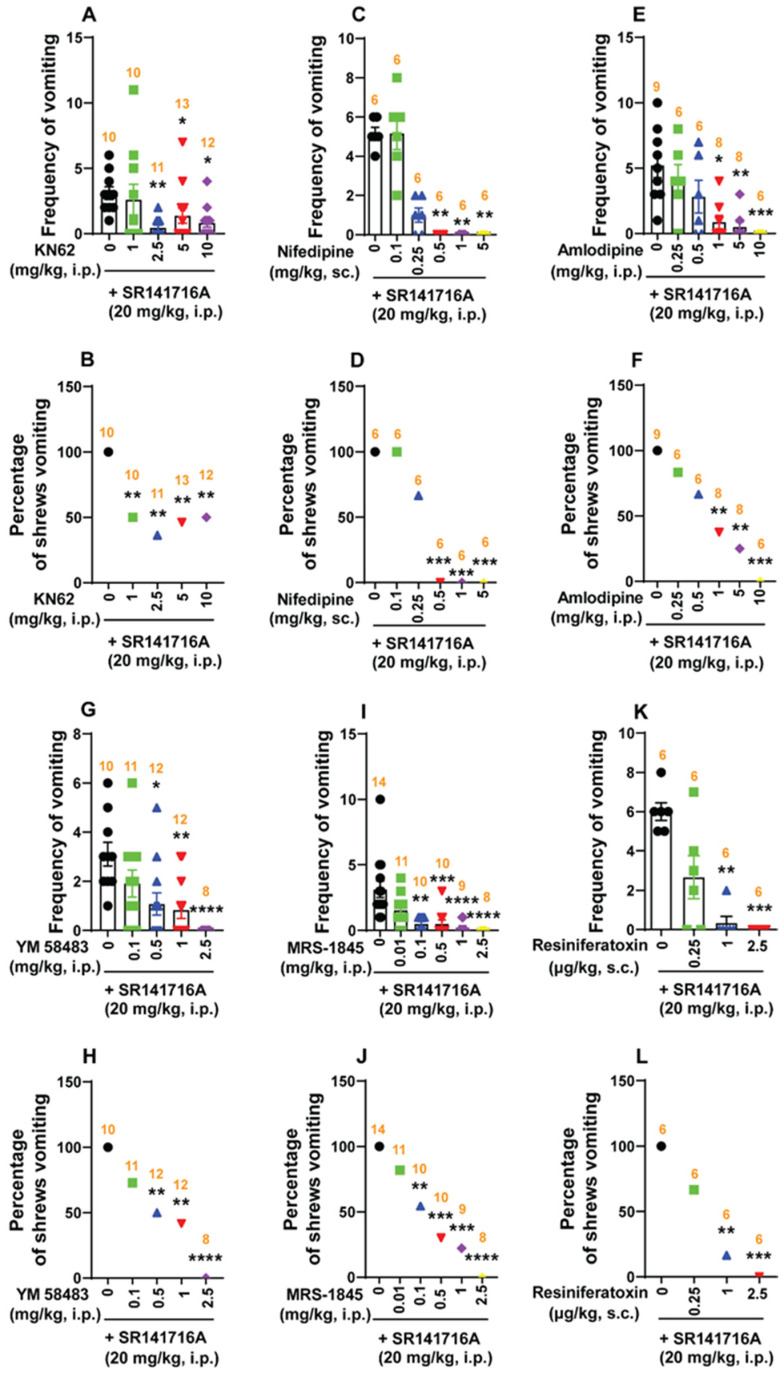
Effects of modulators of cell membrane Ca^2+^ channels on CB_1_ receptor inverse agonist/antagonist SR141716A (20 mg/kg, i.p.)-induced vomiting. Different groups of least shrews were given an injection of either the corresponding vehicle or varying doses of one of the following: (1) CaMKII inhibitor KN62 (i.p.); (2) the L-type Ca^2+^ channel (LTCC) inhibitors nifedipine (s.c.) and amlodipine (i.p.); (3) store-operated Ca^2+^ entry blockers YM 58483 (i.p.) and MRS 1845 (i.p.); and (4) the TRPV1R agonist resiniferatoxin (RTX) (s.c.), 30 min prior to SR141716A (20 mg/kg, i.p.) injection. Emetic parameters were recorded for the next 30 min. The frequency of emesis (**A**,**C**,**E**,**G**,**I**,**K**) was analyzed using Kruskal–Wallis non-parametric one-way ANOVA followed by Dunn’s post hoc test and presented as mean ± SEM. The percentage of shrews vomiting (**B**,**D**,**F**,**H**,**J**,**L**) was analyzed using the chi-square test and presented as mean. * *p* < 0.05, ** *p* < 0.01, *** *p* < 0.001, **** *p* < 0.0001 vs. 0 mg/kg. The number of animals in each group is presented at the top of the corresponding column.

**Figure 9 ijms-26-09884-f009:**
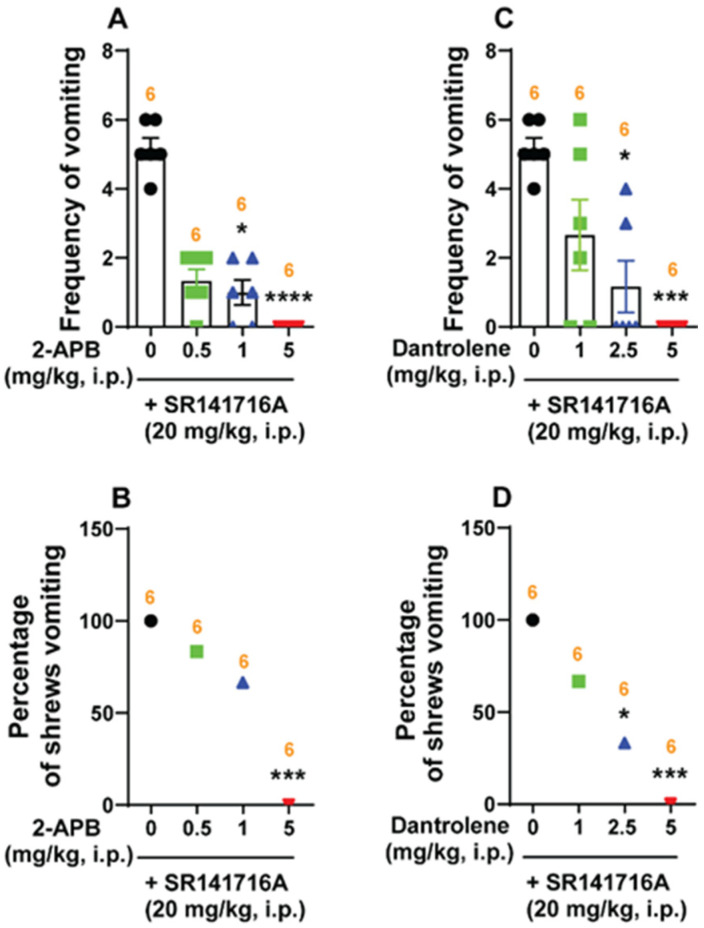
Antiemetic effects of intracellular Ca^2+^ channel blockers on CB_1_ receptor inverse agonist/antagonist SR141716A (20 mg/kg, i.p.)-mediated vomiting. Different groups of least shrews were given an injection of either the corresponding vehicle or varying doses of one of the following: (1) the inositol trisphosphate receptor (IP3R) antagonist 2-APB (i.p.); (2) ryanodine receptor (RyRs) antagonist dantrolene (i.p.), 30 min prior to SR141716A (20 mg/kg, i.p.) injection. Emetic parameters were recorded for the next 30 min. The frequency of emesis (**A**,**C**) was analyzed using Kruskal–Wallis non-parametric one-way ANOVA followed by Dunn’s post hoc test and presented as mean ± SEM. The percentage of shrews vomiting (**B**,**D**) was analyzed using the chi-square test and presented as mean. * *p* < 0.05, *** *p* < 0.001, **** *p* < 0.0001 vs. 0 mg/kg. The number of animals in each group is presented at the top of the corresponding column.

**Figure 10 ijms-26-09884-f010:**
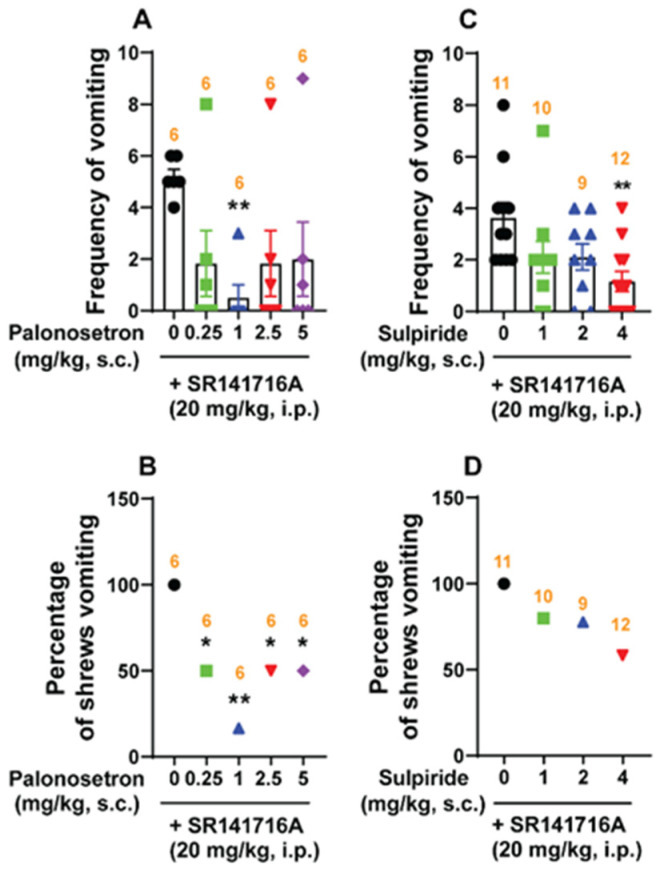
Effects of classical receptor-selective antiemetics against CB_1_ receptor inverse agonist/antagonist SR141716A (20 mg/kg, i.p.)-induced vomiting. Different groups of least shrews were given an injection of either the corresponding vehicle or varying doses of one of the following: (1) 5-HT_3_ receptor antagonist palonosetron (s.c.); (2) the D_2/3_ receptor antagonist sulpiride (s.c.), 30 min prior to SR141716A (20 mg/kg, i.p.) injection. Emetic parameters were recorded for the next 30 min. The frequency of emesis (**A**,**C**) was analyzed using Kruskal–Wallis non-parametric one-way ANOVA followed by Dunn’s post hoc test and presented as mean ± SEM. The percentage of shrews vomiting (**B**,**D**) was analyzed using the chi-square test and presented as mean. * *p* < 0.05, ** *p* < 0.01 vs. 0 mg/kg. The number of animals in each group is presented at the top of the corresponding column.

**Table 1 ijms-26-09884-t001:** Western blot antibody dilutions and manufacturers.

Antibody Name	Antibody Dilution	Manufacturer
Total PKAα/β	1:5000	Invitrogen (Carlsbad, CA, USA)
p-PKAα/β (Thr197)	1:5000	Invitrogen (Carlsbad, CA, USA)
Total Akt	1:2000	Cell Signaling Tech (Danvers, MA, USA)
p-Akt (Ser473)	1:2000	Cell Signaling Tech (Danvers, MA, USA)
Total ERK_1/2_	1:3000	Cell Signaling Tech (Danvers, MA, USA)
p-ERK_1/2_ (Th202/204)	1:1000	Cell Signaling Tech (Danvers, MA, USA)
Total GSK-3α/β mouse	1:2000	Invitrogen Carlsbad, CA, USA)
p-GSK-3α/β rabbit	1:1000	Cell Signaling Tech (Danvers, MA, USA)
p-PKCαβII (Thr638/641)	1:2000	Cell Signaling Tech (Danvers, MA, USA)
GAPDH	1:10,000	Millipore Sigma (Burlington, MA, USA)

## Data Availability

The raw data supporting the conclusions of this article will be made available by the authors, without undue reservation.

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
