# Peer review of "The Cannabinoid CB1 Receptor Inverse Agonist/Antagonist SR141716A Activates the Adenylate Cyclase/PKA Signaling Pathway Among Other Intracellular Emetic Signals to Evoke Vomiting in Least Shrews (Cryptotis parva)"

_ijms, 2025, doi:10.3390/ijms26209884_

Round 1
Reviewer 1 Report (New Reviewer)
Comments and Suggestions for Authors
Sun et al. examined in their paper intracellular emetic signals involved in cannabinoid CB1 receptor inverse agonist/antagonist, SR141716A given at a dose of (20 mg/kg, i.p.) to least shrews (Cryptotis parva). Although the authors suggested numerous potential mechanisms for SR141716A, in my opinion the significance of their conclusions is rather low because of the following reasons:
- SR141716A is the chemical name for rimonabant, which was approved for clinical use in around 40 countries. Unfortunately, its serious CNS-mediated unwanted neuropsychiatric effects such as anxiety and increased suicidal ideation led to its withdrawal from the market in 2008 (e.g. Cinar et al. doi: 10.1016/j.pharmthera.2020.107477).
- Authors wrote: “SR141716A, by itself evokes emesis … and its 20 mg/kg dose causes vomiting in all tested shrews [14]. Likewise, a 20 mg dose of SR141716A causes significant emesis in humans [16].” However, one cannot compare directly a dose of 20 mg/kg in animals with a dose of 20 mg per person (weighing about 70 kg) in humans.
- Authors have sent for the evaluation not final version of their manuscript. I had to read the version with numerous deletions!
Author Response
Referee 1
Sun et al. examined in their paper intracellular emetic signals involved in cannabinoid CB1 receptor inverse agonist/antagonist, SR141716A given at a dose of (20 mg/kg, i.p.) to least shrews (Cryptotis parva). Although the authors suggested numerous potential mechanisms for SR141716A, in my opinion the significance of their conclusions is rather low because of the following reasons:
- SR141716A is the chemical name for rimonabant, which was approved for clinical use in around 40 countries. Unfortunately, its serious CNS-mediated unwanted neuropsychiatric effects such as anxiety and increased suicidal ideationled to its withdrawal from the market in 2008 (e.g. Cinar et al. doi: 10.1016/j.pharmthera.2020.107477).
We agree that SR141716A (rimonabant) was suspended by the European Medicines Agency in 2008 due to neuropsychiatric adverse effects, including anxiety and increased suicidal ideation (Cinar et al., 2020). We would also like to emphasize that, although SR141716A is no longer clinically available, it remains the principal prototypical and most widely studied CB₁ receptor antagonist/inverse agonist. As such, SR141716A has been extensively used as a pharmacological tool to investigate the role of the endocannabinoid system. SR141716A remains highly relevant in experimental neuropharmacology, as it provides unique mechanistic insights into how CB₁ receptor blockade and/or inverse agonism affects constitutive Gi/o signaling—reshapes neuronal and circuit function (Howlett et al., 2002; Pertwee, 2005). By relieving CB₁-mediated inhibition of adenylyl cyclase, SR141716A increases cAMP/PKA activity and engages downstream nodes (ERK1/2, PI3K/Akt–GSK-3, PLC/PKC, CaMKII) and Ca²⁺ mobilization (LTCC/SOCE and IP₃R/RyR-dependent store release) (Howlett et al., 2002; Zhuang et al., 2005). At the synaptic level, it unmasks tonic endocannabinoid control over presynaptic glutamatergic and GABAergic release, revealing how CB₁ tone gates excitability and network gain (Castillo et al., 2012; Robison et al., 2016; Urbanski et al., 2010). At the systems level, it is a powerful tool for dissecting brain–gut-axis pathways: SR141716A-evoked signaling in the DVC (AP/NTS/DMNX) and enteric nervous system highlights how peripheral (vagal/ENS) and central mechanisms converge to drive emetic outputs (Babic and Browning, 2013; Travagli and Anselmi, 2016). Conceptually, comparing SR141716A with neutral CB₁ antagonists [e.g., AM4113, which do not evoke emesis (Salamone et al., 2007, PMID: 17521686)] as well as with peripherally restricted compounds (e.g., JD5037), helps to separate effects due to inverse agonism from simple receptor blockade and distinguishes central from peripheral CB₁ contributions (Sink et al., 2008; Tam et al., 2012).
In our study, its value lies in elucidating the mechanistic role of CB₁ receptor inverse agonism in emesis rather than in its clinical application. We hope that this clarification addresses the reviewer’s concern while also underscoring the continued scientific relevance of SR141716A as an experimental probe, despite its withdrawal from the clinic.
- Authors wrote: “SR141716A, by itself evokes emesis … and its 20 mg/kg dose causes vomiting in all tested shrews [14]. Likewise, a 20 mg dose of SR141716A causes significant emesis in humans [16].” However, one cannot compare directly a dose of 20 mg/kg in animals with a dose of 20 mg per person (weighing about 70 kg) in humans.
We agree and have revised the manuscript to avoid implying cross-species dose equivalence. Indeed, the effective dose of 20 mg/kg used in shrews cannot be directly compared to the that of humans. In the ‘dose-by- or’ approach, the ‘no-observed negative effect level’ (NOAEL) of the drug is scaled by using simple allometry based on body surface area to obtain the ‘human-equivalent dose’ (HED) (PMID: 22407287).
The formula enclosed below is used to calculate the HED according to the FDA guidelines:
HED (mg / kg = Animal NOAEL mg/kg) × (Weightanimal [kg]/Weighthuman [kg])(1–0.67) (PMID: 27057123)
Or simply:
HED mg/kg = Animal mg/kg dose x (animal weight in kg/ Human weight in kg)0.33
Of note, the FDA approach uses the exponent for body surface area, 0.67, to scale doses between species. This practice was normally rationalized as a means of accounting for differences in metabolic rate.
We applied the formula to convert the dose of 20 mg/kg effective dose in shrews to human.
HED= 20 x (0.005/60)0.33 = 0.9 mg/kg
With the mean weight of a shrew = (6 + 4)/2 = 5 g = 0.005 kg
With the average human body weight taken as 60 kg
In conclusion:
Equivalent HED to the 20 mg/ kg effective dose in shrews is (assuming the human weight to be 60 kg) is 0.9 x 60 mg/day= 54 mg
Clinical trials have used a wide range of SR141716A doses varying from 1 to 90 mg per day (e.g., PMID 16504646; PMID 17619859]. The intent of the current study was to note qualitative concordance (that SR141716A/rimonabant can be emetogenic across species), not to equate doses. We therefore limit our discussion to qualitative cross-species concordance and do not extrapolate dose equivalence in the introduction part. which further lengthens the manuscript, and we were asked to reduce the length of the manuscript following its first revision.
We have now revised the introduction section as “We have already shown that large doses (> 10 mg/kg) of SR141716A, by itself evokes emesis in least shrews in a dose-dependent manner [14, 15], and its 20 mg/kg dose causes vomiting in all tested shrews [14], which corresponds to a human equivalent dose of 54 mg assuming human weight of 60 Kg [16]. In clinical trials a diverse range of SR141716A doses (1 – 90 mg per day) have been utilized and a dose of 20 mg/day has been associated with a higher incidence of vomiting relative to placebo in randomized trials [17, 18].” Please see lines 60-66 in the manuscript.
- Authors have sent for the evaluation not final version of their manuscript. I had to read the version with numerous deletions!
We apologize for the confusion. As was required, we had kept Track Changes enabled during the previous revision so that the referee can clearly see that we had significantly reduced the length of the manuscript. We have now accepted track Changes. Please see the revised manuscript.
Reviewer 2 Report (New Reviewer)
Comments and Suggestions for Authors
This study by Sun et. al. presents an extensive investigation into the emetic mechanisms triggered by the cannabinoid CB1 receptor inverse agonist SR141716A in least shrews, revealing complex intracellular signaling pathways involved in vomiting. However, the study, while thorough, leans heavily on descriptive experimental data and would benefit from deeper mechanistic insights.
The authors systematically demonstrate that Rimonabant induces vomiting via both central and peripheral mechanisms, chiefly the dorsal vagal complex (DVC) in the brainstem and the enteric nervous system in the jejunum, with evidence from c-fos and phospho-ERK1/2 immunfluorescence, among others. They provide a link between brainstem cAMP levels and sequential phosphorylation of key kinases such as PKA, Akt, GSK-3α/β, ERK1/2, and PKCαβII to the emetic response. The broad-spectrum pharmacological blockade studies targeting these kinases and various calcium channels strongly support their involvement in SR141716A-evoked vomiting.
Critical Issues:
- Antibody Validity for CB1 Detection and general antibody-related issues: A key weakness undermining the interpretation of CB1 receptor distribution is the use of a commercial anti-CB1 antibody (ab3558, Abcam) which has been withdrawn from the market by the company. This raises questions regarding the specificity and reliability of CB1 receptor immunostaining data presented. In fact, previous studies (Histochem Cell Biol. 2021 Aug 27;156(5):479–502, J Neurosci Methods. 2008 Jun 15;171(1):78-8) investigated CB1 antibody specificity and found that many CB1 antibodies do not specifically detect the protein. Staining specificity of the antibody used should be either verified by using another CB1 antibody, another technique (Western Blot) or by the use of knockout animals (in this case not possible). In Fig. 1 CB1 staining is shown. However, staining doesn’t look specific since I only see a blurry reddish image. Maybe supplying a magnification could shed some light whether the antibody stains specifically. A quantification of only those cells that are labelled positive is not a very good method since this leaves out cells with low level expression. Rather, staining intensity should be considered for quantification. In addition, I would like to see an isotype staining, especially as a control for CB1.
- cfos immunoreactivity seems to be highest outside the areas of interest. Why? Do those cells carry CB1? Why didn’t the authors co-stain with CB1?
- cAMP levels: A positive control would have helped to put these values into perspective. The authors mentioned that rolipram has a similar effect. Why not use this as positive control?
- The study overwhelmingly shows descriptive data without explaining the precise molecular interactions between SR141716A and intracellular effectors. The complexity of GPCR biased signaling and the effects of inverse agonism by rimonabant is acknowledged but insufficiently explored, missing an opportunity to provide insight into why SR141716A acts as an inverse agonist triggering emesis while agonists generally confer antiemetic effects. Are these pathways that were investigated all directly linked to CB1 inverse agonism? Would the compounds used also inhibit vomiting induced by a different mechanism? This would be interesting, because then the observed effects would not be solely attributable to CB1 inhibition. This also leads me to another question: Is the activation of the described intracellular pathways by rimonabant a CB1 specific mechanism or is this a general activation scheme common to all emetic compounds?
- Although multiple pathways are implicated, the study does not sufficiently address alternative or compensatory signaling cascades that may contribute to the emetic response. For example, the lack of increase in endogenous serotonin (5-HT) and substance P (SP) tissue levels contradicts the findings that serotonin and dopamine receptor antagonists are efficacious in the described set-up. This discrepancy merits closer examination, possibly involving release kinetics, metabolism, or receptor sensitivity. Same for pERK: ERK would also be activated by CB1 agonists, which are usually anti-emetic. Please elaborate.
Similarly, cannabis hyperemesis syndrome is caused by CB1 agonists. This is one major side effect in some patients and since it can be attenuated by TRPV1 agonists, this should be discussed here as well.
- The peripheral involvement has only been shown indirectly through stainings in the ENS. However, whether those changes are emetic has not been shown. This can be shown by vagotomy but all evidence shown here is indirect.
- Translational and Clinical Implications are vague: The study's focus on mechanistic dissection in the least shrew model does not adequately bridge to clinical implications for cannabinoid-based antiemetic therapy or adverse emetic effects associated with CB1 inhibitors. More explicit discussion on how these findings may inform drug design or treatment of nausea and vomiting in patients would increase relevance.
Author Response
Referee 2
Comments and Suggestions for Authors
This study by Sun et al. presents an extensive investigation into the emetic mechanisms triggered by the cannabinoid CB1 receptor inverse agonist SR141716A in least shrews, revealing complex intracellular signaling pathways involved in vomiting. However, the study, while thorough, leans heavily on descriptive experimental data and would benefit from deeper mechanistic insights.
The authors systematically demonstrate that Rimonabant induces vomiting via both central and peripheral mechanisms, chiefly the dorsal vagal complex (DVC) in the brainstem and the enteric nervous system in the jejunum, with evidence from c-fos and phospho-ERK1/2 immunfluorescence, among others. They provide a link between brainstem cAMP levels and sequential phosphorylation of key kinases such as PKA, Akt, GSK-3α/β, ERK1/2, and PKCαβII to the emetic response. The broad-spectrum pharmacological blockade studies targeting these kinases and various calcium channels strongly support their involvement in SR141716A-evoked vomiting.
Critical Issues:
- Antibody Validity for CB1 Detection and general antibody-related issues: A key weakness undermining the interpretation of CB1 receptor distribution is the use of a commercial anti-CB1 antibody (ab3558, Abcam) which has been withdrawn from the market by the company. This raises questions regarding the specificity and reliability of CB1 receptor immunostaining data presented. In fact, previous studies (Histochem Cell Biol. 2021 Aug 27;156(5):479–502, J Neurosci Methods. 2008 Jun 15;171(1):78-8) investigated CB1 antibody specificity and found that many CB1 antibodies do not specifically detect the protein. Staining specificity of the antibody used should be either verified by using another CB1 antibody, another technique (Western Blot) or by the use of knockout animals (in this case not possible). In Fig. 1 CB1 staining is shown. However, staining doesn’t look specific since I only see a blurry reddish image. Maybe supplying a magnification could shed some light whether the antibody stains specifically. A quantification of only those cells that are labelled positive is not a very good method since this leaves out cells with low level expression. Rather, staining intensity should be considered for quantification. In addition, I would like to see an isotype staining, especially as a control for CB1.
We agree that CB1 antibodies vary in performance. In our study, CB1 immunostaining was used only to provide regional context (AP/NTS/DMNX mapping) and to note that acute SR141716A did not alter regional CB1 density; our key conclusions (↑cAMP/PKA, engagement of ERK/PI3K–Akt–GSK-3/PLC–PKC/CaMKII, Ca²⁺ mobilization, and inhibitor sensitivity of emesis) do not depend on subtle intensity gradients of CB1 immunoreactivity.
We acknowledge that ab3558 has been discontinued by the vendor; however, discontinuation alone is not evidence of non-specificity. Several studies have used this antibody to detect CB1 receptor expression, including recent publications (Shen et al., 2010; PMID: 20680167; Wang et al., 2022; PMID: 35269905; Sevil et al., 2024; PMID: 39758842; Köse et al., 2024; PMID: 39354544). Given the 10-day revision window, it is not feasible to obtain and validate an alternative antibody and repeat these experiments. Instead of re-performing this component with a different antibody, we strengthened validation and presentation of the existing dataset as follows:
- Negative controls: Secondary-only and host-matched rabbit IgG isotype controls were processed in parallel and imaged with identical acquisition parameters to quantify background; these data are now included. Please see attached supplemental material.
- High-magnification imaging: As noted in Section 4.4.1, tile-scan images were acquired on a Zeiss LSM 880 confocal microscope (1024 × 1024 pixels; Plan-Apochromat 63×/1.4 Oil DIC M27 objective). We now added high-magnification insets in Fig. 1E–G from the boxed regions in panel C (AP, NTS, DMNX).
- Re-analysis: CB1 signal was quantified as mean gray value per µm² within predefined ROIs.
- c-fos immunoreactivity seems to be highest outside the areas of interest. Why? Do those cells carry CB1? Why didn’t the authors co-stain with CB1?
The c-fos panels are tile scans that necessarily include tissue adjacent to the dorsal vagal complex (DVC). However, our quantification was restricted to the prespecified emetic nuclei—AP, NTS, and DMNX. The DVC contains not only neurons that express CB1 receptors but also other neuronal types (e.g., glutamatergic and GABAergic), and its nuclei have complex connections with other brain regions, such as the hypothalamus, amygdala, cortex, and so on. In this study, drug administration was systemic (via the circulation) rather than a local injection into the DVC. The goal of this study was therefore to evaluate the overall activation of the DVC after SR141716A administration rather than cellular co-localization. Functionally, SR141716A-evoked emesis is multideterminant (involving cAMP/PKA, ERK1/2, PI3K/Akt/GSK-3, PLC/PKC, CaMKII, and the classical 5-HT3 and D2/3 receptors), so co-localizing c-fos with CB1 would not, by itself, distinguish activation of CB1-expressing neurons from downstream network activation.
- cAMP levels: A positive control would have helped to put these values into perspective. The authors mentioned that rolipram has a similar effect. Why not use this as positive control?
We thank the reviewer for this insightful suggestion. In the present study, we observed a significant increase in cAMP levels following SR141716A administration, consistent with its established role as a CB₁ inverse agonist that disinhibits adenylate cyclase activity. Although rolipram, a phosphodiesterase-4 inhibitor, is known to elevate intracellular cAMP, we did not include it as a positive control here because we have previously reported that rolipram significantly increases brainstem cAMP in the least shrew (Alkam et al., 2014-PMID: 24513510), which supports our present findings. Nevertheless, we agree that such a control would strengthen the interpretation, and we will incorporate rolipram or other reference modulators in future studies to provide a clearer comparative context for cAMP signaling.
- The study overwhelmingly shows descriptive data without explaining the precise molecular interactions between SR141716A and intracellular effectors. The complexity of GPCR biased signaling and the effects of inverse agonism by rimonabant is acknowledged but insufficiently explored, missing an opportunity to provide insight into why SR141716A acts as an inverse agonist triggering emesis while agonists generally confer antiemetic effects. Are these pathways that were investigated all directly linked to CB1 inverse agonism? Would the compounds used also inhibit vomiting induced by a different mechanism? This would be interesting, because then the observed effects would not be solely attributable to CB1 inhibition. This also leads me to another question: Is the activation of the described intracellular pathways by rimonabant a CB1 specific mechanism or is this a general activation scheme common to all emetic compounds?
In this study, we have demonstrated that SR141716A (20 mg/kg, i.p.) produced a time-dependent increase in brainstem cAMP (↑ at 15 and 30 min), a peak in PKA (Thr197) phosphorylation at 30 min, and rapid phosphorylation of ERK1/2, Akt, GSK-3α/β, and PKCα/βII (peaking at ~15 min and persisting for some nodes). These biochemical changes coincided with the observed emesis and DVC activation (↑c-fos; ↑p-ERK1/2), establishing temporal linkage between receptor perturbation and effector pathways. Functionally, pharmacologic inhibition of these emetic signals (PKA, ERK1/2, GSK-3, PI3K, PLC, PKC, CaMKII; membrane Ca²⁺ entry via LTCC and SOCE; intracellular Ca²⁺ release via IP₃R and RyR) significantly reduced SR141716A-evoked vomiting in a dose-dependent manner, indicating these pathways are necessary for the behavioral output under our conditions.
CB₁ receptor predominantly couples to Gi/o to suppress adenylyl cyclase. The observed ↑cAMP/↑PKA after SR141716A is the expected direct consequence of inverse agonism (relief of constitutive Gi/o tone). Also, we found no acute change in CB1 protein density in DVC after SR141716A, arguing that the signaling pattern reflects a change in receptor signaling state rather than receptor density. Engagement of ERK, PI3K/Akt/GSK-3, PLC/PKC, CaMKII, and Ca²⁺ mobilization (LTCC/SOCE and IP₃R/RyR) is consistent with well-known cross-talk downstream of cAMP/PKA in emetic nuclei. In short, cAMP/PKA is a direct readout of CB1 inverse agonism, and the other activated intracellular emetic signals represent downstream activation within the emetic network. Our data show that blocking any of these signals attenuates emesis, so they are causally involved even if not all are “direct” CB1 outputs in every neuron.
In our previous studies, we have demonstrated that the tested compounds used in this study also inhibited shrews vomiting induced by diverse emetogens. In this study, we show that classical antiemetics acting at 5-HT₃ (palonosetron) and D₂/₃ (sulpiride) receptors attenuate SR141716A-evoked vomiting, and that Ca²⁺-channel blockers (LTCC, SOCE, IP₃R, RyR) and kinase inhibitors (PKA, ERK, GSK-3, PI3K, PLC, PKC, CaMKII) are effective against SR141716A-evoked vomiting. Because several of these drug classes are known to act across multiple emetic triggers, their efficacy here indicates that SR141716A engages shared emetic machinery; i.e., the downstream effects are not solely attributable to CB1 blockade per se, but though CB1 inverse agonism is the initiating event in our paradigm.
Our data indicate that SR141716A-induced vomiting is initiated by a CB₁ receptor–specific mechanism. Specifically: (1) Upstream (CB₁-specific): SR141716A’s inverse agonism at CB₁ relieves Gi/o-mediated restraint on adenylyl cyclase, increasing cAMP/PKA and initiating the intracellular cascade that culminates in emesis; (2) Downstream (general/convergent): the execution layer—ERK, PI3K/Akt–GSK-3, PLC/PKC, CaMKII, and coordinated Ca²⁺ entry/release—constitutes a general emetic program engaged by multiple triggers; accordingly, blocking these signals inhibits SR141716A-evoked vomiting.
In summary, SR141716A triggers emesis via CB1 inverse agonism (CB1-specific initiation), but the response is implemented by shared intracellular effectors (general convergence). Our time-course biochemistry plus pathway-targeted inhibition convert the findings from descriptive to mechanistically causal for the emetic phenotype studied here.
- Although multiple pathways are implicated, the study does not sufficiently address alternative or compensatory signaling cascades that may contribute to the emetic response. For example, the lack of increase in endogenous serotonin (5-HT) and substance P (SP) tissue levels contradicts the findings that serotonin and dopamine receptor antagonists are efficacious in the described set-up. This discrepancy merits closer examination, possibly involving release kinetics, metabolism, or receptor sensitivity. Same for p-ERK: ERK would also be activated by CB1 agonists, which are usually anti-emetic. Please elaborate. Similarly, cannabis hyperemesis syndrome is caused by CB1 agonists. This is one major side effect in some patients and since it can be attenuated by TRPV1 agonists, this should be discussed here as well.
1) We agree that although SR141716A treatment did not increase tissue levels of serotonin (5-HT) or substance P (SP) in the dorsal vagal complex, the efficacy of 5-HT₃ and D₂ receptor antagonists in attenuating SR141716A-induced emesis suggests that receptor-level sensitivity, release dynamics, or neurotransmitter turnover may play a more critical role than static tissue levels.
We appreciate the referee’s suggestion, and in our re-revised manuscript we have added the following discussion in the manuscript, “iv) the release dynamics, or receptor-level sensitivity may also play a more critical role than static tissue levels,” Please see lines 846-847.
2) p-ERK1/2 functions as a convergent integrator of multiple inputs; therefore, the behavioral outcome depends on the net effect at the circuit level. CB1 agonists generally reduce presynaptic glutamate/5-HT/GABA release (Gi/o tone), producing an overall antiemetic network effect, even if ERK1/2 is transiently activated (e.g., via β-arrestin pathways). SR141716A, as a CB1 inverse agonist, disinhibits AC→cAMP/PKA, permitting broader Ca²⁺ mobilization (LTCC/SOCE/IP₃R/RyR) and ERK/PI3K–Akt–GSK-3/PLC–PKC/CaMKII engagement that together promote emesis.
3) Per referee 2’s suggestion, we have added the following discussion in the manuscript, “Cannabis hyperemesis syndrome (CHS) occurs following chronic administration of large doses of CB₁ agonists, further illustrates the complexity of endocannabinoid signaling. Interestingly, TRPV1R agonists such as capsaicin provide relief in CHS patients, supporting the idea that TRPV1R-CB₁ crosstalk and compensatory adaptations of intracellular pathways are critical determinants of the emetic phenotype [128, 129]. In fact, we have demonstrated that a combination of CB1- and TRPV1 receptor agonists have the capacity to completely abolish cisplatin-evoked emesis at doses that are ineffective when used individually [121].” Please see lines 776-783.
- The peripheral involvement has only been shown indirectly through stainings in the ENS. However, whether those changes are emetic has not been shown. This can be shown by vagotomy but all evidence shown here is indirect.
The DVC is the canonical integrator of vagal afferents driving emesis, so concurrent ENS and DVC activation is expected if the brain–gut axis is engaged. ENS activity markers rise in the same time window as vomiting and DVC (AP/NTS/DMNX) activation, compatible with a gut → vagus → DVC pathway. Our ENS staining together with DVC activation supports involvement of the brain–gut axis during SR141716A-evoked emesis. Accordingly, in future studies we will perform subdiaphragmatic vagotomy or reversible perivagal blockade to determine the involvement of the entire brain–gut axis in the induced emetic response. We thank you for your suggestion!
- Translational and Clinical Implications are vague: The study's focus on mechanistic dissection in the least shrew model does not adequately bridge to clinical implications for cannabinoid-based antiemetic therapy or adverse emetic effects associated with CB1 inhibitors. More explicit discussion on how these findings may inform drug design or treatment of nausea and vomiting in patients would increase relevance.
While this study was conducted in the least shrew, a well-validated model for emesis research, the mechanistic findings have potential translational relevance for human conditions. The ability of SR141716A to trigger vomiting through CB₁ inverse agonism underscores the critical role of endocannabinoid signaling in emetic regulation and helps explain why SR141716A was associated with adverse gastrointestinal and neuropsychiatric effects in the clinic. Furthermore, our results suggest that targeting downstream effectors such as ERK1/2, PI3K/Akt, and calcium signaling may provide alternative strategies for antiemetic drug development beyond conventional 5-HT₃ and dopamine receptor antagonists. The observed overlap with signaling pathways implicated in CHS also highlights the translational relevance of our findings, as they may help inform management strategies for patients suffering from this increasingly recognized condition. Overall, while cannabis cessation remains the definitive treatment for CHS, our data suggest that pharmacological modulation of convergent signaling cascades may hold promise as adjunctive therapies. We cannot provide further details since we were asked to reduce the length of the original manuscript.
Referee 2
Comments and Suggestions for Authors
This study by Sun et al. presents an extensive investigation into the emetic mechanisms triggered by the cannabinoid CB1 receptor inverse agonist SR141716A in least shrews, revealing complex intracellular signaling pathways involved in vomiting. However, the study, while thorough, leans heavily on descriptive experimental data and would benefit from deeper mechanistic insights.
The authors systematically demonstrate that Rimonabant induces vomiting via both central and peripheral mechanisms, chiefly the dorsal vagal complex (DVC) in the brainstem and the enteric nervous system in the jejunum, with evidence from c-fos and phospho-ERK1/2 immunfluorescence, among others. They provide a link between brainstem cAMP levels and sequential phosphorylation of key kinases such as PKA, Akt, GSK-3α/β, ERK1/2, and PKCαβII to the emetic response. The broad-spectrum pharmacological blockade studies targeting these kinases and various calcium channels strongly support their involvement in SR141716A-evoked vomiting.
Critical Issues:
- Antibody Validity for CB1 Detection and general antibody-related issues: A key weakness undermining the interpretation of CB1 receptor distribution is the use of a commercial anti-CB1 antibody (ab3558, Abcam) which has been withdrawn from the market by the company. This raises questions regarding the specificity and reliability of CB1 receptor immunostaining data presented. In fact, previous studies (Histochem Cell Biol. 2021 Aug 27;156(5):479–502, J Neurosci Methods. 2008 Jun 15;171(1):78-8) investigated CB1 antibody specificity and found that many CB1 antibodies do not specifically detect the protein. Staining specificity of the antibody used should be either verified by using another CB1 antibody, another technique (Western Blot) or by the use of knockout animals (in this case not possible). In Fig. 1 CB1 staining is shown. However, staining doesn’t look specific since I only see a blurry reddish image. Maybe supplying a magnification could shed some light whether the antibody stains specifically. A quantification of only those cells that are labelled positive is not a very good method since this leaves out cells with low level expression. Rather, staining intensity should be considered for quantification. In addition, I would like to see an isotype staining, especially as a control for CB1.
We agree that CB1 antibodies vary in performance. In our study, CB1 immunostaining was used only to provide regional context (AP/NTS/DMNX mapping) and to note that acute SR141716A did not alter regional CB1 density; our key conclusions (↑cAMP/PKA, engagement of ERK/PI3K–Akt–GSK-3/PLC–PKC/CaMKII, Ca²⁺ mobilization, and inhibitor sensitivity of emesis) do not depend on subtle intensity gradients of CB1 immunoreactivity.
We acknowledge that ab3558 has been discontinued by the vendor; however, discontinuation alone is not evidence of non-specificity. Several studies have used this antibody to detect CB1 receptor expression, including recent publications (Shen et al., 2010; PMID: 20680167; Wang et al., 2022; PMID: 35269905; Sevil et al., 2024; PMID: 39758842; Köse et al., 2024; PMID: 39354544). Given the 10-day revision window, it is not feasible to obtain and validate an alternative antibody and repeat these experiments. Instead of re-performing this component with a different antibody, we strengthened validation and presentation of the existing dataset as follows:
- Negative controls: Secondary-only and host-matched rabbit IgG isotype controls were processed in parallel and imaged with identical acquisition parameters to quantify background; these data are now included. Please see attached supplemental material.
- High-magnification imaging: As noted in Section 4.4.1, tile-scan images were acquired on a Zeiss LSM 880 confocal microscope (1024 × 1024 pixels; Plan-Apochromat 63×/1.4 Oil DIC M27 objective). We now added high-magnification insets in Fig. 1E–G from the boxed regions in panel C (AP, NTS, DMNX).
- Re-analysis: CB1 signal was quantified as mean gray value per µm² within predefined ROIs.
- c-fos immunoreactivity seems to be highest outside the areas of interest. Why? Do those cells carry CB1? Why didn’t the authors co-stain with CB1?
The c-fos panels are tile scans that necessarily include tissue adjacent to the dorsal vagal complex (DVC). However, our quantification was restricted to the prespecified emetic nuclei—AP, NTS, and DMNX. The DVC contains not only neurons that express CB1 receptors but also other neuronal types (e.g., glutamatergic and GABAergic), and its nuclei have complex connections with other brain regions, such as the hypothalamus, amygdala, cortex, and so on. In this study, drug administration was systemic (via the circulation) rather than a local injection into the DVC. The goal of this study was therefore to evaluate the overall activation of the DVC after SR141716A administration rather than cellular co-localization. Functionally, SR141716A-evoked emesis is multideterminant (involving cAMP/PKA, ERK1/2, PI3K/Akt/GSK-3, PLC/PKC, CaMKII, and the classical 5-HT3 and D2/3 receptors), so co-localizing c-fos with CB1 would not, by itself, distinguish activation of CB1-expressing neurons from downstream network activation.
- cAMP levels: A positive control would have helped to put these values into perspective. The authors mentioned that rolipram has a similar effect. Why not use this as positive control?
We thank the reviewer for this insightful suggestion. In the present study, we observed a significant increase in cAMP levels following SR141716A administration, consistent with its established role as a CB₁ inverse agonist that disinhibits adenylate cyclase activity. Although rolipram, a phosphodiesterase-4 inhibitor, is known to elevate intracellular cAMP, we did not include it as a positive control here because we have previously reported that rolipram significantly increases brainstem cAMP in the least shrew (Alkam et al., 2014-PMID: 24513510), which supports our present findings. Nevertheless, we agree that such a control would strengthen the interpretation, and we will incorporate rolipram or other reference modulators in future studies to provide a clearer comparative context for cAMP signaling.
- The study overwhelmingly shows descriptive data without explaining the precise molecular interactions between SR141716A and intracellular effectors. The complexity of GPCR biased signaling and the effects of inverse agonism by rimonabant is acknowledged but insufficiently explored, missing an opportunity to provide insight into why SR141716A acts as an inverse agonist triggering emesis while agonists generally confer antiemetic effects. Are these pathways that were investigated all directly linked to CB1 inverse agonism? Would the compounds used also inhibit vomiting induced by a different mechanism? This would be interesting, because then the observed effects would not be solely attributable to CB1 inhibition. This also leads me to another question: Is the activation of the described intracellular pathways by rimonabant a CB1 specific mechanism or is this a general activation scheme common to all emetic compounds?
In this study, we have demonstrated that SR141716A (20 mg/kg, i.p.) produced a time-dependent increase in brainstem cAMP (↑ at 15 and 30 min), a peak in PKA (Thr197) phosphorylation at 30 min, and rapid phosphorylation of ERK1/2, Akt, GSK-3α/β, and PKCα/βII (peaking at ~15 min and persisting for some nodes). These biochemical changes coincided with the observed emesis and DVC activation (↑c-fos; ↑p-ERK1/2), establishing temporal linkage between receptor perturbation and effector pathways. Functionally, pharmacologic inhibition of these emetic signals (PKA, ERK1/2, GSK-3, PI3K, PLC, PKC, CaMKII; membrane Ca²⁺ entry via LTCC and SOCE; intracellular Ca²⁺ release via IP₃R and RyR) significantly reduced SR141716A-evoked vomiting in a dose-dependent manner, indicating these pathways are necessary for the behavioral output under our conditions.
CB₁ receptor predominantly couples to Gi/o to suppress adenylyl cyclase. The observed ↑cAMP/↑PKA after SR141716A is the expected direct consequence of inverse agonism (relief of constitutive Gi/o tone). Also, we found no acute change in CB1 protein density in DVC after SR141716A, arguing that the signaling pattern reflects a change in receptor signaling state rather than receptor density. Engagement of ERK, PI3K/Akt/GSK-3, PLC/PKC, CaMKII, and Ca²⁺ mobilization (LTCC/SOCE and IP₃R/RyR) is consistent with well-known cross-talk downstream of cAMP/PKA in emetic nuclei. In short, cAMP/PKA is a direct readout of CB1 inverse agonism, and the other activated intracellular emetic signals represent downstream activation within the emetic network. Our data show that blocking any of these signals attenuates emesis, so they are causally involved even if not all are “direct” CB1 outputs in every neuron.
In our previous studies, we have demonstrated that the tested compounds used in this study also inhibited shrews vomiting induced by diverse emetogens. In this study, we show that classical antiemetics acting at 5-HT₃ (palonosetron) and D₂/₃ (sulpiride) receptors attenuate SR141716A-evoked vomiting, and that Ca²⁺-channel blockers (LTCC, SOCE, IP₃R, RyR) and kinase inhibitors (PKA, ERK, GSK-3, PI3K, PLC, PKC, CaMKII) are effective against SR141716A-evoked vomiting. Because several of these drug classes are known to act across multiple emetic triggers, their efficacy here indicates that SR141716A engages shared emetic machinery; i.e., the downstream effects are not solely attributable to CB1 blockade per se, but though CB1 inverse agonism is the initiating event in our paradigm.
Our data indicate that SR141716A-induced vomiting is initiated by a CB₁ receptor–specific mechanism. Specifically: (1) Upstream (CB₁-specific): SR141716A’s inverse agonism at CB₁ relieves Gi/o-mediated restraint on adenylyl cyclase, increasing cAMP/PKA and initiating the intracellular cascade that culminates in emesis; (2) Downstream (general/convergent): the execution layer—ERK, PI3K/Akt–GSK-3, PLC/PKC, CaMKII, and coordinated Ca²⁺ entry/release—constitutes a general emetic program engaged by multiple triggers; accordingly, blocking these signals inhibits SR141716A-evoked vomiting.
In summary, SR141716A triggers emesis via CB1 inverse agonism (CB1-specific initiation), but the response is implemented by shared intracellular effectors (general convergence). Our time-course biochemistry plus pathway-targeted inhibition convert the findings from descriptive to mechanistically causal for the emetic phenotype studied here.
- Although multiple pathways are implicated, the study does not sufficiently address alternative or compensatory signaling cascades that may contribute to the emetic response. For example, the lack of increase in endogenous serotonin (5-HT) and substance P (SP) tissue levels contradicts the findings that serotonin and dopamine receptor antagonists are efficacious in the described set-up. This discrepancy merits closer examination, possibly involving release kinetics, metabolism, or receptor sensitivity. Same for p-ERK: ERK would also be activated by CB1 agonists, which are usually anti-emetic. Please elaborate. Similarly, cannabis hyperemesis syndrome is caused by CB1 agonists. This is one major side effect in some patients and since it can be attenuated by TRPV1 agonists, this should be discussed here as well.
1) We agree that although SR141716A treatment did not increase tissue levels of serotonin (5-HT) or substance P (SP) in the dorsal vagal complex, the efficacy of 5-HT₃ and D₂ receptor antagonists in attenuating SR141716A-induced emesis suggests that receptor-level sensitivity, release dynamics, or neurotransmitter turnover may play a more critical role than static tissue levels.
We appreciate the referee’s suggestion, and in our re-revised manuscript we have added the following discussion in the manuscript, “iv) the release dynamics, or receptor-level sensitivity may also play a more critical role than static tissue levels,” Please see lines 846-847.
2) p-ERK1/2 functions as a convergent integrator of multiple inputs; therefore, the behavioral outcome depends on the net effect at the circuit level. CB1 agonists generally reduce presynaptic glutamate/5-HT/GABA release (Gi/o tone), producing an overall antiemetic network effect, even if ERK1/2 is transiently activated (e.g., via β-arrestin pathways). SR141716A, as a CB1 inverse agonist, disinhibits AC→cAMP/PKA, permitting broader Ca²⁺ mobilization (LTCC/SOCE/IP₃R/RyR) and ERK/PI3K–Akt–GSK-3/PLC–PKC/CaMKII engagement that together promote emesis.
3) Per referee 2’s suggestion, we have added the following discussion in the manuscript, “Cannabis hyperemesis syndrome (CHS) occurs following chronic administration of large doses of CB₁ agonists, further illustrates the complexity of endocannabinoid signaling. Interestingly, TRPV1R agonists such as capsaicin provide relief in CHS patients, supporting the idea that TRPV1R-CB₁ crosstalk and compensatory adaptations of intracellular pathways are critical determinants of the emetic phenotype [128, 129]. In fact, we have demonstrated that a combination of CB1- and TRPV1 receptor agonists have the capacity to completely abolish cisplatin-evoked emesis at doses that are ineffective when used individually [121].” Please see lines 776-783.
- The peripheral involvement has only been shown indirectly through stainings in the ENS. However, whether those changes are emetic has not been shown. This can be shown by vagotomy but all evidence shown here is indirect.
The DVC is the canonical integrator of vagal afferents driving emesis, so concurrent ENS and DVC activation is expected if the brain–gut axis is engaged. ENS activity markers rise in the same time window as vomiting and DVC (AP/NTS/DMNX) activation, compatible with a gut → vagus → DVC pathway. Our ENS staining together with DVC activation supports involvement of the brain–gut axis during SR141716A-evoked emesis. Accordingly, in future studies we will perform subdiaphragmatic vagotomy or reversible perivagal blockade to determine the involvement of the entire brain–gut axis in the induced emetic response. We thank you for your suggestion!
- Translational and Clinical Implications are vague: The study's focus on mechanistic dissection in the least shrew model does not adequately bridge to clinical implications for cannabinoid-based antiemetic therapy or adverse emetic effects associated with CB1 inhibitors. More explicit discussion on how these findings may inform drug design or treatment of nausea and vomiting in patients would increase relevance.
While this study was conducted in the least shrew, a well-validated model for emesis research, the mechanistic findings have potential translational relevance for human conditions. The ability of SR141716A to trigger vomiting through CB₁ inverse agonism underscores the critical role of endocannabinoid signaling in emetic regulation and helps explain why SR141716A was associated with adverse gastrointestinal and neuropsychiatric effects in the clinic. Furthermore, our results suggest that targeting downstream effectors such as ERK1/2, PI3K/Akt, and calcium signaling may provide alternative strategies for antiemetic drug development beyond conventional 5-HT₃ and dopamine receptor antagonists. The observed overlap with signaling pathways implicated in CHS also highlights the translational relevance of our findings, as they may help inform management strategies for patients suffering from this increasingly recognized condition. Overall, while cannabis cessation remains the definitive treatment for CHS, our data suggest that pharmacological modulation of convergent signaling cascades may hold promise as adjunctive therapies. We cannot provide further details since we were asked to reduce the length of the original manuscript.
Referee 2
Comments and Suggestions for Authors
This study by Sun et al. presents an extensive investigation into the emetic mechanisms triggered by the cannabinoid CB1 receptor inverse agonist SR141716A in least shrews, revealing complex intracellular signaling pathways involved in vomiting. However, the study, while thorough, leans heavily on descriptive experimental data and would benefit from deeper mechanistic insights.
The authors systematically demonstrate that Rimonabant induces vomiting via both central and peripheral mechanisms, chiefly the dorsal vagal complex (DVC) in the brainstem and the enteric nervous system in the jejunum, with evidence from c-fos and phospho-ERK1/2 immunfluorescence, among others. They provide a link between brainstem cAMP levels and sequential phosphorylation of key kinases such as PKA, Akt, GSK-3α/β, ERK1/2, and PKCαβII to the emetic response. The broad-spectrum pharmacological blockade studies targeting these kinases and various calcium channels strongly support their involvement in SR141716A-evoked vomiting.
Critical Issues:
- Antibody Validity for CB1 Detection and general antibody-related issues: A key weakness undermining the interpretation of CB1 receptor distribution is the use of a commercial anti-CB1 antibody (ab3558, Abcam) which has been withdrawn from the market by the company. This raises questions regarding the specificity and reliability of CB1 receptor immunostaining data presented. In fact, previous studies (Histochem Cell Biol. 2021 Aug 27;156(5):479–502, J Neurosci Methods. 2008 Jun 15;171(1):78-8) investigated CB1 antibody specificity and found that many CB1 antibodies do not specifically detect the protein. Staining specificity of the antibody used should be either verified by using another CB1 antibody, another technique (Western Blot) or by the use of knockout animals (in this case not possible). In Fig. 1 CB1 staining is shown. However, staining doesn’t look specific since I only see a blurry reddish image. Maybe supplying a magnification could shed some light whether the antibody stains specifically. A quantification of only those cells that are labelled positive is not a very good method since this leaves out cells with low level expression. Rather, staining intensity should be considered for quantification. In addition, I would like to see an isotype staining, especially as a control for CB1.
We agree that CB1 antibodies vary in performance. In our study, CB1 immunostaining was used only to provide regional context (AP/NTS/DMNX mapping) and to note that acute SR141716A did not alter regional CB1 density; our key conclusions (↑cAMP/PKA, engagement of ERK/PI3K–Akt–GSK-3/PLC–PKC/CaMKII, Ca²⁺ mobilization, and inhibitor sensitivity of emesis) do not depend on subtle intensity gradients of CB1 immunoreactivity.
We acknowledge that ab3558 has been discontinued by the vendor; however, discontinuation alone is not evidence of non-specificity. Several studies have used this antibody to detect CB1 receptor expression, including recent publications (Shen et al., 2010; PMID: 20680167; Wang et al., 2022; PMID: 35269905; Sevil et al., 2024; PMID: 39758842; Köse et al., 2024; PMID: 39354544). Given the 10-day revision window, it is not feasible to obtain and validate an alternative antibody and repeat these experiments. Instead of re-performing this component with a different antibody, we strengthened validation and presentation of the existing dataset as follows:
- Negative controls: Secondary-only and host-matched rabbit IgG isotype controls were processed in parallel and imaged with identical acquisition parameters to quantify background; these data are now included. Please see attached supplemental material.
- High-magnification imaging: As noted in Section 4.4.1, tile-scan images were acquired on a Zeiss LSM 880 confocal microscope (1024 × 1024 pixels; Plan-Apochromat 63×/1.4 Oil DIC M27 objective). We now added high-magnification insets in Fig. 1E–G from the boxed regions in panel C (AP, NTS, DMNX).
- Re-analysis: CB1 signal was quantified as mean gray value per µm² within predefined ROIs.
- c-fos immunoreactivity seems to be highest outside the areas of interest. Why? Do those cells carry CB1? Why didn’t the authors co-stain with CB1?
The c-fos panels are tile scans that necessarily include tissue adjacent to the dorsal vagal complex (DVC). However, our quantification was restricted to the prespecified emetic nuclei—AP, NTS, and DMNX. The DVC contains not only neurons that express CB1 receptors but also other neuronal types (e.g., glutamatergic and GABAergic), and its nuclei have complex connections with other brain regions, such as the hypothalamus, amygdala, cortex, and so on. In this study, drug administration was systemic (via the circulation) rather than a local injection into the DVC. The goal of this study was therefore to evaluate the overall activation of the DVC after SR141716A administration rather than cellular co-localization. Functionally, SR141716A-evoked emesis is multideterminant (involving cAMP/PKA, ERK1/2, PI3K/Akt/GSK-3, PLC/PKC, CaMKII, and the classical 5-HT3 and D2/3 receptors), so co-localizing c-fos with CB1 would not, by itself, distinguish activation of CB1-expressing neurons from downstream network activation.
- cAMP levels: A positive control would have helped to put these values into perspective. The authors mentioned that rolipram has a similar effect. Why not use this as positive control?
We thank the reviewer for this insightful suggestion. In the present study, we observed a significant increase in cAMP levels following SR141716A administration, consistent with its established role as a CB₁ inverse agonist that disinhibits adenylate cyclase activity. Although rolipram, a phosphodiesterase-4 inhibitor, is known to elevate intracellular cAMP, we did not include it as a positive control here because we have previously reported that rolipram significantly increases brainstem cAMP in the least shrew (Alkam et al., 2014-PMID: 24513510), which supports our present findings. Nevertheless, we agree that such a control would strengthen the interpretation, and we will incorporate rolipram or other reference modulators in future studies to provide a clearer comparative context for cAMP signaling.
- The study overwhelmingly shows descriptive data without explaining the precise molecular interactions between SR141716A and intracellular effectors. The complexity of GPCR biased signaling and the effects of inverse agonism by rimonabant is acknowledged but insufficiently explored, missing an opportunity to provide insight into why SR141716A acts as an inverse agonist triggering emesis while agonists generally confer antiemetic effects. Are these pathways that were investigated all directly linked to CB1 inverse agonism? Would the compounds used also inhibit vomiting induced by a different mechanism? This would be interesting, because then the observed effects would not be solely attributable to CB1 inhibition. This also leads me to another question: Is the activation of the described intracellular pathways by rimonabant a CB1 specific mechanism or is this a general activation scheme common to all emetic compounds?
In this study, we have demonstrated that SR141716A (20 mg/kg, i.p.) produced a time-dependent increase in brainstem cAMP (↑ at 15 and 30 min), a peak in PKA (Thr197) phosphorylation at 30 min, and rapid phosphorylation of ERK1/2, Akt, GSK-3α/β, and PKCα/βII (peaking at ~15 min and persisting for some nodes). These biochemical changes coincided with the observed emesis and DVC activation (↑c-fos; ↑p-ERK1/2), establishing temporal linkage between receptor perturbation and effector pathways. Functionally, pharmacologic inhibition of these emetic signals (PKA, ERK1/2, GSK-3, PI3K, PLC, PKC, CaMKII; membrane Ca²⁺ entry via LTCC and SOCE; intracellular Ca²⁺ release via IP₃R and RyR) significantly reduced SR141716A-evoked vomiting in a dose-dependent manner, indicating these pathways are necessary for the behavioral output under our conditions.
CB₁ receptor predominantly couples to Gi/o to suppress adenylyl cyclase. The observed ↑cAMP/↑PKA after SR141716A is the expected direct consequence of inverse agonism (relief of constitutive Gi/o tone). Also, we found no acute change in CB1 protein density in DVC after SR141716A, arguing that the signaling pattern reflects a change in receptor signaling state rather than receptor density. Engagement of ERK, PI3K/Akt/GSK-3, PLC/PKC, CaMKII, and Ca²⁺ mobilization (LTCC/SOCE and IP₃R/RyR) is consistent with well-known cross-talk downstream of cAMP/PKA in emetic nuclei. In short, cAMP/PKA is a direct readout of CB1 inverse agonism, and the other activated intracellular emetic signals represent downstream activation within the emetic network. Our data show that blocking any of these signals attenuates emesis, so they are causally involved even if not all are “direct” CB1 outputs in every neuron.
In our previous studies, we have demonstrated that the tested compounds used in this study also inhibited shrews vomiting induced by diverse emetogens. In this study, we show that classical antiemetics acting at 5-HT₃ (palonosetron) and D₂/₃ (sulpiride) receptors attenuate SR141716A-evoked vomiting, and that Ca²⁺-channel blockers (LTCC, SOCE, IP₃R, RyR) and kinase inhibitors (PKA, ERK, GSK-3, PI3K, PLC, PKC, CaMKII) are effective against SR141716A-evoked vomiting. Because several of these drug classes are known to act across multiple emetic triggers, their efficacy here indicates that SR141716A engages shared emetic machinery; i.e., the downstream effects are not solely attributable to CB1 blockade per se, but though CB1 inverse agonism is the initiating event in our paradigm.
Our data indicate that SR141716A-induced vomiting is initiated by a CB₁ receptor–specific mechanism. Specifically: (1) Upstream (CB₁-specific): SR141716A’s inverse agonism at CB₁ relieves Gi/o-mediated restraint on adenylyl cyclase, increasing cAMP/PKA and initiating the intracellular cascade that culminates in emesis; (2) Downstream (general/convergent): the execution layer—ERK, PI3K/Akt–GSK-3, PLC/PKC, CaMKII, and coordinated Ca²⁺ entry/release—constitutes a general emetic program engaged by multiple triggers; accordingly, blocking these signals inhibits SR141716A-evoked vomiting.
In summary, SR141716A triggers emesis via CB1 inverse agonism (CB1-specific initiation), but the response is implemented by shared intracellular effectors (general convergence). Our time-course biochemistry plus pathway-targeted inhibition convert the findings from descriptive to mechanistically causal for the emetic phenotype studied here.
- Although multiple pathways are implicated, the study does not sufficiently address alternative or compensatory signaling cascades that may contribute to the emetic response. For example, the lack of increase in endogenous serotonin (5-HT) and substance P (SP) tissue levels contradicts the findings that serotonin and dopamine receptor antagonists are efficacious in the described set-up. This discrepancy merits closer examination, possibly involving release kinetics, metabolism, or receptor sensitivity. Same for p-ERK: ERK would also be activated by CB1 agonists, which are usually anti-emetic. Please elaborate. Similarly, cannabis hyperemesis syndrome is caused by CB1 agonists. This is one major side effect in some patients and since it can be attenuated by TRPV1 agonists, this should be discussed here as well.
1) We agree that although SR141716A treatment did not increase tissue levels of serotonin (5-HT) or substance P (SP) in the dorsal vagal complex, the efficacy of 5-HT₃ and D₂ receptor antagonists in attenuating SR141716A-induced emesis suggests that receptor-level sensitivity, release dynamics, or neurotransmitter turnover may play a more critical role than static tissue levels.
We appreciate the referee’s suggestion, and in our re-revised manuscript we have added the following discussion in the manuscript, “iv) the release dynamics, or receptor-level sensitivity may also play a more critical role than static tissue levels,” Please see lines 846-847.
2) p-ERK1/2 functions as a convergent integrator of multiple inputs; therefore, the behavioral outcome depends on the net effect at the circuit level. CB1 agonists generally reduce presynaptic glutamate/5-HT/GABA release (Gi/o tone), producing an overall antiemetic network effect, even if ERK1/2 is transiently activated (e.g., via β-arrestin pathways). SR141716A, as a CB1 inverse agonist, disinhibits AC→cAMP/PKA, permitting broader Ca²⁺ mobilization (LTCC/SOCE/IP₃R/RyR) and ERK/PI3K–Akt–GSK-3/PLC–PKC/CaMKII engagement that together promote emesis.
3) Per referee 2’s suggestion, we have added the following discussion in the manuscript, “Cannabis hyperemesis syndrome (CHS) occurs following chronic administration of large doses of CB₁ agonists, further illustrates the complexity of endocannabinoid signaling. Interestingly, TRPV1R agonists such as capsaicin provide relief in CHS patients, supporting the idea that TRPV1R-CB₁ crosstalk and compensatory adaptations of intracellular pathways are critical determinants of the emetic phenotype [128, 129]. In fact, we have demonstrated that a combination of CB1- and TRPV1 receptor agonists have the capacity to completely abolish cisplatin-evoked emesis at doses that are ineffective when used individually [121].” Please see lines 776-783.
- The peripheral involvement has only been shown indirectly through stainings in the ENS. However, whether those changes are emetic has not been shown. This can be shown by vagotomy but all evidence shown here is indirect.
The DVC is the canonical integrator of vagal afferents driving emesis, so concurrent ENS and DVC activation is expected if the brain–gut axis is engaged. ENS activity markers rise in the same time window as vomiting and DVC (AP/NTS/DMNX) activation, compatible with a gut → vagus → DVC pathway. Our ENS staining together with DVC activation supports involvement of the brain–gut axis during SR141716A-evoked emesis. Accordingly, in future studies we will perform subdiaphragmatic vagotomy or reversible perivagal blockade to determine the involvement of the entire brain–gut axis in the induced emetic response. We thank you for your suggestion!
- Translational and Clinical Implications are vague: The study's focus on mechanistic dissection in the least shrew model does not adequately bridge to clinical implications for cannabinoid-based antiemetic therapy or adverse emetic effects associated with CB1 inhibitors. More explicit discussion on how these findings may inform drug design or treatment of nausea and vomiting in patients would increase relevance.
While this study was conducted in the least shrew, a well-validated model for emesis research, the mechanistic findings have potential translational relevance for human conditions. The ability of SR141716A to trigger vomiting through CB₁ inverse agonism underscores the critical role of endocannabinoid signaling in emetic regulation and helps explain why SR141716A was associated with adverse gastrointestinal and neuropsychiatric effects in the clinic. Furthermore, our results suggest that targeting downstream effectors such as ERK1/2, PI3K/Akt, and calcium signaling may provide alternative strategies for antiemetic drug development beyond conventional 5-HT₃ and dopamine receptor antagonists. The observed overlap with signaling pathways implicated in CHS also highlights the translational relevance of our findings, as they may help inform management strategies for patients suffering from this increasingly recognized condition. Overall, while cannabis cessation remains the definitive treatment for CHS, our data suggest that pharmacological modulation of convergent signaling cascades may hold promise as adjunctive therapies. We cannot provide further details since we were asked to reduce the length of the original manuscript.
Referee 2
Comments and Suggestions for Authors
This study by Sun et al. presents an extensive investigation into the emetic mechanisms triggered by the cannabinoid CB1 receptor inverse agonist SR141716A in least shrews, revealing complex intracellular signaling pathways involved in vomiting. However, the study, while thorough, leans heavily on descriptive experimental data and would benefit from deeper mechanistic insights.
The authors systematically demonstrate that Rimonabant induces vomiting via both central and peripheral mechanisms, chiefly the dorsal vagal complex (DVC) in the brainstem and the enteric nervous system in the jejunum, with evidence from c-fos and phospho-ERK1/2 immunfluorescence, among others. They provide a link between brainstem cAMP levels and sequential phosphorylation of key kinases such as PKA, Akt, GSK-3α/β, ERK1/2, and PKCαβII to the emetic response. The broad-spectrum pharmacological blockade studies targeting these kinases and various calcium channels strongly support their involvement in SR141716A-evoked vomiting.
Critical Issues:
- Antibody Validity for CB1 Detection and general antibody-related issues: A key weakness undermining the interpretation of CB1 receptor distribution is the use of a commercial anti-CB1 antibody (ab3558, Abcam) which has been withdrawn from the market by the company. This raises questions regarding the specificity and reliability of CB1 receptor immunostaining data presented. In fact, previous studies (Histochem Cell Biol. 2021 Aug 27;156(5):479–502, J Neurosci Methods. 2008 Jun 15;171(1):78-8) investigated CB1 antibody specificity and found that many CB1 antibodies do not specifically detect the protein. Staining specificity of the antibody used should be either verified by using another CB1 antibody, another technique (Western Blot) or by the use of knockout animals (in this case not possible). In Fig. 1 CB1 staining is shown. However, staining doesn’t look specific since I only see a blurry reddish image. Maybe supplying a magnification could shed some light whether the antibody stains specifically. A quantification of only those cells that are labelled positive is not a very good method since this leaves out cells with low level expression. Rather, staining intensity should be considered for quantification. In addition, I would like to see an isotype staining, especially as a control for CB1.
We agree that CB1 antibodies vary in performance. In our study, CB1 immunostaining was used only to provide regional context (AP/NTS/DMNX mapping) and to note that acute SR141716A did not alter regional CB1 density; our key conclusions (↑cAMP/PKA, engagement of ERK/PI3K–Akt–GSK-3/PLC–PKC/CaMKII, Ca²⁺ mobilization, and inhibitor sensitivity of emesis) do not depend on subtle intensity gradients of CB1 immunoreactivity.
We acknowledge that ab3558 has been discontinued by the vendor; however, discontinuation alone is not evidence of non-specificity. Several studies have used this antibody to detect CB1 receptor expression, including recent publications (Shen et al., 2010; PMID: 20680167; Wang et al., 2022; PMID: 35269905; Sevil et al., 2024; PMID: 39758842; Köse et al., 2024; PMID: 39354544). Given the 10-day revision window, it is not feasible to obtain and validate an alternative antibody and repeat these experiments. Instead of re-performing this component with a different antibody, we strengthened validation and presentation of the existing dataset as follows:
- Negative controls: Secondary-only and host-matched rabbit IgG isotype controls were processed in parallel and imaged with identical acquisition parameters to quantify background; these data are now included. Please see attached supplemental material.
- High-magnification imaging: As noted in Section 4.4.1, tile-scan images were acquired on a Zeiss LSM 880 confocal microscope (1024 × 1024 pixels; Plan-Apochromat 63×/1.4 Oil DIC M27 objective). We now added high-magnification insets in Fig. 1E–G from the boxed regions in panel C (AP, NTS, DMNX).
- Re-analysis: CB1 signal was quantified as mean gray value per µm² within predefined ROIs.
- c-fos immunoreactivity seems to be highest outside the areas of interest. Why? Do those cells carry CB1? Why didn’t the authors co-stain with CB1?
The c-fos panels are tile scans that necessarily include tissue adjacent to the dorsal vagal complex (DVC). However, our quantification was restricted to the prespecified emetic nuclei—AP, NTS, and DMNX. The DVC contains not only neurons that express CB1 receptors but also other neuronal types (e.g., glutamatergic and GABAergic), and its nuclei have complex connections with other brain regions, such as the hypothalamus, amygdala, cortex, and so on. In this study, drug administration was systemic (via the circulation) rather than a local injection into the DVC. The goal of this study was therefore to evaluate the overall activation of the DVC after SR141716A administration rather than cellular co-localization. Functionally, SR141716A-evoked emesis is multideterminant (involving cAMP/PKA, ERK1/2, PI3K/Akt/GSK-3, PLC/PKC, CaMKII, and the classical 5-HT3 and D2/3 receptors), so co-localizing c-fos with CB1 would not, by itself, distinguish activation of CB1-expressing neurons from downstream network activation.
- cAMP levels: A positive control would have helped to put these values into perspective. The authors mentioned that rolipram has a similar effect. Why not use this as positive control?
We thank the reviewer for this insightful suggestion. In the present study, we observed a significant increase in cAMP levels following SR141716A administration, consistent with its established role as a CB₁ inverse agonist that disinhibits adenylate cyclase activity. Although rolipram, a phosphodiesterase-4 inhibitor, is known to elevate intracellular cAMP, we did not include it as a positive control here because we have previously reported that rolipram significantly increases brainstem cAMP in the least shrew (Alkam et al., 2014-PMID: 24513510), which supports our present findings. Nevertheless, we agree that such a control would strengthen the interpretation, and we will incorporate rolipram or other reference modulators in future studies to provide a clearer comparative context for cAMP signaling.
- The study overwhelmingly shows descriptive data without explaining the precise molecular interactions between SR141716A and intracellular effectors. The complexity of GPCR biased signaling and the effects of inverse agonism by rimonabant is acknowledged but insufficiently explored, missing an opportunity to provide insight into why SR141716A acts as an inverse agonist triggering emesis while agonists generally confer antiemetic effects. Are these pathways that were investigated all directly linked to CB1 inverse agonism? Would the compounds used also inhibit vomiting induced by a different mechanism? This would be interesting, because then the observed effects would not be solely attributable to CB1 inhibition. This also leads me to another question: Is the activation of the described intracellular pathways by rimonabant a CB1 specific mechanism or is this a general activation scheme common to all emetic compounds?
In this study, we have demonstrated that SR141716A (20 mg/kg, i.p.) produced a time-dependent increase in brainstem cAMP (↑ at 15 and 30 min), a peak in PKA (Thr197) phosphorylation at 30 min, and rapid phosphorylation of ERK1/2, Akt, GSK-3α/β, and PKCα/βII (peaking at ~15 min and persisting for some nodes). These biochemical changes coincided with the observed emesis and DVC activation (↑c-fos; ↑p-ERK1/2), establishing temporal linkage between receptor perturbation and effector pathways. Functionally, pharmacologic inhibition of these emetic signals (PKA, ERK1/2, GSK-3, PI3K, PLC, PKC, CaMKII; membrane Ca²⁺ entry via LTCC and SOCE; intracellular Ca²⁺ release via IP₃R and RyR) significantly reduced SR141716A-evoked vomiting in a dose-dependent manner, indicating these pathways are necessary for the behavioral output under our conditions.
CB₁ receptor predominantly couples to Gi/o to suppress adenylyl cyclase. The observed ↑cAMP/↑PKA after SR141716A is the expected direct consequence of inverse agonism (relief of constitutive Gi/o tone). Also, we found no acute change in CB1 protein density in DVC after SR141716A, arguing that the signaling pattern reflects a change in receptor signaling state rather than receptor density. Engagement of ERK, PI3K/Akt/GSK-3, PLC/PKC, CaMKII, and Ca²⁺ mobilization (LTCC/SOCE and IP₃R/RyR) is consistent with well-known cross-talk downstream of cAMP/PKA in emetic nuclei. In short, cAMP/PKA is a direct readout of CB1 inverse agonism, and the other activated intracellular emetic signals represent downstream activation within the emetic network. Our data show that blocking any of these signals attenuates emesis, so they are causally involved even if not all are “direct” CB1 outputs in every neuron.
In our previous studies, we have demonstrated that the tested compounds used in this study also inhibited shrews vomiting induced by diverse emetogens. In this study, we show that classical antiemetics acting at 5-HT₃ (palonosetron) and D₂/₃ (sulpiride) receptors attenuate SR141716A-evoked vomiting, and that Ca²⁺-channel blockers (LTCC, SOCE, IP₃R, RyR) and kinase inhibitors (PKA, ERK, GSK-3, PI3K, PLC, PKC, CaMKII) are effective against SR141716A-evoked vomiting. Because several of these drug classes are known to act across multiple emetic triggers, their efficacy here indicates that SR141716A engages shared emetic machinery; i.e., the downstream effects are not solely attributable to CB1 blockade per se, but though CB1 inverse agonism is the initiating event in our paradigm.
Our data indicate that SR141716A-induced vomiting is initiated by a CB₁ receptor–specific mechanism. Specifically: (1) Upstream (CB₁-specific): SR141716A’s inverse agonism at CB₁ relieves Gi/o-mediated restraint on adenylyl cyclase, increasing cAMP/PKA and initiating the intracellular cascade that culminates in emesis; (2) Downstream (general/convergent): the execution layer—ERK, PI3K/Akt–GSK-3, PLC/PKC, CaMKII, and coordinated Ca²⁺ entry/release—constitutes a general emetic program engaged by multiple triggers; accordingly, blocking these signals inhibits SR141716A-evoked vomiting.
In summary, SR141716A triggers emesis via CB1 inverse agonism (CB1-specific initiation), but the response is implemented by shared intracellular effectors (general convergence). Our time-course biochemistry plus pathway-targeted inhibition convert the findings from descriptive to mechanistically causal for the emetic phenotype studied here.
- Although multiple pathways are implicated, the study does not sufficiently address alternative or compensatory signaling cascades that may contribute to the emetic response. For example, the lack of increase in endogenous serotonin (5-HT) and substance P (SP) tissue levels contradicts the findings that serotonin and dopamine receptor antagonists are efficacious in the described set-up. This discrepancy merits closer examination, possibly involving release kinetics, metabolism, or receptor sensitivity. Same for p-ERK: ERK would also be activated by CB1 agonists, which are usually anti-emetic. Please elaborate. Similarly, cannabis hyperemesis syndrome is caused by CB1 agonists. This is one major side effect in some patients and since it can be attenuated by TRPV1 agonists, this should be discussed here as well.
1) We agree that although SR141716A treatment did not increase tissue levels of serotonin (5-HT) or substance P (SP) in the dorsal vagal complex, the efficacy of 5-HT₃ and D₂ receptor antagonists in attenuating SR141716A-induced emesis suggests that receptor-level sensitivity, release dynamics, or neurotransmitter turnover may play a more critical role than static tissue levels.
We appreciate the referee’s suggestion, and in our re-revised manuscript we have added the following discussion in the manuscript, “iv) the release dynamics, or receptor-level sensitivity may also play a more critical role than static tissue levels,” Please see lines 846-847.
2) p-ERK1/2 functions as a convergent integrator of multiple inputs; therefore, the behavioral outcome depends on the net effect at the circuit level. CB1 agonists generally reduce presynaptic glutamate/5-HT/GABA release (Gi/o tone), producing an overall antiemetic network effect, even if ERK1/2 is transiently activated (e.g., via β-arrestin pathways). SR141716A, as a CB1 inverse agonist, disinhibits AC→cAMP/PKA, permitting broader Ca²⁺ mobilization (LTCC/SOCE/IP₃R/RyR) and ERK/PI3K–Akt–GSK-3/PLC–PKC/CaMKII engagement that together promote emesis.
3) Per referee 2’s suggestion, we have added the following discussion in the manuscript, “Cannabis hyperemesis syndrome (CHS) occurs following chronic administration of large doses of CB₁ agonists, further illustrates the complexity of endocannabinoid signaling. Interestingly, TRPV1R agonists such as capsaicin provide relief in CHS patients, supporting the idea that TRPV1R-CB₁ crosstalk and compensatory adaptations of intracellular pathways are critical determinants of the emetic phenotype [128, 129]. In fact, we have demonstrated that a combination of CB1- and TRPV1 receptor agonists have the capacity to completely abolish cisplatin-evoked emesis at doses that are ineffective when used individually [121].” Please see lines 776-783.
- The peripheral involvement has only been shown indirectly through stainings in the ENS. However, whether those changes are emetic has not been shown. This can be shown by vagotomy but all evidence shown here is indirect.
The DVC is the canonical integrator of vagal afferents driving emesis, so concurrent ENS and DVC activation is expected if the brain–gut axis is engaged. ENS activity markers rise in the same time window as vomiting and DVC (AP/NTS/DMNX) activation, compatible with a gut → vagus → DVC pathway. Our ENS staining together with DVC activation supports involvement of the brain–gut axis during SR141716A-evoked emesis. Accordingly, in future studies we will perform subdiaphragmatic vagotomy or reversible perivagal blockade to determine the involvement of the entire brain–gut axis in the induced emetic response. We thank you for your suggestion!
- Translational and Clinical Implications are vague: The study's focus on mechanistic dissection in the least shrew model does not adequately bridge to clinical implications for cannabinoid-based antiemetic therapy or adverse emetic effects associated with CB1 inhibitors. More explicit discussion on how these findings may inform drug design or treatment of nausea and vomiting in patients would increase relevance.
While this study was conducted in the least shrew, a well-validated model for emesis research, the mechanistic findings have potential translational relevance for human conditions. The ability of SR141716A to trigger vomiting through CB₁ inverse agonism underscores the critical role of endocannabinoid signaling in emetic regulation and helps explain why SR141716A was associated with adverse gastrointestinal and neuropsychiatric effects in the clinic. Furthermore, our results suggest that targeting downstream effectors such as ERK1/2, PI3K/Akt, and calcium signaling may provide alternative strategies for antiemetic drug development beyond conventional 5-HT₃ and dopamine receptor antagonists. The observed overlap with signaling pathways implicated in CHS also highlights the translational relevance of our findings, as they may help inform management strategies for patients suffering from this increasingly recognized condition. Overall, while cannabis cessation remains the definitive treatment for CHS, our data suggest that pharmacological modulation of convergent signaling cascades may hold promise as adjunctive therapies. We cannot provide further details since we were asked to reduce the length of the original manuscript.
Referee 2
Comments and Suggestions for Authors
This study by Sun et al. presents an extensive investigation into the emetic mechanisms triggered by the cannabinoid CB1 receptor inverse agonist SR141716A in least shrews, revealing complex intracellular signaling pathways involved in vomiting. However, the study, while thorough, leans heavily on descriptive experimental data and would benefit from deeper mechanistic insights.
The authors systematically demonstrate that Rimonabant induces vomiting via both central and peripheral mechanisms, chiefly the dorsal vagal complex (DVC) in the brainstem and the enteric nervous system in the jejunum, with evidence from c-fos and phospho-ERK1/2 immunfluorescence, among others. They provide a link between brainstem cAMP levels and sequential phosphorylation of key kinases such as PKA, Akt, GSK-3α/β, ERK1/2, and PKCαβII to the emetic response. The broad-spectrum pharmacological blockade studies targeting these kinases and various calcium channels strongly support their involvement in SR141716A-evoked vomiting.
Critical Issues:
- Antibody Validity for CB1 Detection and general antibody-related issues: A key weakness undermining the interpretation of CB1 receptor distribution is the use of a commercial anti-CB1 antibody (ab3558, Abcam) which has been withdrawn from the market by the company. This raises questions regarding the specificity and reliability of CB1 receptor immunostaining data presented. In fact, previous studies (Histochem Cell Biol. 2021 Aug 27;156(5):479–502, J Neurosci Methods. 2008 Jun 15;171(1):78-8) investigated CB1 antibody specificity and found that many CB1 antibodies do not specifically detect the protein. Staining specificity of the antibody used should be either verified by using another CB1 antibody, another technique (Western Blot) or by the use of knockout animals (in this case not possible). In Fig. 1 CB1 staining is shown. However, staining doesn’t look specific since I only see a blurry reddish image. Maybe supplying a magnification could shed some light whether the antibody stains specifically. A quantification of only those cells that are labelled positive is not a very good method since this leaves out cells with low level expression. Rather, staining intensity should be considered for quantification. In addition, I would like to see an isotype staining, especially as a control for CB1.
We agree that CB1 antibodies vary in performance. In our study, CB1 immunostaining was used only to provide regional context (AP/NTS/DMNX mapping) and to note that acute SR141716A did not alter regional CB1 density; our key conclusions (↑cAMP/PKA, engagement of ERK/PI3K–Akt–GSK-3/PLC–PKC/CaMKII, Ca²⁺ mobilization, and inhibitor sensitivity of emesis) do not depend on subtle intensity gradients of CB1 immunoreactivity.
We acknowledge that ab3558 has been discontinued by the vendor; however, discontinuation alone is not evidence of non-specificity. Several studies have used this antibody to detect CB1 receptor expression, including recent publications (Shen et al., 2010; PMID: 20680167; Wang et al., 2022; PMID: 35269905; Sevil et al., 2024; PMID: 39758842; Köse et al., 2024; PMID: 39354544). Given the 10-day revision window, it is not feasible to obtain and validate an alternative antibody and repeat these experiments. Instead of re-performing this component with a different antibody, we strengthened validation and presentation of the existing dataset as follows:
- Negative controls: Secondary-only and host-matched rabbit IgG isotype controls were processed in parallel and imaged with identical acquisition parameters to quantify background; these data are now included. Please see attached supplemental material.
- High-magnification imaging: As noted in Section 4.4.1, tile-scan images were acquired on a Zeiss LSM 880 confocal microscope (1024 × 1024 pixels; Plan-Apochromat 63×/1.4 Oil DIC M27 objective). We now added high-magnification insets in Fig. 1E–G from the boxed regions in panel C (AP, NTS, DMNX).
- Re-analysis: CB1 signal was quantified as mean gray value per µm² within predefined ROIs.
- c-fos immunoreactivity seems to be highest outside the areas of interest. Why? Do those cells carry CB1? Why didn’t the authors co-stain with CB1?
The c-fos panels are tile scans that necessarily include tissue adjacent to the dorsal vagal complex (DVC). However, our quantification was restricted to the prespecified emetic nuclei—AP, NTS, and DMNX. The DVC contains not only neurons that express CB1 receptors but also other neuronal types (e.g., glutamatergic and GABAergic), and its nuclei have complex connections with other brain regions, such as the hypothalamus, amygdala, cortex, and so on. In this study, drug administration was systemic (via the circulation) rather than a local injection into the DVC. The goal of this study was therefore to evaluate the overall activation of the DVC after SR141716A administration rather than cellular co-localization. Functionally, SR141716A-evoked emesis is multideterminant (involving cAMP/PKA, ERK1/2, PI3K/Akt/GSK-3, PLC/PKC, CaMKII, and the classical 5-HT3 and D2/3 receptors), so co-localizing c-fos with CB1 would not, by itself, distinguish activation of CB1-expressing neurons from downstream network activation.
- cAMP levels: A positive control would have helped to put these values into perspective. The authors mentioned that rolipram has a similar effect. Why not use this as positive control?
We thank the reviewer for this insightful suggestion. In the present study, we observed a significant increase in cAMP levels following SR141716A administration, consistent with its established role as a CB₁ inverse agonist that disinhibits adenylate cyclase activity. Although rolipram, a phosphodiesterase-4 inhibitor, is known to elevate intracellular cAMP, we did not include it as a positive control here because we have previously reported that rolipram significantly increases brainstem cAMP in the least shrew (Alkam et al., 2014-PMID: 24513510), which supports our present findings. Nevertheless, we agree that such a control would strengthen the interpretation, and we will incorporate rolipram or other reference modulators in future studies to provide a clearer comparative context for cAMP signaling.
- The study overwhelmingly shows descriptive data without explaining the precise molecular interactions between SR141716A and intracellular effectors. The complexity of GPCR biased signaling and the effects of inverse agonism by rimonabant is acknowledged but insufficiently explored, missing an opportunity to provide insight into why SR141716A acts as an inverse agonist triggering emesis while agonists generally confer antiemetic effects. Are these pathways that were investigated all directly linked to CB1 inverse agonism? Would the compounds used also inhibit vomiting induced by a different mechanism? This would be interesting, because then the observed effects would not be solely attributable to CB1 inhibition. This also leads me to another question: Is the activation of the described intracellular pathways by rimonabant a CB1 specific mechanism or is this a general activation scheme common to all emetic compounds?
In this study, we have demonstrated that SR141716A (20 mg/kg, i.p.) produced a time-dependent increase in brainstem cAMP (↑ at 15 and 30 min), a peak in PKA (Thr197) phosphorylation at 30 min, and rapid phosphorylation of ERK1/2, Akt, GSK-3α/β, and PKCα/βII (peaking at ~15 min and persisting for some nodes). These biochemical changes coincided with the observed emesis and DVC activation (↑c-fos; ↑p-ERK1/2), establishing temporal linkage between receptor perturbation and effector pathways. Functionally, pharmacologic inhibition of these emetic signals (PKA, ERK1/2, GSK-3, PI3K, PLC, PKC, CaMKII; membrane Ca²⁺ entry via LTCC and SOCE; intracellular Ca²⁺ release via IP₃R and RyR) significantly reduced SR141716A-evoked vomiting in a dose-dependent manner, indicating these pathways are necessary for the behavioral output under our conditions.
CB₁ receptor predominantly couples to Gi/o to suppress adenylyl cyclase. The observed ↑cAMP/↑PKA after SR141716A is the expected direct consequence of inverse agonism (relief of constitutive Gi/o tone). Also, we found no acute change in CB1 protein density in DVC after SR141716A, arguing that the signaling pattern reflects a change in receptor signaling state rather than receptor density. Engagement of ERK, PI3K/Akt/GSK-3, PLC/PKC, CaMKII, and Ca²⁺ mobilization (LTCC/SOCE and IP₃R/RyR) is consistent with well-known cross-talk downstream of cAMP/PKA in emetic nuclei. In short, cAMP/PKA is a direct readout of CB1 inverse agonism, and the other activated intracellular emetic signals represent downstream activation within the emetic network. Our data show that blocking any of these signals attenuates emesis, so they are causally involved even if not all are “direct” CB1 outputs in every neuron.
In our previous studies, we have demonstrated that the tested compounds used in this study also inhibited shrews vomiting induced by diverse emetogens. In this study, we show that classical antiemetics acting at 5-HT₃ (palonosetron) and D₂/₃ (sulpiride) receptors attenuate SR141716A-evoked vomiting, and that Ca²⁺-channel blockers (LTCC, SOCE, IP₃R, RyR) and kinase inhibitors (PKA, ERK, GSK-3, PI3K, PLC, PKC, CaMKII) are effective against SR141716A-evoked vomiting. Because several of these drug classes are known to act across multiple emetic triggers, their efficacy here indicates that SR141716A engages shared emetic machinery; i.e., the downstream effects are not solely attributable to CB1 blockade per se, but though CB1 inverse agonism is the initiating event in our paradigm.
Our data indicate that SR141716A-induced vomiting is initiated by a CB₁ receptor–specific mechanism. Specifically: (1) Upstream (CB₁-specific): SR141716A’s inverse agonism at CB₁ relieves Gi/o-mediated restraint on adenylyl cyclase, increasing cAMP/PKA and initiating the intracellular cascade that culminates in emesis; (2) Downstream (general/convergent): the execution layer—ERK, PI3K/Akt–GSK-3, PLC/PKC, CaMKII, and coordinated Ca²⁺ entry/release—constitutes a general emetic program engaged by multiple triggers; accordingly, blocking these signals inhibits SR141716A-evoked vomiting.
In summary, SR141716A triggers emesis via CB1 inverse agonism (CB1-specific initiation), but the response is implemented by shared intracellular effectors (general convergence). Our time-course biochemistry plus pathway-targeted inhibition convert the findings from descriptive to mechanistically causal for the emetic phenotype studied here.
- Although multiple pathways are implicated, the study does not sufficiently address alternative or compensatory signaling cascades that may contribute to the emetic response. For example, the lack of increase in endogenous serotonin (5-HT) and substance P (SP) tissue levels contradicts the findings that serotonin and dopamine receptor antagonists are efficacious in the described set-up. This discrepancy merits closer examination, possibly involving release kinetics, metabolism, or receptor sensitivity. Same for p-ERK: ERK would also be activated by CB1 agonists, which are usually anti-emetic. Please elaborate. Similarly, cannabis hyperemesis syndrome is caused by CB1 agonists. This is one major side effect in some patients and since it can be attenuated by TRPV1 agonists, this should be discussed here as well.
1) We agree that although SR141716A treatment did not increase tissue levels of serotonin (5-HT) or substance P (SP) in the dorsal vagal complex, the efficacy of 5-HT₃ and D₂ receptor antagonists in attenuating SR141716A-induced emesis suggests that receptor-level sensitivity, release dynamics, or neurotransmitter turnover may play a more critical role than static tissue levels.
We appreciate the referee’s suggestion, and in our re-revised manuscript we have added the following discussion in the manuscript, “iv) the release dynamics, or receptor-level sensitivity may also play a more critical role than static tissue levels,” Please see lines 846-847.
2) p-ERK1/2 functions as a convergent integrator of multiple inputs; therefore, the behavioral outcome depends on the net effect at the circuit level. CB1 agonists generally reduce presynaptic glutamate/5-HT/GABA release (Gi/o tone), producing an overall antiemetic network effect, even if ERK1/2 is transiently activated (e.g., via β-arrestin pathways). SR141716A, as a CB1 inverse agonist, disinhibits AC→cAMP/PKA, permitting broader Ca²⁺ mobilization (LTCC/SOCE/IP₃R/RyR) and ERK/PI3K–Akt–GSK-3/PLC–PKC/CaMKII engagement that together promote emesis.
3) Per referee 2’s suggestion, we have added the following discussion in the manuscript, “Cannabis hyperemesis syndrome (CHS) occurs following chronic administration of large doses of CB₁ agonists, further illustrates the complexity of endocannabinoid signaling. Interestingly, TRPV1R agonists such as capsaicin provide relief in CHS patients, supporting the idea that TRPV1R-CB₁ crosstalk and compensatory adaptations of intracellular pathways are critical determinants of the emetic phenotype [128, 129]. In fact, we have demonstrated that a combination of CB1- and TRPV1 receptor agonists have the capacity to completely abolish cisplatin-evoked emesis at doses that are ineffective when used individually [121].” Please see lines 776-783.
- The peripheral involvement has only been shown indirectly through stainings in the ENS. However, whether those changes are emetic has not been shown. This can be shown by vagotomy but all evidence shown here is indirect.
The DVC is the canonical integrator of vagal afferents driving emesis, so concurrent ENS and DVC activation is expected if the brain–gut axis is engaged. ENS activity markers rise in the same time window as vomiting and DVC (AP/NTS/DMNX) activation, compatible with a gut → vagus → DVC pathway. Our ENS staining together with DVC activation supports involvement of the brain–gut axis during SR141716A-evoked emesis. Accordingly, in future studies we will perform subdiaphragmatic vagotomy or reversible perivagal blockade to determine the involvement of the entire brain–gut axis in the induced emetic response. We thank you for your suggestion!
- Translational and Clinical Implications are vague: The study's focus on mechanistic dissection in the least shrew model does not adequately bridge to clinical implications for cannabinoid-based antiemetic therapy or adverse emetic effects associated with CB1 inhibitors. More explicit discussion on how these findings may inform drug design or treatment of nausea and vomiting in patients would increase relevance.
While this study was conducted in the least shrew, a well-validated model for emesis research, the mechanistic findings have potential translational relevance for human conditions. The ability of SR141716A to trigger vomiting through CB₁ inverse agonism underscores the critical role of endocannabinoid signaling in emetic regulation and helps explain why SR141716A was associated with adverse gastrointestinal and neuropsychiatric effects in the clinic. Furthermore, our results suggest that targeting downstream effectors such as ERK1/2, PI3K/Akt, and calcium signaling may provide alternative strategies for antiemetic drug development beyond conventional 5-HT₃ and dopamine receptor antagonists. The observed overlap with signaling pathways implicated in CHS also highlights the translational relevance of our findings, as they may help inform management strategies for patients suffering from this increasingly recognized condition. Overall, while cannabis cessation remains the definitive treatment for CHS, our data suggest that pharmacological modulation of convergent signaling cascades may hold promise as adjunctive therapies. We cannot provide further details since we were asked to reduce the length of the original manuscript.
Referee 2
Comments and Suggestions for Authors
This study by Sun et al. presents an extensive investigation into the emetic mechanisms triggered by the cannabinoid CB1 receptor inverse agonist SR141716A in least shrews, revealing complex intracellular signaling pathways involved in vomiting. However, the study, while thorough, leans heavily on descriptive experimental data and would benefit from deeper mechanistic insights.
The authors systematically demonstrate that Rimonabant induces vomiting via both central and peripheral mechanisms, chiefly the dorsal vagal complex (DVC) in the brainstem and the enteric nervous system in the jejunum, with evidence from c-fos and phospho-ERK1/2 immunfluorescence, among others. They provide a link between brainstem cAMP levels and sequential phosphorylation of key kinases such as PKA, Akt, GSK-3α/β, ERK1/2, and PKCαβII to the emetic response. The broad-spectrum pharmacological blockade studies targeting these kinases and various calcium channels strongly support their involvement in SR141716A-evoked vomiting.
Critical Issues:
- Antibody Validity for CB1 Detection and general antibody-related issues: A key weakness undermining the interpretation of CB1 receptor distribution is the use of a commercial anti-CB1 antibody (ab3558, Abcam) which has been withdrawn from the market by the company. This raises questions regarding the specificity and reliability of CB1 receptor immunostaining data presented. In fact, previous studies (Histochem Cell Biol. 2021 Aug 27;156(5):479–502, J Neurosci Methods. 2008 Jun 15;171(1):78-8) investigated CB1 antibody specificity and found that many CB1 antibodies do not specifically detect the protein. Staining specificity of the antibody used should be either verified by using another CB1 antibody, another technique (Western Blot) or by the use of knockout animals (in this case not possible). In Fig. 1 CB1 staining is shown. However, staining doesn’t look specific since I only see a blurry reddish image. Maybe supplying a magnification could shed some light whether the antibody stains specifically. A quantification of only those cells that are labelled positive is not a very good method since this leaves out cells with low level expression. Rather, staining intensity should be considered for quantification. In addition, I would like to see an isotype staining, especially as a control for CB1.
We agree that CB1 antibodies vary in performance. In our study, CB1 immunostaining was used only to provide regional context (AP/NTS/DMNX mapping) and to note that acute SR141716A did not alter regional CB1 density; our key conclusions (↑cAMP/PKA, engagement of ERK/PI3K–Akt–GSK-3/PLC–PKC/CaMKII, Ca²⁺ mobilization, and inhibitor sensitivity of emesis) do not depend on subtle intensity gradients of CB1 immunoreactivity.
We acknowledge that ab3558 has been discontinued by the vendor; however, discontinuation alone is not evidence of non-specificity. Several studies have used this antibody to detect CB1 receptor expression, including recent publications (Shen et al., 2010; PMID: 20680167; Wang et al., 2022; PMID: 35269905; Sevil et al., 2024; PMID: 39758842; Köse et al., 2024; PMID: 39354544). Given the 10-day revision window, it is not feasible to obtain and validate an alternative antibody and repeat these experiments. Instead of re-performing this component with a different antibody, we strengthened validation and presentation of the existing dataset as follows:
- Negative controls: Secondary-only and host-matched rabbit IgG isotype controls were processed in parallel and imaged with identical acquisition parameters to quantify background; these data are now included. Please see attached supplemental material.
- High-magnification imaging: As noted in Section 4.4.1, tile-scan images were acquired on a Zeiss LSM 880 confocal microscope (1024 × 1024 pixels; Plan-Apochromat 63×/1.4 Oil DIC M27 objective). We now added high-magnification insets in Fig. 1E–G from the boxed regions in panel C (AP, NTS, DMNX).
- Re-analysis: CB1 signal was quantified as mean gray value per µm² within predefined ROIs.
- c-fos immunoreactivity seems to be highest outside the areas of interest. Why? Do those cells carry CB1? Why didn’t the authors co-stain with CB1?
The c-fos panels are tile scans that necessarily include tissue adjacent to the dorsal vagal complex (DVC). However, our quantification was restricted to the prespecified emetic nuclei—AP, NTS, and DMNX. The DVC contains not only neurons that express CB1 receptors but also other neuronal types (e.g., glutamatergic and GABAergic), and its nuclei have complex connections with other brain regions, such as the hypothalamus, amygdala, cortex, and so on. In this study, drug administration was systemic (via the circulation) rather than a local injection into the DVC. The goal of this study was therefore to evaluate the overall activation of the DVC after SR141716A administration rather than cellular co-localization. Functionally, SR141716A-evoked emesis is multideterminant (involving cAMP/PKA, ERK1/2, PI3K/Akt/GSK-3, PLC/PKC, CaMKII, and the classical 5-HT3 and D2/3 receptors), so co-localizing c-fos with CB1 would not, by itself, distinguish activation of CB1-expressing neurons from downstream network activation.
- cAMP levels: A positive control would have helped to put these values into perspective. The authors mentioned that rolipram has a similar effect. Why not use this as positive control?
We thank the reviewer for this insightful suggestion. In the present study, we observed a significant increase in cAMP levels following SR141716A administration, consistent with its established role as a CB₁ inverse agonist that disinhibits adenylate cyclase activity. Although rolipram, a phosphodiesterase-4 inhibitor, is known to elevate intracellular cAMP, we did not include it as a positive control here because we have previously reported that rolipram significantly increases brainstem cAMP in the least shrew (Alkam et al., 2014-PMID: 24513510), which supports our present findings. Nevertheless, we agree that such a control would strengthen the interpretation, and we will incorporate rolipram or other reference modulators in future studies to provide a clearer comparative context for cAMP signaling.
- The study overwhelmingly shows descriptive data without explaining the precise molecular interactions between SR141716A and intracellular effectors. The complexity of GPCR biased signaling and the effects of inverse agonism by rimonabant is acknowledged but insufficiently explored, missing an opportunity to provide insight into why SR141716A acts as an inverse agonist triggering emesis while agonists generally confer antiemetic effects. Are these pathways that were investigated all directly linked to CB1 inverse agonism? Would the compounds used also inhibit vomiting induced by a different mechanism? This would be interesting, because then the observed effects would not be solely attributable to CB1 inhibition. This also leads me to another question: Is the activation of the described intracellular pathways by rimonabant a CB1 specific mechanism or is this a general activation scheme common to all emetic compounds?
In this study, we have demonstrated that SR141716A (20 mg/kg, i.p.) produced a time-dependent increase in brainstem cAMP (↑ at 15 and 30 min), a peak in PKA (Thr197) phosphorylation at 30 min, and rapid phosphorylation of ERK1/2, Akt, GSK-3α/β, and PKCα/βII (peaking at ~15 min and persisting for some nodes). These biochemical changes coincided with the observed emesis and DVC activation (↑c-fos; ↑p-ERK1/2), establishing temporal linkage between receptor perturbation and effector pathways. Functionally, pharmacologic inhibition of these emetic signals (PKA, ERK1/2, GSK-3, PI3K, PLC, PKC, CaMKII; membrane Ca²⁺ entry via LTCC and SOCE; intracellular Ca²⁺ release via IP₃R and RyR) significantly reduced SR141716A-evoked vomiting in a dose-dependent manner, indicating these pathways are necessary for the behavioral output under our conditions.
CB₁ receptor predominantly couples to Gi/o to suppress adenylyl cyclase. The observed ↑cAMP/↑PKA after SR141716A is the expected direct consequence of inverse agonism (relief of constitutive Gi/o tone). Also, we found no acute change in CB1 protein density in DVC after SR141716A, arguing that the signaling pattern reflects a change in receptor signaling state rather than receptor density. Engagement of ERK, PI3K/Akt/GSK-3, PLC/PKC, CaMKII, and Ca²⁺ mobilization (LTCC/SOCE and IP₃R/RyR) is consistent with well-known cross-talk downstream of cAMP/PKA in emetic nuclei. In short, cAMP/PKA is a direct readout of CB1 inverse agonism, and the other activated intracellular emetic signals represent downstream activation within the emetic network. Our data show that blocking any of these signals attenuates emesis, so they are causally involved even if not all are “direct” CB1 outputs in every neuron.
In our previous studies, we have demonstrated that the tested compounds used in this study also inhibited shrews vomiting induced by diverse emetogens. In this study, we show that classical antiemetics acting at 5-HT₃ (palonosetron) and D₂/₃ (sulpiride) receptors attenuate SR141716A-evoked vomiting, and that Ca²⁺-channel blockers (LTCC, SOCE, IP₃R, RyR) and kinase inhibitors (PKA, ERK, GSK-3, PI3K, PLC, PKC, CaMKII) are effective against SR141716A-evoked vomiting. Because several of these drug classes are known to act across multiple emetic triggers, their efficacy here indicates that SR141716A engages shared emetic machinery; i.e., the downstream effects are not solely attributable to CB1 blockade per se, but though CB1 inverse agonism is the initiating event in our paradigm.
Our data indicate that SR141716A-induced vomiting is initiated by a CB₁ receptor–specific mechanism. Specifically: (1) Upstream (CB₁-specific): SR141716A’s inverse agonism at CB₁ relieves Gi/o-mediated restraint on adenylyl cyclase, increasing cAMP/PKA and initiating the intracellular cascade that culminates in emesis; (2) Downstream (general/convergent): the execution layer—ERK, PI3K/Akt–GSK-3, PLC/PKC, CaMKII, and coordinated Ca²⁺ entry/release—constitutes a general emetic program engaged by multiple triggers; accordingly, blocking these signals inhibits SR141716A-evoked vomiting.
In summary, SR141716A triggers emesis via CB1 inverse agonism (CB1-specific initiation), but the response is implemented by shared intracellular effectors (general convergence). Our time-course biochemistry plus pathway-targeted inhibition convert the findings from descriptive to mechanistically causal for the emetic phenotype studied here.
- Although multiple pathways are implicated, the study does not sufficiently address alternative or compensatory signaling cascades that may contribute to the emetic response. For example, the lack of increase in endogenous serotonin (5-HT) and substance P (SP) tissue levels contradicts the findings that serotonin and dopamine receptor antagonists are efficacious in the described set-up. This discrepancy merits closer examination, possibly involving release kinetics, metabolism, or receptor sensitivity. Same for p-ERK: ERK would also be activated by CB1 agonists, which are usually anti-emetic. Please elaborate. Similarly, cannabis hyperemesis syndrome is caused by CB1 agonists. This is one major side effect in some patients and since it can be attenuated by TRPV1 agonists, this should be discussed here as well.
1) We agree that although SR141716A treatment did not increase tissue levels of serotonin (5-HT) or substance P (SP) in the dorsal vagal complex, the efficacy of 5-HT₃ and D₂ receptor antagonists in attenuating SR141716A-induced emesis suggests that receptor-level sensitivity, release dynamics, or neurotransmitter turnover may play a more critical role than static tissue levels.
We appreciate the referee’s suggestion, and in our re-revised manuscript we have added the following discussion in the manuscript, “iv) the release dynamics, or receptor-level sensitivity may also play a more critical role than static tissue levels,” Please see lines 846-847.
2) p-ERK1/2 functions as a convergent integrator of multiple inputs; therefore, the behavioral outcome depends on the net effect at the circuit level. CB1 agonists generally reduce presynaptic glutamate/5-HT/GABA release (Gi/o tone), producing an overall antiemetic network effect, even if ERK1/2 is transiently activated (e.g., via β-arrestin pathways). SR141716A, as a CB1 inverse agonist, disinhibits AC→cAMP/PKA, permitting broader Ca²⁺ mobilization (LTCC/SOCE/IP₃R/RyR) and ERK/PI3K–Akt–GSK-3/PLC–PKC/CaMKII engagement that together promote emesis.
3) Per referee 2’s suggestion, we have added the following discussion in the manuscript, “Cannabis hyperemesis syndrome (CHS) occurs following chronic administration of large doses of CB₁ agonists, further illustrates the complexity of endocannabinoid signaling. Interestingly, TRPV1R agonists such as capsaicin provide relief in CHS patients, supporting the idea that TRPV1R-CB₁ crosstalk and compensatory adaptations of intracellular pathways are critical determinants of the emetic phenotype [128, 129]. In fact, we have demonstrated that a combination of CB1- and TRPV1 receptor agonists have the capacity to completely abolish cisplatin-evoked emesis at doses that are ineffective when used individually [121].” Please see lines 776-783.
- The peripheral involvement has only been shown indirectly through stainings in the ENS. However, whether those changes are emetic has not been shown. This can be shown by vagotomy but all evidence shown here is indirect.
The DVC is the canonical integrator of vagal afferents driving emesis, so concurrent ENS and DVC activation is expected if the brain–gut axis is engaged. ENS activity markers rise in the same time window as vomiting and DVC (AP/NTS/DMNX) activation, compatible with a gut → vagus → DVC pathway. Our ENS staining together with DVC activation supports involvement of the brain–gut axis during SR141716A-evoked emesis. Accordingly, in future studies we will perform subdiaphragmatic vagotomy or reversible perivagal blockade to determine the involvement of the entire brain–gut axis in the induced emetic response. We thank you for your suggestion!
- Translational and Clinical Implications are vague: The study's focus on mechanistic dissection in the least shrew model does not adequately bridge to clinical implications for cannabinoid-based antiemetic therapy or adverse emetic effects associated with CB1 inhibitors. More explicit discussion on how these findings may inform drug design or treatment of nausea and vomiting in patients would increase relevance.
While this study was conducted in the least shrew, a well-validated model for emesis research, the mechanistic findings have potential translational relevance for human conditions. The ability of SR141716A to trigger vomiting through CB₁ inverse agonism underscores the critical role of endocannabinoid signaling in emetic regulation and helps explain why SR141716A was associated with adverse gastrointestinal and neuropsychiatric effects in the clinic. Furthermore, our results suggest that targeting downstream effectors such as ERK1/2, PI3K/Akt, and calcium signaling may provide alternative strategies for antiemetic drug development beyond conventional 5-HT₃ and dopamine receptor antagonists. The observed overlap with signaling pathways implicated in CHS also highlights the translational relevance of our findings, as they may help inform management strategies for patients suffering from this increasingly recognized condition. Overall, while cannabis cessation remains the definitive treatment for CHS, our data suggest that pharmacological modulation of convergent signaling cascades may hold promise as adjunctive therapies. We cannot provide further details since we were asked to reduce the length of the original manuscript.
Referee 2
Comments and Suggestions for Authors
This study by Sun et al. presents an extensive investigation into the emetic mechanisms triggered by the cannabinoid CB1 receptor inverse agonist SR141716A in least shrews, revealing complex intracellular signaling pathways involved in vomiting. However, the study, while thorough, leans heavily on descriptive experimental data and would benefit from deeper mechanistic insights.
The authors systematically demonstrate that Rimonabant induces vomiting via both central and peripheral mechanisms, chiefly the dorsal vagal complex (DVC) in the brainstem and the enteric nervous system in the jejunum, with evidence from c-fos and phospho-ERK1/2 immunfluorescence, among others. They provide a link between brainstem cAMP levels and sequential phosphorylation of key kinases such as PKA, Akt, GSK-3α/β, ERK1/2, and PKCαβII to the emetic response. The broad-spectrum pharmacological blockade studies targeting these kinases and various calcium channels strongly support their involvement in SR141716A-evoked vomiting.
Critical Issues:
- Antibody Validity for CB1 Detection and general antibody-related issues: A key weakness undermining the interpretation of CB1 receptor distribution is the use of a commercial anti-CB1 antibody (ab3558, Abcam) which has been withdrawn from the market by the company. This raises questions regarding the specificity and reliability of CB1 receptor immunostaining data presented. In fact, previous studies (Histochem Cell Biol. 2021 Aug 27;156(5):479–502, J Neurosci Methods. 2008 Jun 15;171(1):78-8) investigated CB1 antibody specificity and found that many CB1 antibodies do not specifically detect the protein. Staining specificity of the antibody used should be either verified by using another CB1 antibody, another technique (Western Blot) or by the use of knockout animals (in this case not possible). In Fig. 1 CB1 staining is shown. However, staining doesn’t look specific since I only see a blurry reddish image. Maybe supplying a magnification could shed some light whether the antibody stains specifically. A quantification of only those cells that are labelled positive is not a very good method since this leaves out cells with low level expression. Rather, staining intensity should be considered for quantification. In addition, I would like to see an isotype staining, especially as a control for CB1.
We agree that CB1 antibodies vary in performance. In our study, CB1 immunostaining was used only to provide regional context (AP/NTS/DMNX mapping) and to note that acute SR141716A did not alter regional CB1 density; our key conclusions (↑cAMP/PKA, engagement of ERK/PI3K–Akt–GSK-3/PLC–PKC/CaMKII, Ca²⁺ mobilization, and inhibitor sensitivity of emesis) do not depend on subtle intensity gradients of CB1 immunoreactivity.
We acknowledge that ab3558 has been discontinued by the vendor; however, discontinuation alone is not evidence of non-specificity. Several studies have used this antibody to detect CB1 receptor expression, including recent publications (Shen et al., 2010; PMID: 20680167; Wang et al., 2022; PMID: 35269905; Sevil et al., 2024; PMID: 39758842; Köse et al., 2024; PMID: 39354544). Given the 10-day revision window, it is not feasible to obtain and validate an alternative antibody and repeat these experiments. Instead of re-performing this component with a different antibody, we strengthened validation and presentation of the existing dataset as follows:
- Negative controls: Secondary-only and host-matched rabbit IgG isotype controls were processed in parallel and imaged with identical acquisition parameters to quantify background; these data are now included. Please see attached supplemental material.
- High-magnification imaging: As noted in Section 4.4.1, tile-scan images were acquired on a Zeiss LSM 880 confocal microscope (1024 × 1024 pixels; Plan-Apochromat 63×/1.4 Oil DIC M27 objective). We now added high-magnification insets in Fig. 1E–G from the boxed regions in panel C (AP, NTS, DMNX).
- Re-analysis: CB1 signal was quantified as mean gray value per µm² within predefined ROIs.
- c-fos immunoreactivity seems to be highest outside the areas of interest. Why? Do those cells carry CB1? Why didn’t the authors co-stain with CB1?
The c-fos panels are tile scans that necessarily include tissue adjacent to the dorsal vagal complex (DVC). However, our quantification was restricted to the prespecified emetic nuclei—AP, NTS, and DMNX. The DVC contains not only neurons that express CB1 receptors but also other neuronal types (e.g., glutamatergic and GABAergic), and its nuclei have complex connections with other brain regions, such as the hypothalamus, amygdala, cortex, and so on. In this study, drug administration was systemic (via the circulation) rather than a local injection into the DVC. The goal of this study was therefore to evaluate the overall activation of the DVC after SR141716A administration rather than cellular co-localization. Functionally, SR141716A-evoked emesis is multideterminant (involving cAMP/PKA, ERK1/2, PI3K/Akt/GSK-3, PLC/PKC, CaMKII, and the classical 5-HT3 and D2/3 receptors), so co-localizing c-fos with CB1 would not, by itself, distinguish activation of CB1-expressing neurons from downstream network activation.
- cAMP levels: A positive control would have helped to put these values into perspective. The authors mentioned that rolipram has a similar effect. Why not use this as positive control?
We thank the reviewer for this insightful suggestion. In the present study, we observed a significant increase in cAMP levels following SR141716A administration, consistent with its established role as a CB₁ inverse agonist that disinhibits adenylate cyclase activity. Although rolipram, a phosphodiesterase-4 inhibitor, is known to elevate intracellular cAMP, we did not include it as a positive control here because we have previously reported that rolipram significantly increases brainstem cAMP in the least shrew (Alkam et al., 2014-PMID: 24513510), which supports our present findings. Nevertheless, we agree that such a control would strengthen the interpretation, and we will incorporate rolipram or other reference modulators in future studies to provide a clearer comparative context for cAMP signaling.
- The study overwhelmingly shows descriptive data without explaining the precise molecular interactions between SR141716A and intracellular effectors. The complexity of GPCR biased signaling and the effects of inverse agonism by rimonabant is acknowledged but insufficiently explored, missing an opportunity to provide insight into why SR141716A acts as an inverse agonist triggering emesis while agonists generally confer antiemetic effects. Are these pathways that were investigated all directly linked to CB1 inverse agonism? Would the compounds used also inhibit vomiting induced by a different mechanism? This would be interesting, because then the observed effects would not be solely attributable to CB1 inhibition. This also leads me to another question: Is the activation of the described intracellular pathways by rimonabant a CB1 specific mechanism or is this a general activation scheme common to all emetic compounds?
In this study, we have demonstrated that SR141716A (20 mg/kg, i.p.) produced a time-dependent increase in brainstem cAMP (↑ at 15 and 30 min), a peak in PKA (Thr197) phosphorylation at 30 min, and rapid phosphorylation of ERK1/2, Akt, GSK-3α/β, and PKCα/βII (peaking at ~15 min and persisting for some nodes). These biochemical changes coincided with the observed emesis and DVC activation (↑c-fos; ↑p-ERK1/2), establishing temporal linkage between receptor perturbation and effector pathways. Functionally, pharmacologic inhibition of these emetic signals (PKA, ERK1/2, GSK-3, PI3K, PLC, PKC, CaMKII; membrane Ca²⁺ entry via LTCC and SOCE; intracellular Ca²⁺ release via IP₃R and RyR) significantly reduced SR141716A-evoked vomiting in a dose-dependent manner, indicating these pathways are necessary for the behavioral output under our conditions.
CB₁ receptor predominantly couples to Gi/o to suppress adenylyl cyclase. The observed ↑cAMP/↑PKA after SR141716A is the expected direct consequence of inverse agonism (relief of constitutive Gi/o tone). Also, we found no acute change in CB1 protein density in DVC after SR141716A, arguing that the signaling pattern reflects a change in receptor signaling state rather than receptor density. Engagement of ERK, PI3K/Akt/GSK-3, PLC/PKC, CaMKII, and Ca²⁺ mobilization (LTCC/SOCE and IP₃R/RyR) is consistent with well-known cross-talk downstream of cAMP/PKA in emetic nuclei. In short, cAMP/PKA is a direct readout of CB1 inverse agonism, and the other activated intracellular emetic signals represent downstream activation within the emetic network. Our data show that blocking any of these signals attenuates emesis, so they are causally involved even if not all are “direct” CB1 outputs in every neuron.
In our previous studies, we have demonstrated that the tested compounds used in this study also inhibited shrews vomiting induced by diverse emetogens. In this study, we show that classical antiemetics acting at 5-HT₃ (palonosetron) and D₂/₃ (sulpiride) receptors attenuate SR141716A-evoked vomiting, and that Ca²⁺-channel blockers (LTCC, SOCE, IP₃R, RyR) and kinase inhibitors (PKA, ERK, GSK-3, PI3K, PLC, PKC, CaMKII) are effective against SR141716A-evoked vomiting. Because several of these drug classes are known to act across multiple emetic triggers, their efficacy here indicates that SR141716A engages shared emetic machinery; i.e., the downstream effects are not solely attributable to CB1 blockade per se, but though CB1 inverse agonism is the initiating event in our paradigm.
Our data indicate that SR141716A-induced vomiting is initiated by a CB₁ receptor–specific mechanism. Specifically: (1) Upstream (CB₁-specific): SR141716A’s inverse agonism at CB₁ relieves Gi/o-mediated restraint on adenylyl cyclase, increasing cAMP/PKA and initiating the intracellular cascade that culminates in emesis; (2) Downstream (general/convergent): the execution layer—ERK, PI3K/Akt–GSK-3, PLC/PKC, CaMKII, and coordinated Ca²⁺ entry/release—constitutes a general emetic program engaged by multiple triggers; accordingly, blocking these signals inhibits SR141716A-evoked vomiting.
In summary, SR141716A triggers emesis via CB1 inverse agonism (CB1-specific initiation), but the response is implemented by shared intracellular effectors (general convergence). Our time-course biochemistry plus pathway-targeted inhibition convert the findings from descriptive to mechanistically causal for the emetic phenotype studied here.
- Although multiple pathways are implicated, the study does not sufficiently address alternative or compensatory signaling cascades that may contribute to the emetic response. For example, the lack of increase in endogenous serotonin (5-HT) and substance P (SP) tissue levels contradicts the findings that serotonin and dopamine receptor antagonists are efficacious in the described set-up. This discrepancy merits closer examination, possibly involving release kinetics, metabolism, or receptor sensitivity. Same for p-ERK: ERK would also be activated by CB1 agonists, which are usually anti-emetic. Please elaborate. Similarly, cannabis hyperemesis syndrome is caused by CB1 agonists. This is one major side effect in some patients and since it can be attenuated by TRPV1 agonists, this should be discussed here as well.
1) We agree that although SR141716A treatment did not increase tissue levels of serotonin (5-HT) or substance P (SP) in the dorsal vagal complex, the efficacy of 5-HT₃ and D₂ receptor antagonists in attenuating SR141716A-induced emesis suggests that receptor-level sensitivity, release dynamics, or neurotransmitter turnover may play a more critical role than static tissue levels.
We appreciate the referee’s suggestion, and in our re-revised manuscript we have added the following discussion in the manuscript, “iv) the release dynamics, or receptor-level sensitivity may also play a more critical role than static tissue levels,” Please see lines 846-847.
2) p-ERK1/2 functions as a convergent integrator of multiple inputs; therefore, the behavioral outcome depends on the net effect at the circuit level. CB1 agonists generally reduce presynaptic glutamate/5-HT/GABA release (Gi/o tone), producing an overall antiemetic network effect, even if ERK1/2 is transiently activated (e.g., via β-arrestin pathways). SR141716A, as a CB1 inverse agonist, disinhibits AC→cAMP/PKA, permitting broader Ca²⁺ mobilization (LTCC/SOCE/IP₃R/RyR) and ERK/PI3K–Akt–GSK-3/PLC–PKC/CaMKII engagement that together promote emesis.
3) Per referee 2’s suggestion, we have added the following discussion in the manuscript, “Cannabis hyperemesis syndrome (CHS) occurs following chronic administration of large doses of CB₁ agonists, further illustrates the complexity of endocannabinoid signaling. Interestingly, TRPV1R agonists such as capsaicin provide relief in CHS patients, supporting the idea that TRPV1R-CB₁ crosstalk and compensatory adaptations of intracellular pathways are critical determinants of the emetic phenotype [128, 129]. In fact, we have demonstrated that a combination of CB1- and TRPV1 receptor agonists have the capacity to completely abolish cisplatin-evoked emesis at doses that are ineffective when used individually [121].” Please see lines 776-783.
- The peripheral involvement has only been shown indirectly through stainings in the ENS. However, whether those changes are emetic has not been shown. This can be shown by vagotomy but all evidence shown here is indirect.
The DVC is the canonical integrator of vagal afferents driving emesis, so concurrent ENS and DVC activation is expected if the brain–gut axis is engaged. ENS activity markers rise in the same time window as vomiting and DVC (AP/NTS/DMNX) activation, compatible with a gut → vagus → DVC pathway. Our ENS staining together with DVC activation supports involvement of the brain–gut axis during SR141716A-evoked emesis. Accordingly, in future studies we will perform subdiaphragmatic vagotomy or reversible perivagal blockade to determine the involvement of the entire brain–gut axis in the induced emetic response. We thank you for your suggestion!
- Translational and Clinical Implications are vague: The study's focus on mechanistic dissection in the least shrew model does not adequately bridge to clinical implications for cannabinoid-based antiemetic therapy or adverse emetic effects associated with CB1 inhibitors. More explicit discussion on how these findings may inform drug design or treatment of nausea and vomiting in patients would increase relevance.
While this study was conducted in the least shrew, a well-validated model for emesis research, the mechanistic findings have potential translational relevance for human conditions. The ability of SR141716A to trigger vomiting through CB₁ inverse agonism underscores the critical role of endocannabinoid signaling in emetic regulation and helps explain why SR141716A was associated with adverse gastrointestinal and neuropsychiatric effects in the clinic. Furthermore, our results suggest that targeting downstream effectors such as ERK1/2, PI3K/Akt, and calcium signaling may provide alternative strategies for antiemetic drug development beyond conventional 5-HT₃ and dopamine receptor antagonists. The observed overlap with signaling pathways implicated in CHS also highlights the translational relevance of our findings, as they may help inform management strategies for patients suffering from this increasingly recognized condition. Overall, while cannabis cessation remains the definitive treatment for CHS, our data suggest that pharmacological modulation of convergent signaling cascades may hold promise as adjunctive therapies. We cannot provide further details since we were asked to reduce the length of the original manuscript.
Referee 2
Comments and Suggestions for Authors
This study by Sun et al. presents an extensive investigation into the emetic mechanisms triggered by the cannabinoid CB1 receptor inverse agonist SR141716A in least shrews, revealing complex intracellular signaling pathways involved in vomiting. However, the study, while thorough, leans heavily on descriptive experimental data and would benefit from deeper mechanistic insights.
The authors systematically demonstrate that Rimonabant induces vomiting via both central and peripheral mechanisms, chiefly the dorsal vagal complex (DVC) in the brainstem and the enteric nervous system in the jejunum, with evidence from c-fos and phospho-ERK1/2 immunfluorescence, among others. They provide a link between brainstem cAMP levels and sequential phosphorylation of key kinases such as PKA, Akt, GSK-3α/β, ERK1/2, and PKCαβII to the emetic response. The broad-spectrum pharmacological blockade studies targeting these kinases and various calcium channels strongly support their involvement in SR141716A-evoked vomiting.
Critical Issues:
- Antibody Validity for CB1 Detection and general antibody-related issues: A key weakness undermining the interpretation of CB1 receptor distribution is the use of a commercial anti-CB1 antibody (ab3558, Abcam) which has been withdrawn from the market by the company. This raises questions regarding the specificity and reliability of CB1 receptor immunostaining data presented. In fact, previous studies (Histochem Cell Biol. 2021 Aug 27;156(5):479–502, J Neurosci Methods. 2008 Jun 15;171(1):78-8) investigated CB1 antibody specificity and found that many CB1 antibodies do not specifically detect the protein. Staining specificity of the antibody used should be either verified by using another CB1 antibody, another technique (Western Blot) or by the use of knockout animals (in this case not possible). In Fig. 1 CB1 staining is shown. However, staining doesn’t look specific since I only see a blurry reddish image. Maybe supplying a magnification could shed some light whether the antibody stains specifically. A quantification of only those cells that are labelled positive is not a very good method since this leaves out cells with low level expression. Rather, staining intensity should be considered for quantification. In addition, I would like to see an isotype staining, especially as a control for CB1.
We agree that CB1 antibodies vary in performance. In our study, CB1 immunostaining was used only to provide regional context (AP/NTS/DMNX mapping) and to note that acute SR141716A did not alter regional CB1 density; our key conclusions (↑cAMP/PKA, engagement of ERK/PI3K–Akt–GSK-3/PLC–PKC/CaMKII, Ca²⁺ mobilization, and inhibitor sensitivity of emesis) do not depend on subtle intensity gradients of CB1 immunoreactivity.
We acknowledge that ab3558 has been discontinued by the vendor; however, discontinuation alone is not evidence of non-specificity. Several studies have used this antibody to detect CB1 receptor expression, including recent publications (Shen et al., 2010; PMID: 20680167; Wang et al., 2022; PMID: 35269905; Sevil et al., 2024; PMID: 39758842; Köse et al., 2024; PMID: 39354544). Given the 10-day revision window, it is not feasible to obtain and validate an alternative antibody and repeat these experiments. Instead of re-performing this component with a different antibody, we strengthened validation and presentation of the existing dataset as follows:
- Negative controls: Secondary-only and host-matched rabbit IgG isotype controls were processed in parallel and imaged with identical acquisition parameters to quantify background; these data are now included. Please see attached supplemental material.
- High-magnification imaging: As noted in Section 4.4.1, tile-scan images were acquired on a Zeiss LSM 880 confocal microscope (1024 × 1024 pixels; Plan-Apochromat 63×/1.4 Oil DIC M27 objective). We now added high-magnification insets in Fig. 1E–G from the boxed regions in panel C (AP, NTS, DMNX).
- Re-analysis: CB1 signal was quantified as mean gray value per µm² within predefined ROIs.
- c-fos immunoreactivity seems to be highest outside the areas of interest. Why? Do those cells carry CB1? Why didn’t the authors co-stain with CB1?
The c-fos panels are tile scans that necessarily include tissue adjacent to the dorsal vagal complex (DVC). However, our quantification was restricted to the prespecified emetic nuclei—AP, NTS, and DMNX. The DVC contains not only neurons that express CB1 receptors but also other neuronal types (e.g., glutamatergic and GABAergic), and its nuclei have complex connections with other brain regions, such as the hypothalamus, amygdala, cortex, and so on. In this study, drug administration was systemic (via the circulation) rather than a local injection into the DVC. The goal of this study was therefore to evaluate the overall activation of the DVC after SR141716A administration rather than cellular co-localization. Functionally, SR141716A-evoked emesis is multideterminant (involving cAMP/PKA, ERK1/2, PI3K/Akt/GSK-3, PLC/PKC, CaMKII, and the classical 5-HT3 and D2/3 receptors), so co-localizing c-fos with CB1 would not, by itself, distinguish activation of CB1-expressing neurons from downstream network activation.
- cAMP levels: A positive control would have helped to put these values into perspective. The authors mentioned that rolipram has a similar effect. Why not use this as positive control?
We thank the reviewer for this insightful suggestion. In the present study, we observed a significant increase in cAMP levels following SR141716A administration, consistent with its established role as a CB₁ inverse agonist that disinhibits adenylate cyclase activity. Although rolipram, a phosphodiesterase-4 inhibitor, is known to elevate intracellular cAMP, we did not include it as a positive control here because we have previously reported that rolipram significantly increases brainstem cAMP in the least shrew (Alkam et al., 2014-PMID: 24513510), which supports our present findings. Nevertheless, we agree that such a control would strengthen the interpretation, and we will incorporate rolipram or other reference modulators in future studies to provide a clearer comparative context for cAMP signaling.
- The study overwhelmingly shows descriptive data without explaining the precise molecular interactions between SR141716A and intracellular effectors. The complexity of GPCR biased signaling and the effects of inverse agonism by rimonabant is acknowledged but insufficiently explored, missing an opportunity to provide insight into why SR141716A acts as an inverse agonist triggering emesis while agonists generally confer antiemetic effects. Are these pathways that were investigated all directly linked to CB1 inverse agonism? Would the compounds used also inhibit vomiting induced by a different mechanism? This would be interesting, because then the observed effects would not be solely attributable to CB1 inhibition. This also leads me to another question: Is the activation of the described intracellular pathways by rimonabant a CB1 specific mechanism or is this a general activation scheme common to all emetic compounds?
In this study, we have demonstrated that SR141716A (20 mg/kg, i.p.) produced a time-dependent increase in brainstem cAMP (↑ at 15 and 30 min), a peak in PKA (Thr197) phosphorylation at 30 min, and rapid phosphorylation of ERK1/2, Akt, GSK-3α/β, and PKCα/βII (peaking at ~15 min and persisting for some nodes). These biochemical changes coincided with the observed emesis and DVC activation (↑c-fos; ↑p-ERK1/2), establishing temporal linkage between receptor perturbation and effector pathways. Functionally, pharmacologic inhibition of these emetic signals (PKA, ERK1/2, GSK-3, PI3K, PLC, PKC, CaMKII; membrane Ca²⁺ entry via LTCC and SOCE; intracellular Ca²⁺ release via IP₃R and RyR) significantly reduced SR141716A-evoked vomiting in a dose-dependent manner, indicating these pathways are necessary for the behavioral output under our conditions.
CB₁ receptor predominantly couples to Gi/o to suppress adenylyl cyclase. The observed ↑cAMP/↑PKA after SR141716A is the expected direct consequence of inverse agonism (relief of constitutive Gi/o tone). Also, we found no acute change in CB1 protein density in DVC after SR141716A, arguing that the signaling pattern reflects a change in receptor signaling state rather than receptor density. Engagement of ERK, PI3K/Akt/GSK-3, PLC/PKC, CaMKII, and Ca²⁺ mobilization (LTCC/SOCE and IP₃R/RyR) is consistent with well-known cross-talk downstream of cAMP/PKA in emetic nuclei. In short, cAMP/PKA is a direct readout of CB1 inverse agonism, and the other activated intracellular emetic signals represent downstream activation within the emetic network. Our data show that blocking any of these signals attenuates emesis, so they are causally involved even if not all are “direct” CB1 outputs in every neuron.
In our previous studies, we have demonstrated that the tested compounds used in this study also inhibited shrews vomiting induced by diverse emetogens. In this study, we show that classical antiemetics acting at 5-HT₃ (palonosetron) and D₂/₃ (sulpiride) receptors attenuate SR141716A-evoked vomiting, and that Ca²⁺-channel blockers (LTCC, SOCE, IP₃R, RyR) and kinase inhibitors (PKA, ERK, GSK-3, PI3K, PLC, PKC, CaMKII) are effective against SR141716A-evoked vomiting. Because several of these drug classes are known to act across multiple emetic triggers, their efficacy here indicates that SR141716A engages shared emetic machinery; i.e., the downstream effects are not solely attributable to CB1 blockade per se, but though CB1 inverse agonism is the initiating event in our paradigm.
Our data indicate that SR141716A-induced vomiting is initiated by a CB₁ receptor–specific mechanism. Specifically: (1) Upstream (CB₁-specific): SR141716A’s inverse agonism at CB₁ relieves Gi/o-mediated restraint on adenylyl cyclase, increasing cAMP/PKA and initiating the intracellular cascade that culminates in emesis; (2) Downstream (general/convergent): the execution layer—ERK, PI3K/Akt–GSK-3, PLC/PKC, CaMKII, and coordinated Ca²⁺ entry/release—constitutes a general emetic program engaged by multiple triggers; accordingly, blocking these signals inhibits SR141716A-evoked vomiting.
In summary, SR141716A triggers emesis via CB1 inverse agonism (CB1-specific initiation), but the response is implemented by shared intracellular effectors (general convergence). Our time-course biochemistry plus pathway-targeted inhibition convert the findings from descriptive to mechanistically causal for the emetic phenotype studied here.
- Although multiple pathways are implicated, the study does not sufficiently address alternative or compensatory signaling cascades that may contribute to the emetic response. For example, the lack of increase in endogenous serotonin (5-HT) and substance P (SP) tissue levels contradicts the findings that serotonin and dopamine receptor antagonists are efficacious in the described set-up. This discrepancy merits closer examination, possibly involving release kinetics, metabolism, or receptor sensitivity. Same for p-ERK: ERK would also be activated by CB1 agonists, which are usually anti-emetic. Please elaborate. Similarly, cannabis hyperemesis syndrome is caused by CB1 agonists. This is one major side effect in some patients and since it can be attenuated by TRPV1 agonists, this should be discussed here as well.
1) We agree that although SR141716A treatment did not increase tissue levels of serotonin (5-HT) or substance P (SP) in the dorsal vagal complex, the efficacy of 5-HT₃ and D₂ receptor antagonists in attenuating SR141716A-induced emesis suggests that receptor-level sensitivity, release dynamics, or neurotransmitter turnover may play a more critical role than static tissue levels.
We appreciate the referee’s suggestion, and in our re-revised manuscript we have added the following discussion in the manuscript, “iv) the release dynamics, or receptor-level sensitivity may also play a more critical role than static tissue levels,” Please see lines 846-847.
2) p-ERK1/2 functions as a convergent integrator of multiple inputs; therefore, the behavioral outcome depends on the net effect at the circuit level. CB1 agonists generally reduce presynaptic glutamate/5-HT/GABA release (Gi/o tone), producing an overall antiemetic network effect, even if ERK1/2 is transiently activated (e.g., via β-arrestin pathways). SR141716A, as a CB1 inverse agonist, disinhibits AC→cAMP/PKA, permitting broader Ca²⁺ mobilization (LTCC/SOCE/IP₃R/RyR) and ERK/PI3K–Akt–GSK-3/PLC–PKC/CaMKII engagement that together promote emesis.
3) Per referee 2’s suggestion, we have added the following discussion in the manuscript, “Cannabis hyperemesis syndrome (CHS) occurs following chronic administration of large doses of CB₁ agonists, further illustrates the complexity of endocannabinoid signaling. Interestingly, TRPV1R agonists such as capsaicin provide relief in CHS patients, supporting the idea that TRPV1R-CB₁ crosstalk and compensatory adaptations of intracellular pathways are critical determinants of the emetic phenotype [128, 129]. In fact, we have demonstrated that a combination of CB1- and TRPV1 receptor agonists have the capacity to completely abolish cisplatin-evoked emesis at doses that are ineffective when used individually [121].” Please see lines 776-783.
- The peripheral involvement has only been shown indirectly through stainings in the ENS. However, whether those changes are emetic has not been shown. This can be shown by vagotomy but all evidence shown here is indirect.
The DVC is the canonical integrator of vagal afferents driving emesis, so concurrent ENS and DVC activation is expected if the brain–gut axis is engaged. ENS activity markers rise in the same time window as vomiting and DVC (AP/NTS/DMNX) activation, compatible with a gut → vagus → DVC pathway. Our ENS staining together with DVC activation supports involvement of the brain–gut axis during SR141716A-evoked emesis. Accordingly, in future studies we will perform subdiaphragmatic vagotomy or reversible perivagal blockade to determine the involvement of the entire brain–gut axis in the induced emetic response. We thank you for your suggestion!
- Translational and Clinical Implications are vague: The study's focus on mechanistic dissection in the least shrew model does not adequately bridge to clinical implications for cannabinoid-based antiemetic therapy or adverse emetic effects associated with CB1 inhibitors. More explicit discussion on how these findings may inform drug design or treatment of nausea and vomiting in patients would increase relevance.
While this study was conducted in the least shrew, a well-validated model for emesis research, the mechanistic findings have potential translational relevance for human conditions. The ability of SR141716A to trigger vomiting through CB₁ inverse agonism underscores the critical role of endocannabinoid signaling in emetic regulation and helps explain why SR141716A was associated with adverse gastrointestinal and neuropsychiatric effects in the clinic. Furthermore, our results suggest that targeting downstream effectors such as ERK1/2, PI3K/Akt, and calcium signaling may provide alternative strategies for antiemetic drug development beyond conventional 5-HT₃ and dopamine receptor antagonists. The observed overlap with signaling pathways implicated in CHS also highlights the translational relevance of our findings, as they may help inform management strategies for patients suffering from this increasingly recognized condition. Overall, while cannabis cessation remains the definitive treatment for CHS, our data suggest that pharmacological modulation of convergent signaling cascades may hold promise as adjunctive therapies. We cannot provide further details since we were asked to reduce the length of the original manuscript.
Round 2
Reviewer 1 Report (New Reviewer)
Comments and Suggestions for Authors
Unfortunately, I'm not fully satisfied with the Authors’ response. I agree with the Authors that rimonabant is still applied in various experiments as CB₁ receptor antagonist/inverse agonist even after its withdrawal from the market in 2008. However:
- its use has been steadily declining; according to PubMed 265, 189, 98 and 45 publications with rimonabant in 2007, 2010, 2015 and 2024, respectively and 283, 200, 102 and 49 publications with SR141716A in 2007, 2010, 2015 and 2024, respectively were published;
- The Authors examined a very high dose of SR141716A (rimonabant) that stimulates vomiting 20 mg/kg (given intraperitoneally; i.p.); such a large dose is not typically used in experiments where rimonabant (i.p) is used as CB₁ receptor antagonist/inverse agonist. Even the Authors’ group demonstrated previously on shrews that SR141716A (rimonabant) reversed the antiemetic effects of cannabinoid receptor agonist WIN 55, 212-2 against cisplatin-induced emesis with an ID (i.e. the inhibitory dose that prevented emesis in 50% of shrews) of 0.27+/-1.56 mg/kg (Darmani NA. The cannabinoid CB1 receptor antagonist SR 141716A reverses the antiemetic and motor depressant actions of WIN 55, 212-2. Eur J Pharmacol.;430(1):49-58)
Below additional examples of the use of SR141716A (rimonabant; i.p.) as a pharmacological tool are given:
- 1.0 mg/kg−1 SR141716: O'Brien LD, Limebeer CL, Rock EM, Bottegoni G, Piomelli D, Parker LA. Anandamide transport inhibition by ARN272 attenuates nausea-induced behaviour in rats, and vomiting in shrews (Suncus murinus). Br J Pharmacol. 2013 Nov;170(5):1130-6. doi: 10.1111/bph.12360;
why was this publication not quoted by the Authors?
- rimonabant (3, 10 mg/kg), Soler-Cedeño O, et al. AM6527, a neutral CB1 receptor antagonist, suppresses opioid taking and seeking, as well as cocaine seeking in rodents ;without aversive effects. Neuropsychopharmacology. 2024;49(11):1678-1688;
- rimonabant (1 mg/kg); Li et al. Resting State Brain Networks under Inverse Agonist versus Complete Knockout of the Cannabinoid Receptor 1. ACS Chem Neurosci. 2024;15(8):1669-1683;
- rimonabant (10 mg/kg); Jiang et al. A monoacylglycerol lipase inhibitor showing therapeutic efficacy in mice without central side effects or dependence. Nat Commun. 2023;14(1):8039;
- rimonabant (5 mg/kg); Cao et al. Microglial adenosine A2A receptor in the paraventricular thalamic nucleus regulates pain sensation and analgesic effects independent of opioid and cannabinoid receptors. Front Pharmacol. 2024 Dec 19;15:1467305.
In conclusion, Authors did not prove sufficiently the significance their manuscript. Moreover, they failed to discuss the significance of their results.
Author Response
Referee 1
Sun et al. examined in their paper intracellular emetic signals involved in cannabinoid CB1 receptor inverse agonist/antagonist, SR141716A given at a dose of (20 mg/kg, i.p.) to least shrews (Cryptotis parva). Although the authors suggested numerous potential mechanisms for SR141716A, in my opinion the significance of their conclusions is rather low because of the following reasons:
- SR141716A is the chemical name for rimonabant, which was approved for clinical use in around 40 countries. Unfortunately, its serious CNS-mediated unwanted neuropsychiatric effects such as anxiety and increased suicidal ideationled to its withdrawal from the market in 2008 (e.g. Cinar et al. doi: 10.1016/j.pharmthera.2020.107477).
We agree that SR141716A (rimonabant) was suspended by the European Medicines Agency in 2008 due to neuropsychiatric adverse effects, including anxiety and increased suicidal ideation (Cinar et al., 2020). We would also like to emphasize that, although SR141716A is no longer clinically available, it remains the principal prototypical and most widely studied CB₁ receptor antagonist/inverse agonist. As such, SR141716A has been extensively used as a pharmacological tool to investigate the role of the endocannabinoid system. SR141716A remains highly relevant in experimental neuropharmacology, as it provides unique mechanistic insights into how CB₁ receptor blockade and/or inverse agonism affects constitutive Gi/o signaling—reshapes neuronal and circuit function (Howlett et al., 2002; Pertwee, 2005). By relieving CB₁-mediated inhibition of adenylyl cyclase, SR141716A increases cAMP/PKA activity and engages downstream nodes (ERK1/2, PI3K/Akt–GSK-3, PLC/PKC, CaMKII) and Ca²⁺ mobilization (LTCC/SOCE and IP₃R/RyR-dependent store release) (Howlett et al., 2002; Zhuang et al., 2005). At the synaptic level, it unmasks tonic endocannabinoid control over presynaptic glutamatergic and GABAergic release, revealing how CB₁ tone gates excitability and network gain (Castillo et al., 2012; Robison et al., 2016; Urbanski et al., 2010). At the systems level, it is a powerful tool for dissecting brain–gut-axis pathways: SR141716A-evoked signaling in the DVC (AP/NTS/DMNX) and enteric nervous system highlights how peripheral (vagal/ENS) and central mechanisms converge to drive emetic outputs (Babic and Browning, 2013; Travagli and Anselmi, 2016). Conceptually, comparing SR141716A with neutral CB₁ antagonists [e.g., AM4113, which do not evoke emesis (Salamone et al., 2007, PMID: 17521686)] as well as with peripherally restricted compounds (e.g., JD5037), helps to separate effects due to inverse agonism from simple receptor blockade and distinguishes central from peripheral CB₁ contributions (Sink et al., 2008; Tam et al., 2012).
In our study, its value lies in elucidating the mechanistic role of CB₁ receptor inverse agonism in emesis rather than in its clinical application. We hope that this clarification addresses the reviewer’s concern while also underscoring the continued scientific relevance of SR141716A as an experimental probe, despite its withdrawal from the clinic.
- Authors wrote: “SR141716A, by itself evokes emesis … and its 20 mg/kg dose causes vomiting in all tested shrews [14]. Likewise, a 20 mg dose of SR141716A causes significant emesis in humans [16].” However, one cannot compare directly a dose of 20 mg/kg in animals with a dose of 20 mg per person (weighing about 70 kg) in humans.
We agree and have revised the manuscript to avoid implying cross-species dose equivalence. Indeed, the effective dose of 20 mg/kg used in shrews cannot be directly compared to the that of humans. In the ‘dose-by- or’ approach, the ‘no-observed negative effect level’ (NOAEL) of the drug is scaled by using simple allometry based on body surface area to obtain the ‘human-equivalent dose’ (HED) (PMID: 22407287).
The formula enclosed below is used to calculate the HED according to the FDA guidelines:
HED (mg / kg = Animal NOAEL mg/kg) × (Weightanimal [kg]/Weighthuman [kg])(1–0.67) (PMID: 27057123)
Or simply:
HED mg/kg = Animal mg/kg dose x (animal weight in kg/ Human weight in kg)0.33
Of note, the FDA approach uses the exponent for body surface area, 0.67, to scale doses between species. This practice was normally rationalized as a means of accounting for differences in metabolic rate.
We applied the formula to convert the dose of 20 mg/kg effective dose in shrews to human.
HED= 20 x (0.005/60)0.33 = 0.9 mg/kg
With the mean weight of a shrew = (6 + 4)/2 = 5 g = 0.005 kg
With the average human body weight taken as 60 kg
In conclusion:
Equivalent HED to the 20 mg/ kg effective dose in shrews is (assuming the human weight to be 60 kg) is 0.9 x 60 mg/day= 54 mg
Clinical trials have used a wide range of SR141716A doses varying from 1 to 90 mg per day (e.g., PMID 16504646; PMID 17619859]. The intent of the current study was to note qualitative concordance (that SR141716A/rimonabant can be emetogenic across species), not to equate doses. We therefore limit our discussion to qualitative cross-species concordance and do not extrapolate dose equivalence in the introduction part. which further lengthens the manuscript, and we were asked to reduce the length of the manuscript following its first revision.
We have now revised the introduction section as “We have already shown that large doses (> 10 mg/kg) of SR141716A, by itself evokes emesis in least shrews in a dose-dependent manner [14, 15], and its 20 mg/kg dose causes vomiting in all tested shrews [14], which corresponds to a human equivalent dose of 54 mg assuming human weight of 60 Kg [16]. In clinical trials a diverse range of SR141716A doses (1 – 90 mg per day) have been utilized and a dose of 20 mg/day has been associated with a higher incidence of vomiting relative to placebo in randomized trials [17, 18].” Please see lines 60-66 in the manuscript.
- Authors have sent for the evaluation not final version of their manuscript. I had to read the version with numerous deletions!
We apologize for the confusion. As was required, we had kept Track Changes enabled during the previous revision so that the referee can clearly see that we had significantly reduced the length of the manuscript. We have now accepted track Changes. Please see the revised manuscript.
Reviewer 2 Report (New Reviewer)
Comments and Suggestions for Authors
In my last review I brought up the specificity of the CB1 antibody. The authors replied that "Several studies have used this antibody to detect CB1 receptor expression, including recent publications". This of course is no justification for the lack of controls. Most commercial antibodies are used in one study or another; however in most studies nobody bothers to look at the specificity of antibodies. These studies (PMIDs: 18406468, 34453219) investigated CB1 antibodies and found that most of them bind unspecific targets. Although ab3358 was not included, the studies used other rabbit polyclonals that employed a similar antigen for immunization. In fact, they stained brain tissue but just not CB1. Therefore, for CB1 validation of the antibody is required.
I do see the problem that time for revision is far too short, so some of my points can't be adressed.
Author Response
Referee 2
Comments and Suggestions for Authors
This study by Sun et al. presents an extensive investigation into the emetic mechanisms triggered by the cannabinoid CB1 receptor inverse agonist SR141716A in least shrews, revealing complex intracellular signaling pathways involved in vomiting. However, the study, while thorough, leans heavily on descriptive experimental data and would benefit from deeper mechanistic insights.
The authors systematically demonstrate that Rimonabant induces vomiting via both central and peripheral mechanisms, chiefly the dorsal vagal complex (DVC) in the brainstem and the enteric nervous system in the jejunum, with evidence from c-fos and phospho-ERK1/2 immunfluorescence, among others. They provide a link between brainstem cAMP levels and sequential phosphorylation of key kinases such as PKA, Akt, GSK-3α/β, ERK1/2, and PKCαβII to the emetic response. The broad-spectrum pharmacological blockade studies targeting these kinases and various calcium channels strongly support their involvement in SR141716A-evoked vomiting.
Critical Issues:
- Antibody Validity for CB1 Detection and general antibody-related issues: A key weakness undermining the interpretation of CB1 receptor distribution is the use of a commercial anti-CB1 antibody (ab3558, Abcam) which has been withdrawn from the market by the company. This raises questions regarding the specificity and reliability of CB1 receptor immunostaining data presented. In fact, previous studies (Histochem Cell Biol. 2021 Aug 27;156(5):479–502, J Neurosci Methods. 2008 Jun 15;171(1):78-8) investigated CB1 antibody specificity and found that many CB1 antibodies do not specifically detect the protein. Staining specificity of the antibody used should be either verified by using another CB1 antibody, another technique (Western Blot) or by the use of knockout animals (in this case not possible). In Fig. 1 CB1 staining is shown. However, staining doesn’t look specific since I only see a blurry reddish image. Maybe supplying a magnification could shed some light whether the antibody stains specifically. A quantification of only those cells that are labelled positive is not a very good method since this leaves out cells with low level expression. Rather, staining intensity should be considered for quantification. In addition, I would like to see an isotype staining, especially as a control for CB1.
We agree that CB1 antibodies vary in performance. In our study, CB1 immunostaining was used only to provide regional context (AP/NTS/DMNX mapping) and to note that acute SR141716A did not alter regional CB1 density; our key conclusions (↑cAMP/PKA, engagement of ERK/PI3K–Akt–GSK-3/PLC–PKC/CaMKII, Ca²⁺ mobilization, and inhibitor sensitivity of emesis) do not depend on subtle intensity gradients of CB1 immunoreactivity.
We acknowledge that ab3558 has been discontinued by the vendor; however, discontinuation alone is not evidence of non-specificity. Several studies have used this antibody to detect CB1 receptor expression, including recent publications (Shen et al., 2010; PMID: 20680167; Wang et al., 2022; PMID: 35269905; Sevil et al., 2024; PMID: 39758842; Köse et al., 2024; PMID: 39354544). Given the 10-day revision window, it is not feasible to obtain and validate an alternative antibody and repeat these experiments. Instead of re-performing this component with a different antibody, we strengthened validation and presentation of the existing dataset as follows:
- Negative controls: Secondary-only and host-matched rabbit IgG isotype controls were processed in parallel and imaged with identical acquisition parameters to quantify background; these data are now included. Please see attached supplemental material.
- High-magnification imaging: As noted in Section 4.4.1, tile-scan images were acquired on a Zeiss LSM 880 confocal microscope (1024 × 1024 pixels; Plan-Apochromat 63×/1.4 Oil DIC M27 objective). We now added high-magnification insets in Fig. 1E–G from the boxed regions in panel C (AP, NTS, DMNX).
- Re-analysis: CB1 signal was quantified as mean gray value per µm² within predefined ROIs.
- c-fos immunoreactivity seems to be highest outside the areas of interest. Why? Do those cells carry CB1? Why didn’t the authors co-stain with CB1?
The c-fos panels are tile scans that necessarily include tissue adjacent to the dorsal vagal complex (DVC). However, our quantification was restricted to the prespecified emetic nuclei—AP, NTS, and DMNX. The DVC contains not only neurons that express CB1 receptors but also other neuronal types (e.g., glutamatergic and GABAergic), and its nuclei have complex connections with other brain regions, such as the hypothalamus, amygdala, cortex, and so on. In this study, drug administration was systemic (via the circulation) rather than a local injection into the DVC. The goal of this study was therefore to evaluate the overall activation of the DVC after SR141716A administration rather than cellular co-localization. Functionally, SR141716A-evoked emesis is multideterminant (involving cAMP/PKA, ERK1/2, PI3K/Akt/GSK-3, PLC/PKC, CaMKII, and the classical 5-HT3 and D2/3 receptors), so co-localizing c-fos with CB1 would not, by itself, distinguish activation of CB1-expressing neurons from downstream network activation.
- cAMP levels: A positive control would have helped to put these values into perspective. The authors mentioned that rolipram has a similar effect. Why not use this as positive control?
We thank the reviewer for this insightful suggestion. In the present study, we observed a significant increase in cAMP levels following SR141716A administration, consistent with its established role as a CB₁ inverse agonist that disinhibits adenylate cyclase activity. Although rolipram, a phosphodiesterase-4 inhibitor, is known to elevate intracellular cAMP, we did not include it as a positive control here because we have previously reported that rolipram significantly increases brainstem cAMP in the least shrew (Alkam et al., 2014-PMID: 24513510), which supports our present findings. Nevertheless, we agree that such a control would strengthen the interpretation, and we will incorporate rolipram or other reference modulators in future studies to provide a clearer comparative context for cAMP signaling.
- The study overwhelmingly shows descriptive data without explaining the precise molecular interactions between SR141716A and intracellular effectors. The complexity of GPCR biased signaling and the effects of inverse agonism by rimonabant is acknowledged but insufficiently explored, missing an opportunity to provide insight into why SR141716A acts as an inverse agonist triggering emesis while agonists generally confer antiemetic effects. Are these pathways that were investigated all directly linked to CB1 inverse agonism? Would the compounds used also inhibit vomiting induced by a different mechanism? This would be interesting, because then the observed effects would not be solely attributable to CB1 inhibition. This also leads me to another question: Is the activation of the described intracellular pathways by rimonabant a CB1 specific mechanism or is this a general activation scheme common to all emetic compounds?
In this study, we have demonstrated that SR141716A (20 mg/kg, i.p.) produced a time-dependent increase in brainstem cAMP (↑ at 15 and 30 min), a peak in PKA (Thr197) phosphorylation at 30 min, and rapid phosphorylation of ERK1/2, Akt, GSK-3α/β, and PKCα/βII (peaking at ~15 min and persisting for some nodes). These biochemical changes coincided with the observed emesis and DVC activation (↑c-fos; ↑p-ERK1/2), establishing temporal linkage between receptor perturbation and effector pathways. Functionally, pharmacologic inhibition of these emetic signals (PKA, ERK1/2, GSK-3, PI3K, PLC, PKC, CaMKII; membrane Ca²⁺ entry via LTCC and SOCE; intracellular Ca²⁺ release via IP₃R and RyR) significantly reduced SR141716A-evoked vomiting in a dose-dependent manner, indicating these pathways are necessary for the behavioral output under our conditions.
CB₁ receptor predominantly couples to Gi/o to suppress adenylyl cyclase. The observed ↑cAMP/↑PKA after SR141716A is the expected direct consequence of inverse agonism (relief of constitutive Gi/o tone). Also, we found no acute change in CB1 protein density in DVC after SR141716A, arguing that the signaling pattern reflects a change in receptor signaling state rather than receptor density. Engagement of ERK, PI3K/Akt/GSK-3, PLC/PKC, CaMKII, and Ca²⁺ mobilization (LTCC/SOCE and IP₃R/RyR) is consistent with well-known cross-talk downstream of cAMP/PKA in emetic nuclei. In short, cAMP/PKA is a direct readout of CB1 inverse agonism, and the other activated intracellular emetic signals represent downstream activation within the emetic network. Our data show that blocking any of these signals attenuates emesis, so they are causally involved even if not all are “direct” CB1 outputs in every neuron.
In our previous studies, we have demonstrated that the tested compounds used in this study also inhibited shrews vomiting induced by diverse emetogens. In this study, we show that classical antiemetics acting at 5-HT₃ (palonosetron) and D₂/₃ (sulpiride) receptors attenuate SR141716A-evoked vomiting, and that Ca²⁺-channel blockers (LTCC, SOCE, IP₃R, RyR) and kinase inhibitors (PKA, ERK, GSK-3, PI3K, PLC, PKC, CaMKII) are effective against SR141716A-evoked vomiting. Because several of these drug classes are known to act across multiple emetic triggers, their efficacy here indicates that SR141716A engages shared emetic machinery; i.e., the downstream effects are not solely attributable to CB1 blockade per se, but though CB1 inverse agonism is the initiating event in our paradigm.
Our data indicate that SR141716A-induced vomiting is initiated by a CB₁ receptor–specific mechanism. Specifically: (1) Upstream (CB₁-specific): SR141716A’s inverse agonism at CB₁ relieves Gi/o-mediated restraint on adenylyl cyclase, increasing cAMP/PKA and initiating the intracellular cascade that culminates in emesis; (2) Downstream (general/convergent): the execution layer—ERK, PI3K/Akt–GSK-3, PLC/PKC, CaMKII, and coordinated Ca²⁺ entry/release—constitutes a general emetic program engaged by multiple triggers; accordingly, blocking these signals inhibits SR141716A-evoked vomiting.
In summary, SR141716A triggers emesis via CB1 inverse agonism (CB1-specific initiation), but the response is implemented by shared intracellular effectors (general convergence). Our time-course biochemistry plus pathway-targeted inhibition convert the findings from descriptive to mechanistically causal for the emetic phenotype studied here.
- Although multiple pathways are implicated, the study does not sufficiently address alternative or compensatory signaling cascades that may contribute to the emetic response. For example, the lack of increase in endogenous serotonin (5-HT) and substance P (SP) tissue levels contradicts the findings that serotonin and dopamine receptor antagonists are efficacious in the described set-up. This discrepancy merits closer examination, possibly involving release kinetics, metabolism, or receptor sensitivity. Same for p-ERK: ERK would also be activated by CB1 agonists, which are usually anti-emetic. Please elaborate. Similarly, cannabis hyperemesis syndrome is caused by CB1 agonists. This is one major side effect in some patients and since it can be attenuated by TRPV1 agonists, this should be discussed here as well.
1) We agree that although SR141716A treatment did not increase tissue levels of serotonin (5-HT) or substance P (SP) in the dorsal vagal complex, the efficacy of 5-HT₃ and D₂ receptor antagonists in attenuating SR141716A-induced emesis suggests that receptor-level sensitivity, release dynamics, or neurotransmitter turnover may play a more critical role than static tissue levels.
We appreciate the referee’s suggestion, and in our re-revised manuscript we have added the following discussion in the manuscript, “iv) the release dynamics, or receptor-level sensitivity may also play a more critical role than static tissue levels,” Please see lines 846-847.
2) p-ERK1/2 functions as a convergent integrator of multiple inputs; therefore, the behavioral outcome depends on the net effect at the circuit level. CB1 agonists generally reduce presynaptic glutamate/5-HT/GABA release (Gi/o tone), producing an overall antiemetic network effect, even if ERK1/2 is transiently activated (e.g., via β-arrestin pathways). SR141716A, as a CB1 inverse agonist, disinhibits AC→cAMP/PKA, permitting broader Ca²⁺ mobilization (LTCC/SOCE/IP₃R/RyR) and ERK/PI3K–Akt–GSK-3/PLC–PKC/CaMKII engagement that together promote emesis.
3) Per referee 2’s suggestion, we have added the following discussion in the manuscript, “Cannabis hyperemesis syndrome (CHS) occurs following chronic administration of large doses of CB₁ agonists, further illustrates the complexity of endocannabinoid signaling. Interestingly, TRPV1R agonists such as capsaicin provide relief in CHS patients, supporting the idea that TRPV1R-CB₁ crosstalk and compensatory adaptations of intracellular pathways are critical determinants of the emetic phenotype [128, 129]. In fact, we have demonstrated that a combination of CB1- and TRPV1 receptor agonists have the capacity to completely abolish cisplatin-evoked emesis at doses that are ineffective when used individually [121].” Please see lines 776-783.
- The peripheral involvement has only been shown indirectly through stainings in the ENS. However, whether those changes are emetic has not been shown. This can be shown by vagotomy but all evidence shown here is indirect.
The DVC is the canonical integrator of vagal afferents driving emesis, so concurrent ENS and DVC activation is expected if the brain–gut axis is engaged. ENS activity markers rise in the same time window as vomiting and DVC (AP/NTS/DMNX) activation, compatible with a gut → vagus → DVC pathway. Our ENS staining together with DVC activation supports involvement of the brain–gut axis during SR141716A-evoked emesis. Accordingly, in future studies we will perform subdiaphragmatic vagotomy or reversible perivagal blockade to determine the involvement of the entire brain–gut axis in the induced emetic response. We thank you for your suggestion!
- Translational and Clinical Implications are vague: The study's focus on mechanistic dissection in the least shrew model does not adequately bridge to clinical implications for cannabinoid-based antiemetic therapy or adverse emetic effects associated with CB1 inhibitors. More explicit discussion on how these findings may inform drug design or treatment of nausea and vomiting in patients would increase relevance.
While this study was conducted in the least shrew, a well-validated model for emesis research, the mechanistic findings have potential translational relevance for human conditions. The ability of SR141716A to trigger vomiting through CB₁ inverse agonism underscores the critical role of endocannabinoid signaling in emetic regulation and helps explain why SR141716A was associated with adverse gastrointestinal and neuropsychiatric effects in the clinic. Furthermore, our results suggest that targeting downstream effectors such as ERK1/2, PI3K/Akt, and calcium signaling may provide alternative strategies for antiemetic drug development beyond conventional 5-HT₃ and dopamine receptor antagonists. The observed overlap with signaling pathways implicated in CHS also highlights the translational relevance of our findings, as they may help inform management strategies for patients suffering from this increasingly recognized condition. Overall, while cannabis cessation remains the definitive treatment for CHS, our data suggest that pharmacological modulation of convergent signaling cascades may hold promise as adjunctive therapies. We cannot provide further details since we were asked to reduce the length of the original manuscript.
Round 3
Reviewer 1 Report (New Reviewer)
Comments and Suggestions for Authors
Unfortunately, I'm not fully satisfied with the Authors’ response. I agree with the Authors that rimonabant is still applied in various experiments as CB₁ receptor antagonist/inverse agonist even after its withdrawal from the market in 2008. However:
- its use has been steadily declining; according to PubMed 265, 189, 98 and 45 publications with rimonabant in 2007, 2010, 2015 and 2024, respectively and 283, 200, 102 and 49 publications with SR141716A in 2007, 2010, 2015 and 2024, respectively were published;
- The Authors examined a very high dose of SR141716A (rimonabant) that stimulates vomiting 20 mg/kg (given intraperitoneally; i.p.); such a large dose is not typically used in experiments where rimonabant (i.p) is used as CB₁ receptor antagonist/inverse agonist. Even the Authors’ group demonstrated previously on shrews that SR141716A (rimonabant) reversed the antiemetic effects of cannabinoid receptor agonist WIN 55, 212-2 against cisplatin-induced emesis with an ID (i.e. the inhibitory dose that prevented emesis in 50% of shrews) of 0.27+/-1.56 mg/kg (Darmani NA. The cannabinoid CB1 receptor antagonist SR 141716A reverses the antiemetic and motor depressant actions of WIN 55, 212-2. Eur J Pharmacol.;430(1):49-58)
Below additional examples of the use of SR141716A (rimonabant; i.p.) as a pharmacological tool are given:
- 1.0 mg/kg−1 SR141716: O'Brien LD, Limebeer CL, Rock EM, Bottegoni G, Piomelli D, Parker LA. Anandamide transport inhibition by ARN272 attenuates nausea-induced behaviour in rats, and vomiting in shrews (Suncus murinus). Br J Pharmacol. 2013 Nov;170(5):1130-6. doi: 10.1111/bph.12360;
why was this publication not quoted by the Authors?
- rimonabant (3, 10 mg/kg), Soler-Cedeño O, et al. AM6527, a neutral CB1 receptor antagonist, suppresses opioid taking and seeking, as well as cocaine seeking in rodents ;without aversive effects. Neuropsychopharmacology. 2024;49(11):1678-1688;
- rimonabant (1 mg/kg); Li et al. Resting State Brain Networks under Inverse Agonist versus Complete Knockout of the Cannabinoid Receptor 1. ACS Chem Neurosci. 2024;15(8):1669-1683;
- rimonabant (10 mg/kg); Jiang et al. A monoacylglycerol lipase inhibitor showing therapeutic efficacy in mice without central side effects or dependence. Nat Commun. 2023;14(1):8039;
- rimonabant (5 mg/kg); Cao et al. Microglial adenosine A2A receptor in the paraventricular thalamic nucleus regulates pain sensation and analgesic effects independent of opioid and cannabinoid receptors. Front Pharmacol. 2024 Dec 19;15:1467305.
In conclusion, Authors did not prove sufficiently the significance their manuscript. Moreover, they failed to discuss the significance of their results.
This manuscript is a resubmission of an earlier submission. The following is a list of the peer review reports and author responses from that submission.
Round 1
Reviewer 1 Report
Comments and Suggestions for Authors
The study explores the emetic mechanisms of SR141716A in the least shrew model, with a focus on intracellular Ca²⁺ signaling and the involvement of classical emetic neurotransmitter pathways, and demonstrates :
- SR141716A-induced vomiting involves both extracellular calcium influx and intracellular calcium release (via IP₃R and RyR).
- Calcium-modulating agents (2-APB and dantrolene) suppress vomiting.
- Classical receptor antagonists (palonosetron for 5-HT₃ and sulpiride for D₂/₃ receptors) attenuate SR141716A-evoked vomiting.
This suggests involvement of serotonin, dopamine, and possibly substance P (NK1) pathways in the emetic effect of SR141716A.
The study indeed adds in vivo evidence to the understanding of how SR141716A induces emesis by implicating both intracellular and extracellular Ca²⁺ pathways. Using 2-APB and dantrolene, the authors further dissect the calcium-dependent signaling. Although it is relevant to clinical Antiemesis (investigating 5-HT₃ and D₂/₃ receptor involvement aligns with antiemetic targets of clinically relevant drugs like palonosetron and sulpiride) and clearly shows a translational potential (understanding of emesis induced by CBR-1 modulation) for the clinical cannabinoid-based therapeutics, the majority of the study currently lacks an apparent mechanistic depth on SR141716A-induced emesis. While it's mentioned that SR141716A may increase intracellular Ca²⁺, the mechanism of this effect is not explored in detail. Is it via direct receptor action, second messengers, or disruption of ER calcium homeostasis?
Similarly, no direct measurement of neurotransmitters is linked to the observed effect. The claim that SR141716A promotes the release of 5-HT, DA, or SP is based on previous work, not directly demonstrated here. In many cases, there is an overreliance on Previous References without fully incorporating new, independent data to support or expand those conclusions. Therefore, I do not see any significant novelty in the current version of the study.
Suggestions for improvements
The results would benefit from more detailed quantitative data, such as exact vomiting counts or statistical comparisons, and a better description of the behavioral studies.
Perform calcium imaging or biosensor assays to observe intracellular calcium changes following SR141716A treatment directly.
Explore combination therapy.
Author Response
We thank both referees for their comprehensive review, time, effort, and constructive suggestions. When possible, we have fully attended to the proposed suggestions in the appropriate sections of the revised manuscript. We believe these changes have further increased the quality of our manuscript. Below we answer each concern/suggestion point-by-point in blue color as they appear in the critique:
Reviewer 1
Comments and Suggestions for Authors
The study explores the emetic mechanisms of SR141716A in the least shrew model, with a focus on intracellular Ca²⁺ signaling and the involvement of classical emetic neurotransmitter pathways, and demonstrates :
- SR141716A-induced vomiting involves both extracellular calcium influx and intracellular calcium release (via IP₃R and RyR).
- Calcium-modulating agents (2-APB and dantrolene) suppress vomiting.
- Classical receptor antagonists (palonosetron for 5-HT₃ and sulpiride for D₂/₃ receptors) attenuate SR141716A-evoked vomiting.
This suggests involvement of serotonin, dopamine, and possibly substance P (NK1) pathways in the emetic effect of SR141716A.
- The study indeed adds in vivoevidence to the understanding of how SR141716A induces emesis by implicating both intracellular and extracellular Ca²⁺ pathways. Using 2-APB and dantrolene, the authors further dissect the calcium-dependent signaling. Although it is relevant to clinical Antiemesis (investigating 5-HT₃ and D₂/₃ receptor involvement aligns with antiemetic targets of clinically relevant drugs like palonosetron and sulpiride) and clearly shows a translational potential (understanding of emesis induced by CBR-1 modulation) for the clinical cannabinoid-based therapeutics, the majority of the study currently lacks an apparent mechanistic depth on SR141716A-induced emesis. While it's mentioned that SR141716A may increase intracellular Ca²⁺, the mechanism of this effect is not explored in detail. Is it via direct receptor action, second messengers, or disruption of ER calcium homeostasis?
We agree with the reviewer # 1 in that the current study we have provided evidence for both extracellular Ca²⁺ entry (via L-type Ca²⁺ channels, SOCE, and TRPV1 receptors) and intracellular Ca²⁺ release (via IP₃Rs and RyRs), using pharmacological blockers such as nifedipine, MRS-1845, 2-APB, and dantrolene. The attained results collectively suggest that SR141716A-induced emesis involves Ca²⁺ mobilization through both plasma membrane channels and endoplasmic reticulum stores. However, this is only one part of our new findings since we also have described the unique inverse agonist action of SR141716A which causes adenylate cyclase activation and production of cAMP in the brainstem that leads to SR141716A-evoked vomiting. This finding supports the well-established phenomenon that phosphodiesterase inhibition which prevents cAMP metabolism also leads to increased cAMP tissue levels in the brainstem which subsequently causes emesis (e.g., Robichaud et al., 1999; Alkam et al., 2014). Furthermore, our current new findings detail phosphorylation changes in key downstream intracellular enetic signaling proteins (e.g., cAMP/PKA, ERK1/2, PI3K/Akt/GSK-3, PLC/PKCαβII, and CaMKII) which strongly support second-messenger-mediated mechanisms linked to Ca²⁺ signaling.
We appreciate the reviewer’s suggestion and agree that deeper mechanistic studies are needed to fully delineate how SR141716A modulates intracellular Ca²⁺ dynamics which are currently being planned as part of our future research efforts. Our future studies will be focused on specific aspects of current results which are currently presented in 10 figures containing 89 subparts.
- Similarly, no direct measurement of neurotransmitters is linked to the observed effect. The claim that SR141716A promotes the release of 5-HT, DA, or SP is based on previous work, not directly demonstrated here. In many cases, there is an overreliance on Previous References without fully incorporating new, independent data to support or expand those conclusions. Therefore, I do not see any significant novelty in the current version of the study.
In this study, we performed immunohistochemical analysis to assess the release of 5-HT and substance P (SP) following SR141716A (20 mg/kg, i.p.)-induced vomiting. Please refer to Sections 2.2.4 and 4.4.5, as well as Figure 5. This approach is based on our previous findings, which have shown that several other emetogens such as cisplatin-, FPL64176-, or thapsigargin-induced vomiting are accompanied by increased 5-HT or SP immunoreactivity in the brainstem emetic nuclei (Andrews et al., 1998; Zhong et al., 2016; Zhong et al., 2018).
In the current study, figure 5A shows that highest density of 5-HT-positive fibers are in the dorsomedial subdivision of the NTS. A lower density of 5-HT-immunoreactive profile is noted in the adjacent subnuclei of NTS and DMNX, and the AP had fewer 5-HT-containg neurons in vehicle-treated control group. Figure 5B shows that SP-immunoreactive fibers were found in highest concentration within the DMNX and to a lesser extent in the NTS, but rarely in the AP of vehicle-treated control group. However, SR141716A failed to increase 5-HT or SP immunoreactivities in either the AP, NTS or DMNX nuclei of the shrew brainstem DVC at 15-min post injection. These results are further discussed in Section 3.4. It is also relevant to note that in our original manuscript we had stated that both cannabinoid CB1 receptor agonists (inhibit neurotransmitter release) or the inverse agonist SR141716A (promotes transmitter release) produce “regionally-specific-effects” in neurotransmitter levels in different brain regions measured by the HPLC technique but not in the brainstem (Darmani et al., 2003; Cadogan et al., 1997; Tzavara, 2001)). Thus, we have not over relied on our or other investigators’ published reports that SR141716A causes release of emetic monoamine neurotransmitters in other brain regions but do confirm that in agreement with available literature SR141716A does not specifically increase such emetic neurotransmitter release in the brainstem (Please read lines 849-856 of our revised manuscript). Furthermore, this is unlike the situation with cisplatin which not only increases tissue levels of emetic neurotransmitters such as serotonin, dopamine and substance P in the brainstem emetic nuclei but also their turnover during vomiting (Darmani et al., Brain Research 2009 12: 1248-1258).
Suggestions for improvements
- The results would benefit from more detailed quantitative data, such as exact vomiting counts or statistical comparisons, and a better description of the behavioral studies.
In this study, we have fully described in its method section that each shrew was placed in the observation cage after injection of SR141716A and the number of vomiting was counted for the next 30 min. Also, as we described in section 4.7, the vomit frequency data were analyzed using the Kruskal-Wallis non-parametric one-way analysis of variance (ANOVA) followed by post hoc analysis by Dunn’s multiple comparisons test and expressed as the mean ± SEM. The percentage of animals vomiting across groups at different doses was compared using the Chi-square test. In addition, figures 7-10 show the frequency of emesis and percentage of shrews vomiting after different treatments in detail. Thus, we are not sure why this question was raised? In fact, reviewer 2 has appreciated the detailed description of our methodology and results.
- Perform calcium imaging or biosensor assays to observe intracellular calcium changes following SR141716A treatment directly.
Thank you for your suggestion regarding intracellular calcium measurements. In future experiments we plan to dynamically monitor intracellular Ca²⁺ concentration changes in neurons within the shrew DVC following treatment with various emetogens as we have done previously (Hutchinson et al., 2015; Zhong et al., 2014). As was discussed above, this manuscript already contains a lot of new information.
- Explore combination therapy.
Combination therapy is widely used in clinical practice. In our previous studies, we have explored the combination therapy in various emetogen-evoked emesis model, such as: i) the combined use of selective PKCαβII inhibitor GF109203X and ERK1/2 inhibitor U0126 against emesis induced by the D2 receptor agonist quinpirole (2 mg/kg, i.p.; Belkacemi et al., 2021); ii) a combination of low doses of the L-type Ca²⁺ channel blocker amlodipine and the ryanodine receptor antagonist dantrolene to counteract emesis induced by the selective 5-HT3 receptor agonist 2-Me-5-HT (5 mg/kg, i.p.; Zhong et al., 2014).
In this study, we have included multiple anti-emetics to exam their effects on SR141716A-induced vomiting. Due to capacity constraints in a single manuscript, combination therapy against SR141716A-induced emesis will be explored in future studies. Thank you for this valuable suggestion
We thank both referees for their comprehensive review, time, effort, and constructive suggestions. When possible, we have fully attended to the proposed suggestions in the appropriate sections of the revised manuscript. We believe these changes have further increased the quality of our manuscript. Below we answer each concern/suggestion point-by-point in blue color as they appear in the critique:
Reviewer 1
Comments and Suggestions for Authors
The study explores the emetic mechanisms of SR141716A in the least shrew model, with a focus on intracellular Ca²⁺ signaling and the involvement of classical emetic neurotransmitter pathways, and demonstrates :
- SR141716A-induced vomiting involves both extracellular calcium influx and intracellular calcium release (via IP₃R and RyR).
- Calcium-modulating agents (2-APB and dantrolene) suppress vomiting.
- Classical receptor antagonists (palonosetron for 5-HT₃ and sulpiride for D₂/₃ receptors) attenuate SR141716A-evoked vomiting.
This suggests involvement of serotonin, dopamine, and possibly substance P (NK1) pathways in the emetic effect of SR141716A.
- The study indeed adds in vivoevidence to the understanding of how SR141716A induces emesis by implicating both intracellular and extracellular Ca²⁺ pathways. Using 2-APB and dantrolene, the authors further dissect the calcium-dependent signaling. Although it is relevant to clinical Antiemesis (investigating 5-HT₃ and D₂/₃ receptor involvement aligns with antiemetic targets of clinically relevant drugs like palonosetron and sulpiride) and clearly shows a translational potential (understanding of emesis induced by CBR-1 modulation) for the clinical cannabinoid-based therapeutics, the majority of the study currently lacks an apparent mechanistic depth on SR141716A-induced emesis. While it's mentioned that SR141716A may increase intracellular Ca²⁺, the mechanism of this effect is not explored in detail. Is it via direct receptor action, second messengers, or disruption of ER calcium homeostasis?
We agree with the reviewer # 1 in that the current study we have provided evidence for both extracellular Ca²⁺ entry (via L-type Ca²⁺ channels, SOCE, and TRPV1 receptors) and intracellular Ca²⁺ release (via IP₃Rs and RyRs), using pharmacological blockers such as nifedipine, MRS-1845, 2-APB, and dantrolene. The attained results collectively suggest that SR141716A-induced emesis involves Ca²⁺ mobilization through both plasma membrane channels and endoplasmic reticulum stores. However, this is only one part of our new findings since we also have described the unique inverse agonist action of SR141716A which causes adenylate cyclase activation and production of cAMP in the brainstem that leads to SR141716A-evoked vomiting. This finding supports the well-established phenomenon that phosphodiesterase inhibition which prevents cAMP metabolism also leads to increased cAMP tissue levels in the brainstem which subsequently causes emesis (e.g., Robichaud et al., 1999; Alkam et al., 2014). Furthermore, our current new findings detail phosphorylation changes in key downstream intracellular enetic signaling proteins (e.g., cAMP/PKA, ERK1/2, PI3K/Akt/GSK-3, PLC/PKCαβII, and CaMKII) which strongly support second-messenger-mediated mechanisms linked to Ca²⁺ signaling.
We appreciate the reviewer’s suggestion and agree that deeper mechanistic studies are needed to fully delineate how SR141716A modulates intracellular Ca²⁺ dynamics which are currently being planned as part of our future research efforts. Our future studies will be focused on specific aspects of current results which are currently presented in 10 figures containing 89 subparts.
- Similarly, no direct measurement of neurotransmitters is linked to the observed effect. The claim that SR141716A promotes the release of 5-HT, DA, or SP is based on previous work, not directly demonstrated here. In many cases, there is an overreliance on Previous References without fully incorporating new, independent data to support or expand those conclusions. Therefore, I do not see any significant novelty in the current version of the study.
In this study, we performed immunohistochemical analysis to assess the release of 5-HT and substance P (SP) following SR141716A (20 mg/kg, i.p.)-induced vomiting. Please refer to Sections 2.2.4 and 4.4.5, as well as Figure 5. This approach is based on our previous findings, which have shown that several other emetogens such as cisplatin-, FPL64176-, or thapsigargin-induced vomiting are accompanied by increased 5-HT or SP immunoreactivity in the brainstem emetic nuclei (Andrews et al., 1998; Zhong et al., 2016; Zhong et al., 2018).
In the current study, figure 5A shows that highest density of 5-HT-positive fibers are in the dorsomedial subdivision of the NTS. A lower density of 5-HT-immunoreactive profile is noted in the adjacent subnuclei of NTS and DMNX, and the AP had fewer 5-HT-containg neurons in vehicle-treated control group. Figure 5B shows that SP-immunoreactive fibers were found in highest concentration within the DMNX and to a lesser extent in the NTS, but rarely in the AP of vehicle-treated control group. However, SR141716A failed to increase 5-HT or SP immunoreactivities in either the AP, NTS or DMNX nuclei of the shrew brainstem DVC at 15-min post injection. These results are further discussed in Section 3.4. It is also relevant to note that in our original manuscript we had stated that both cannabinoid CB1 receptor agonists (inhibit neurotransmitter release) or the inverse agonist SR141716A (promotes transmitter release) produce “regionally-specific-effects” in neurotransmitter levels in different brain regions measured by the HPLC technique but not in the brainstem (Darmani et al., 2003; Cadogan et al., 1997; Tzavara, 2001)). Thus, we have not over relied on our or other investigators’ published reports that SR141716A causes release of emetic monoamine neurotransmitters in other brain regions but do confirm that in agreement with available literature SR141716A does not specifically increase such emetic neurotransmitter release in the brainstem (Please read lines 849-856 of our revised manuscript). Furthermore, this is unlike the situation with cisplatin which not only increases tissue levels of emetic neurotransmitters such as serotonin, dopamine and substance P in the brainstem emetic nuclei but also their turnover during vomiting (Darmani et al., Brain Research 2009 12: 1248-1258).
Suggestions for improvements
- The results would benefit from more detailed quantitative data, such as exact vomiting counts or statistical comparisons, and a better description of the behavioral studies.
In this study, we have fully described in its method section that each shrew was placed in the observation cage after injection of SR141716A and the number of vomiting was counted for the next 30 min. Also, as we described in section 4.7, the vomit frequency data were analyzed using the Kruskal-Wallis non-parametric one-way analysis of variance (ANOVA) followed by post hoc analysis by Dunn’s multiple comparisons test and expressed as the mean ± SEM. The percentage of animals vomiting across groups at different doses was compared using the Chi-square test. In addition, figures 7-10 show the frequency of emesis and percentage of shrews vomiting after different treatments in detail. Thus, we are not sure why this question was raised? In fact, reviewer 2 has appreciated the detailed description of our methodology and results.
- Perform calcium imaging or biosensor assays to observe intracellular calcium changes following SR141716A treatment directly.
Thank you for your suggestion regarding intracellular calcium measurements. In future experiments we plan to dynamically monitor intracellular Ca²⁺ concentration changes in neurons within the shrew DVC following treatment with various emetogens as we have done previously (Hutchinson et al., 2015; Zhong et al., 2014). As was discussed above, this manuscript already contains a lot of new information.
- Explore combination therapy.
Combination therapy is widely used in clinical practice. In our previous studies, we have explored the combination therapy in various emetogen-evoked emesis model, such as: i) the combined use of selective PKCαβII inhibitor GF109203X and ERK1/2 inhibitor U0126 against emesis induced by the D2 receptor agonist quinpirole (2 mg/kg, i.p.; Belkacemi et al., 2021); ii) a combination of low doses of the L-type Ca²⁺ channel blocker amlodipine and the ryanodine receptor antagonist dantrolene to counteract emesis induced by the selective 5-HT3 receptor agonist 2-Me-5-HT (5 mg/kg, i.p.; Zhong et al., 2014).
In this study, we have included multiple anti-emetics to exam their effects on SR141716A-induced vomiting. Due to capacity constraints in a single manuscript, combination therapy against SR141716A-induced emesis will be explored in future studies. Thank you for this valuable suggestion.
Reviewer 2 Report
Comments and Suggestions for Authors
The manuscript covers mechanistic studies of the emetogen SR141716A, presumably acting via the CB1 receptor. The experiments are described in detail (although I am not familiar with the specificity of some Ab like anti-5-HT3-Ab), the results are consistent. The discussion is long (the whole manuscript is long, but for results and methods this is appropriate), and should be shortened and focused.
A few notes of caution:
-
In the results, 2.2.1 contains the expression levels of cfos and NeuN. For these experiments, no time difference between agonist application and sacrificing the mice is given. For clarity, this should be included (although I do not expect de novo synthesis to increase within 15 min).
-
I wonder why THC has not been used in the experiments, since THC is the prototypical cannabinoid, and the classical CB1 agonist. Adding this compound (at least for behavioural studies) would be welcome; at least in Europe, THC as solution for inhalation is licensed for postoperative nausea and vomiting (PONV), and would further confirm the experimental design and conclusions.
-
The manuscript would also gain from inclusion of a control region for the immunhistochemistry; even though I cannot see stains in adjacent areas in most pictures, I also sometimes cannot see stains in control animals, but find a value >0 in the Figures and results. Ideally, for control regions a brain area with no connection to the emesis reflex should be chosen.
The discussion should be shortened, and possibly focused on larger pathway segments. For me, protein synthesis, protein phosphorylation and Ca flux/release would be three relevant functions; as far as I see the results nicely confirm in short term applications no change in protein amount, but larger differences in phosphorylation, and consecutive changes in Ca flow.
The text should be checked for mistakes; sometimes a singular (is) is found, whereas a plural (are) should have been used. Please capitalize ELISA
Author Response
We thank both referees for their comprehensive review, time, effort, and constructive suggestions. When possible, we have fully attended to the proposed suggestions in the appropriate sections of the revised manuscript. We believe these changes have further increased the quality of our manuscript. Below we answer each concern/suggestion point-by-point in blue color as they appear in the critique:
Reviewer 2
Comments and Suggestions for Authors
- The manuscript covers mechanistic studies of the emetogen SR141716A, presumably acting via the CB1 receptor. The experiments are described in detail (although I am not familiar with the specificity of some Ab like anti-5-HT3-Ab), the results are consistent. The discussion is long (the whole manuscript is long, but for results and methods this is appropriate), and should be shortened and focused.
We thank the reviewer for the thoughtful, supportive and constructive comments. Furthermore, we appreciate his/her acknowledgment of our detailed experimental design and consistent results.
Regarding the suggestion to shorten and focus the Discussion section: while we understand the value of concise presentation, we respectfully believe that the current length and structure of the Discussion are necessary to comprehensively interpret the broad scope of mechanistic findings in this study. Given the complexity of the signaling pathways examined and the integration of multiple levels of analysis (behavioral, immunohistochemical, pharmacological), we feel that a more condensed version might compromise clarity and completeness. Furthermore, we have written a long piece in the discussion section regarding the concept of “inverse-agonism” with out which many reviewers would be confused to say the least or turn to other published literature to grasp the meaning of it. Although the concept of inverse- agonism was proposed during the era of benzodiazepine (e.g. valium) development, only recently it has become a topic of discussion in the clinical world.
Therefore, we respectfully request to retain the current version of the Discussion, which we believe is appropriate given the depth and novelty of the mechanistic insights presented.
A few notes of caution:
- In the results, 2.2.1 contains the expression levels of c-fos and NeuN. For these experiments, no time difference between agonist application and sacrificing the mice is given. For clarity, this should be included (although I do not expect de novo synthesis to increase within 15 min).
The expression of c-fos protein exhibited a transient time course peaking after 60-90 min (Skar et al., 1994). In the current study, following vehicle or SR141716A (20 mg/kg, i. p.) injection, vomiting shrews were subjected to c-fos staining. 90 min after the first emesis occurred, shrews were deeply anesthetized and transcardially perfused. This protocol is described in detail in Section 4.4.2 (see lines 992–995).
- I wonder why THC has not been used in the experiments, since THC is the prototypical cannabinoid, and the classical CB1 agonist. Adding this compound (at least for behavioural studies) would be welcome; at least in Europe, THC as solution for inhalation is licensed for postoperative nausea and vomiting (PONV), and would further confirm the experimental design and conclusions.
We have previously published data showing that three structurally distinct classes of cannabinoid receptor agonists—CP 55,940; WIN 55,212-2; and Δ⁹-THC—can block SR141716A-induced emesis in a dose-dependent manner in least shrews. The rank order of antiemetic ID₅₀ potency was CP 55,940 > WIN 55,212-2 > Δ⁹-THC (Darmani, 2001). So, we did not include THC in this study and we are currently investigating the intracellular signaling components of THC in the field of emesis.
- The manuscript would also gain from inclusion of a control region for the immunohistochemistry; even though I cannot see stains in adjacent areas in most pictures, I also sometimes cannot see stains in control animals, but find a value >0 in the Figures and results. Ideally, for control regions a brain area with no connection to the emesis reflex should be chosen.
We performed the immunohistochemistry experiments in accordance with the behavioral emesis induction studies. Following injection with either vehicle or SR141716A (20 mg/kg, i.p.), shrews were deeply anesthetized and transcardially perfused for immunohistochemical analysis. We then compared the expressions of CB1 receptor, c-fos, ERK1/2, GSK-3α/β, and 5-HT/SP between vehicle- and SR141716A-treated groups. Furthermore, our methodology is also based on our previously published manuscripts.
In Figures 1–5, the treatment condition (vehicle or SR141716A) is indicated in the lower left corner of each image. All tissue slices were stained under the same conditions within each experiment. Only the DVC (dorsal vagal complex) region was activated by SR141716A, indicating that the DVC was involved in SR141716A-induced vomiting. Adjacent regions, such as the cuneate nucleus and hypoglossal nucleus, are also shown in each image. To avoid confusion, only the DVC subregions (AP, DMNX, NTS) are labeled in the images.
- The discussion should be shortened, and possibly focused on larger pathway segments. For me, protein synthesis, protein phosphorylation and Ca flux/release would be three relevant functions; as far as I see the results nicely confirm in short term applications no change in protein amount, but larger differences in phosphorylation, and consecutive changes in Ca flow.
We appreciate the reviewer’s thoughtful suggestion regarding the structure and focus of the Discussion section. While we understand the recommendation to shorten and center the discussion around broader pathway segments such as protein synthesis, phosphorylation, and calcium flux, we believe that the current structure is necessary to fully present and interpret the complex and multifaceted signaling mechanisms involved in SR141716A-induced emesis.
In particular, the Discussion aims to integrate a wide range of experimental findings—including behavioral, immunohistochemical, and pharmacological data—with the relevant literature. This comprehensive approach, although detailed, we believe is essential to support the novel insights regarding the roles of multiple signaling cascades (e.g., cAMP/PKA, ERK1/2, PI3K/Akt/GSK-3, PLC/PKCαβII, and CaMKII) and calcium mobilization pathways in emesis.
Therefore, we respectfully request to retain the current version of the Discussion to preserve the completeness and clarity of the manuscript.
- The text should be checked for mistakes; sometimes a singular (is) is found, whereas a plural (are) should have been used. Please capitalize ELISA.
We completed the singular/plural grammatical changes (we labeled the changes in red). We also made some other minor grammar changes, and capitalized ELISA. Please see the revised manuscript.
Round 2
Reviewer 1 Report
Comments and Suggestions for Authors
I regret to inform you that there have been no substantial modifications concerning the principal points I previously highlighted. As stated in my initial report, although the study possesses clinical significance and potential translational implications, it nonetheless remains limited in terms of novelty.